# On Convergence of Adam for Stochastic Optimization under Relaxed Assumptions

**Yusu Hong**
Center for Data Science
and School of Mathematical Sciences
Zhejiang University
yusuhong@zju.edu.cn

**Junhong Lin**$^*$
Center for Data Science
Zhejiang University
junhong@zju.edu.cn

## Abstract

In this paper, we study Adam in non-convex smooth scenarios with potential unbounded gradients and affine variance noise. We consider a general noise model which governs affine variance noise, bounded noise, and sub-Gaussian noise. We show that Adam with a specific hyper-parameter setup can find a stationary point with a $\tilde{\mathcal{O}}(1/\sqrt{T})$ rate in high probability under this general noise model where $T$ denotes total number iterations, matching the lower rate of stochastic first-order algorithms up to logarithm factors. Moreover, we show that under the same setup, Adam without corrective terms and RMSProp can find a stationary point with a $\tilde{\mathcal{O}}(1/T + \sigma_0/\sqrt{T})$ rate which is adaptive to the noise level $\sigma_0$. We also provide a probabilistic convergence result for Adam under a generalized smooth condition which allows unbounded smoothness parameters and has been illustrated empirically to capture the smooth property of many practical objective functions more accurately.

## 1 Introduction

Since its introduction by [34], the Stochastic Gradient Descent (SGD): $\boldsymbol{x}_{t+1} = \boldsymbol{x}_t - \eta_t \boldsymbol{g}_t$ has achieved significant success in solving the unconstrained stochastic optimization problems:

$$\min_{\boldsymbol{x} \in \mathbb{R}^d} f(\boldsymbol{x}), \quad \text{where} \quad f(\boldsymbol{x}) = \mathbb{E}_{\boldsymbol{\xi}}[f_{\boldsymbol{\xi}}(\boldsymbol{x}, \boldsymbol{\xi})], \tag{1}$$

where $\boldsymbol{\xi}$ is a random variable, $\boldsymbol{g}_t$ is the stochastic gradients and $\eta_t$ is the step-size. From then on, numerous literature focused on the convergence behavior of SGD in various scenarios. Several studies focused on the non-convex smooth scenario where the stochastic gradient $g(\boldsymbol{x})$ is unbiased with affine variance noise, i.e., for some constants $\sigma_0, \sigma_1 \geq 0$ and all $\boldsymbol{x} \in \mathbb{R}^d$,

$$\mathbb{E}[\|g(\boldsymbol{x}) - \nabla f(\boldsymbol{x})\|^2] \leq \sigma_0^2 + \sigma_1^2 \|\nabla f(\boldsymbol{x})\|^2. \tag{2}$$

Under the noise assumption (2), [3] provided an almost-sure convergence bound for SGD. [4] proved that SGD could reach a stationary point with a $\mathcal{O}(1/\sqrt{T})$ rate when step-sizes are tuned by problem-parameters such as the smooth parameter $L$. The theoretical result also revealed that the analysis of SGD under (2) is not essentially different from the bounded noise case [17].

In the popular field of deep learning, a range of variants based on SGD, known as adaptive gradient methods have emerged. These methods employ the past gradients to adaptively tune their step-sizes and are preferred to SGD for minimizing various objective functions due to their efficiency. Among these methods, Adam [23] has been one of the most effective methods empirically. Generally

---

$^*$The corresponding author is Junhong Lin.

38th Conference on Neural Information Processing Systems (NeurIPS 2024).

Table 1: Comparison for existing Adam analyses with ours.

| | FCT | Grad. | Noise | Smooth | $\beta_1, \beta_2$ | $\epsilon$ | Conv. Rate | Conv. Type |
|---|---|---|---|---|---|---|---|---|
| [49] | ✗ | Bounded | Bounded | $L$ | $1-\beta_2 \le c\epsilon^2$ | poly($\frac{1}{\epsilon}$) | $\frac{1}{T}+\sigma^2$ | $\mathbb{E}$ |
| [7] | ✗ | Bounded | Bounded | $L$ | $\beta_{1,t} < \beta_1, \beta_2 = 1-\frac{1}{T}$ | - | $\frac{1}{\sqrt{T}}$ | $\mathbb{E}$ |
| [58] | ✗ | Bounded | - | $L$ | $\beta_2 = 1-\frac{c}{T}$ | poly(log $\frac{1}{\epsilon}$) | $\frac{1}{\sqrt{T}}$ | $\mathbb{E}$ |
| [35] | ✗ | - | Finite Sum Affine | $L$ | $T(\beta_1,\beta_2) \to 0$[1] | - | - | $\mathbb{E}$ |
| [10] | ✗ | Bounded | Bounded | $L$ | $\beta_1 < \beta_2, \beta_2 = 1-\frac{1}{T}$ | poly(log $\frac{1}{\epsilon}$) | $\frac{1}{\sqrt{T}}$ | $\mathbb{E}$ |
| [18] | ✗ | Bounded | **Affine** | $L$ | $\beta_1 = 1-\frac{c}{\sqrt{T}}$ | poly($\frac{1}{\epsilon}$) | $\frac{1}{\sqrt{T}}$ | $\mathbb{E}$ |
| [53] | ✓ | - | Finite Sum Affine | $L$ | $\beta_1 < \sqrt{\beta_2}, \beta_2 = 1-\frac{c}{T}$[3] | - | $\frac{1}{\sqrt{T}}$ | $\mathbb{E}$ |
| [42] | ✗ | - | Finite Sum Affine | $(L_0, L_1)$ | $\beta_1 < \sqrt{\beta_2}$ | - | - | $\mathbb{E}$ |
| [25] | ✓ | - | Sub-Gaussian | $(L_0, L_1)$ | $\beta_1 = 1-\frac{c}{\sqrt{T}}$ | $\frac{1}{\sqrt{\epsilon}}$ | $\frac{1}{\sqrt{T}}$ | **w.h.p.** |
| [40] | ✗ | - | Coordinate-wise Affine | $L$ | $\beta_1 = b\sqrt{\beta_2}, \beta_2 = 1-\frac{c}{T}$ | poly(log $\frac{1}{\epsilon}$) | $\frac{1}{\sqrt{T}}$ | $\mathbb{E}$ |
| [20] | ✓ | - | Coordinate-wise Affine | $L$ | $\beta_1 < \beta_2, \beta_2 = 1-\frac{1}{T}$ | poly(log $\frac{1}{\epsilon}$) | $\frac{1}{\sqrt{T}}$ | **w.h.p.** |
| **Thm. 3.1** | ✓ | - | **Affine** | $L$ | $\beta_1 < \beta_2, \beta_2 = 1-\frac{c}{T}$ | poly(log $\frac{1}{\epsilon}$) | $\frac{1}{\sqrt{T}}$ | **w.h.p.** |
| **Thm. 4.1** | ✓ | - | **Affine** | $(L_0, L_1)$ | $\beta_1 < \beta_2, \beta_2 = 1-\frac{c}{T}$ | poly(log $\frac{1}{\epsilon}$) | $\frac{1}{\sqrt{T}}$ | **w.h.p.** |

[1] [35] requires $T(\beta_1, \beta_2) = \mathcal{O}\left(\frac{\beta_1}{\beta_2^n}\left(\frac{1-\beta_1}{1-\beta_1^n}+1\right)\right) \to 0$, which seems could only achieve when $\beta_1 = 0$ .

[2] Though not explicitly stated, the results in (Zhang et al., 2022) could imply convergence to the stationary point when with some calculations.
[3] "FCT" refers to "full corrective terms". The "Conv. rate" column presents the convergence rate omitting logarithm factors.

speaking, Adam absorbs some key ideas from previous adaptive methods such as AdaGrad [12, 37] and RMSProp [38] while adding more unique structures. It combines the exponential moving average mechanism from RMSProp and meanwhile adds the heavy-ball style momentum [30] and two unique corrective terms. This unique structure leads to a huge success for Adam in practical applications but at the same time brings more challenges to the theoretical analysis.

Considering the significance of affine variance noise and Adam in both theoretical and empirical fields, it's natural to question whether Adam can find a stationary point at a rate comparable to SGD under the same smooth condition and (2). Earlier researches [14, 41, 2] have shown that AdaGrad-Norm, a scalar version of AdaGrad, can find a stationary point at the same rate as SGD, not tuning step-sizes based on problem-parameters. Moreover, they addressed an essential challenge brought by the correlation of adaptive step-sizes and noise from (2) which does not appear in SGD's cases. However, since AdaGrad-Norm applies a cumulative step-sizes mechanism which is rather different from the exponential moving average step-sizes in Adam, the analysis for AdaGrad-Norm could not be trivially extended to Adam. Furthermore, the coordinate-wise step-size architecture of Adam, rather than the unified step-size for all coordinates in AdaGrad-Norm, brings more challenge when considering (2). In affine variance noise landscape, existing literature could only ensure the Adam's convergence with random-reshuffling scheme under certain parameter restrictions [53, 42], or deduce the convergence at the expense of requiring bounded gradient assumption and using problem-parameters to tune the step-sizes [18], both of which ignored the corrective terms. Some other works proved convergence to a stationary point by altering the original Adam algorithm such as removing certain corrective terms and modifying (2) to a stronger coordinate-wise variant [40, 20].

To the best of our knowledge, existing research has not yet fully confirmed the convergence of Adam under affine variance noise. To address this gap, we conduct an in-depth analysis and prove that Adam with the right parameter can find a stationary point in high probability. We assume a milder noise model (detailed in Assumption (A3)), covering almost surely affine variance noise, the bounded noise, and sub-Gaussian noise. We show that the convergence rate can reach at $\mathcal{O}\left(\text{poly}(\log T)/\sqrt{T}\right)$ matching the lower rate in [1] up to logarithm factors. Our proof employs the descent lemma over the introduced proxy iterative sequence and adopts techniques related to the new proxy step-sizes and error decomposition. Based on this, we are able to handle the correlation between stochastic gradients and adaptive step-sizes and transform the first-order term from the descent lemma into the gradient norm.

Finally, we apply the analysis to the $(L_0, L_q)$-smooth condition [52]. Several researchers have found empirical evidence of objective functions satisfying $(L_0, L_q)$-smoothness but out of $L$-smoothness range, especially in large-scale language models [50, 39, 11, 8]. Theoretical analysis of adaptive methods under this relaxed condition is more complicated and needs further nontrivial proof techniques. Also, prior knowledge of problem-parameters to tune step-sizes is needed, as indicated by the counter-examples from [41] for the AdaGrad. Existing works [13, 41] obtained a convergence bound for AdaGrad-Norm with (2), and [25] considered Adam with sub-Gaussian noise. In this

---

**Algorithm 1** Adam

---

**Input:** Horizon $T$, $\boldsymbol{x}_1 \in \mathbb{R}^d$, $\beta_1, \beta_2 \in [0, 1)$, $\boldsymbol{m}_0 = \boldsymbol{v}_0 = \boldsymbol{0}_d$, $\eta, \epsilon > 0$, $\boldsymbol{\epsilon} = \epsilon \boldsymbol{1}_d$
**for** $s = 1, \cdots, T$ **do**
  Draw a new sample $\boldsymbol{z}_s$ and generate $\boldsymbol{g}_s = g(\boldsymbol{x}_s, \boldsymbol{z}_s)$;
  $\boldsymbol{m}_s = \beta_1 \boldsymbol{m}_{s-1} + (1 - \beta_1) \boldsymbol{g}_s$;
  $\boldsymbol{v}_s = \beta_2 \boldsymbol{v}_{s-1} + (1 - \beta_2) \boldsymbol{g}_s^2$;
  $\eta_s = \eta \sqrt{1 - \beta_2^s}/(1 - \beta_1^s)$, $\boldsymbol{\epsilon}_s = \boldsymbol{\epsilon}\sqrt{1 - \beta_2^s}$;
  $\boldsymbol{x}_{s+1} = \boldsymbol{x}_s - \eta_s \cdot \boldsymbol{m}_s / \left( \sqrt{\boldsymbol{v}_s} + \boldsymbol{\epsilon}_s \right)$;
**end for**

---

paper, we provide a probabilistic convergence result for Adam with the affine variance noise and the generalized smoothness condition.

We also refer readers to see the main contributions of our works and comparisons with the existing works in Table 1.

**Notations** We use $[T]$ to denote the set $\{1, 2, \cdots, T\}$ for any positive integer $T$, $\|\cdot\|, \|\cdot\|_1$ and $\|\cdot\|_\infty$ to denote $l_2$-norm, $l_1$-norm and $l_\infty$-norm respectively. $a \sim \mathcal{O}(b)$ and $a \leq \mathcal{O}(b)$ denote $a = C_1 b$ and $a \leq C_2 b$ for some positive universal constants $C_1, C_2$, and $a \leq \tilde{\mathcal{O}}(b)$ denotes $a \leq \mathcal{O}(b)\text{poly}(\log b)$. $a \lesssim b$ denotes $a \leq \mathcal{O}(b)$. For any vector $\boldsymbol{x} \in \mathbb{R}^d$, $\boldsymbol{x}^2$ and $\sqrt{\boldsymbol{x}}$ denote coordinate-wise square and square root respectively. $\boldsymbol{x}_i$ denotes the $i$-th coordinate of $\boldsymbol{x}$. For any two vectors $\boldsymbol{x}, \boldsymbol{y} \in \mathbb{R}^d$, we use $\boldsymbol{x} \odot \boldsymbol{y}$ and $\boldsymbol{x}/\boldsymbol{y}$ to denote the coordinate-wise product and quotient respectively. $\boldsymbol{0}_d$ and $\boldsymbol{1}_d$ represent zero and one $d$-dimensional vectors respectively.

## 2 Problem set up and algorithm

We consider unconstrained stochastic optimization (1) over $\mathbb{R}^d$ with $l_2$-norm. The objective function $f : \mathbb{R}^d \to \mathbb{R}$ is differentiable. Given $\boldsymbol{x} \in \mathbb{R}^d$, we assume a gradient oracle that returns a random vector $g(\boldsymbol{x}, \boldsymbol{z}) \in \mathbb{R}^d$ dependent by the random sample $\boldsymbol{z}$. The true gradient of $f$ at $\boldsymbol{x}$ is denoted by $\nabla f(\boldsymbol{x}) \in \mathbb{R}^d$.

**Assumptions.** We make the following assumptions throughout the paper.

- **(A1)** Bounded below: There exists $f^* > -\infty$ such that $f(\boldsymbol{x}) \geq f^*, \forall \boldsymbol{x} \in \mathbb{R}^d$;

- **(A2)** Unbiased estimator: The gradient oracle provides an unbiased estimator of $\nabla f(\boldsymbol{x})$, i.e., $\mathbb{E}_{\boldsymbol{z}}\left[g(\boldsymbol{x}, \boldsymbol{z})\right] = \nabla f(\boldsymbol{x}), \forall \boldsymbol{x} \in \mathbb{R}^d$;

- **(A3)** Generalized affine variance noise: The gradient oracle satisfies that there are some constants $\sigma_0, \sigma_1 > 0, p \in [0, 4)$, $\mathbb{E}_{\boldsymbol{z}}\left[\exp\left(\frac{\|g(\boldsymbol{x}, \boldsymbol{z}) - \nabla f(\boldsymbol{x})\|^2}{\sigma_0^2 + \sigma_1^2 \|\nabla f(\boldsymbol{x})\|^p}\right)\right] \leq \exp(1), \forall \boldsymbol{x} \in \mathbb{R}^d$.

The first two assumptions are standard in the stochastic optimization. The third assumption provides a mild noise model that covers the almost surely bounded noise and sub-Gaussian noise. Moreover, it's more general than almost surely affine variance noise as follows

$$\|g(\boldsymbol{x}, \boldsymbol{z}) - \nabla f(\boldsymbol{x})\|^2 \leq \sigma_0^2 + \sigma_1^2 \|\nabla f(\boldsymbol{x})\|^2, a.s., \tag{3}$$

and enlarge the range of $p$ to $[0, 4)$. Assumption **(A3)** with $p = 2$ and (3) are also utilized in [2] to establish high probability results for AdaGrad-Norm. It represents a stronger condition than the expected version of (2) that is commonly employed for deriving the expected convergence of algorithms. However, almost surely assumption enables the derivation of stronger high-probability convergence guarantees for algorithms, while still ensuring expected convergence.

The affine noise variance assumption is important for machine learning applications with feature noise (including missing features) [15, 22], in robust linear regression [46], and generally whenever the model parameters are multiplicatively perturbed by noise (e.g., a multilayer network, where noise from a previous layer multiplies the parameters in subsequent layers). We refer interested readers to see e.g., [3, 46, 4, 14, 41, 2] for more discussions about the affine variance noise.

**Adam.** For the stochastic optimization problem, we study Algorithm 1, which is an equivalent form of Adam [23] with the two corrective terms for $\boldsymbol{m}_s$ and $\boldsymbol{v}_s$ included into $\eta_s$ for notation simplicity.

The iterative relationship in Algorithm 1 can be also written as for any $s \in [T]$,

$$\boldsymbol{x}_{s+1} = \boldsymbol{x}_s - \eta_s(1-\beta_1) \cdot \frac{\boldsymbol{g}_s}{\sqrt{\boldsymbol{v}_s} + \boldsymbol{\epsilon}_s} + \beta_1 \cdot \frac{\eta_s(\sqrt{\boldsymbol{v}_{s-1}} + \boldsymbol{\epsilon}_{s-1})}{\eta_{s-1}(\sqrt{\boldsymbol{v}_s} + \boldsymbol{\epsilon}_s)} \odot (\boldsymbol{x}_s - \boldsymbol{x}_{s-1}), \tag{4}$$

where we let $\boldsymbol{x}_0 = \boldsymbol{x}_1$ and $\eta_0 = \eta$. (4) plays a key role in the convergence analysis, showing that Adam incorporates a heavy-ball style momentum and dynamically adjusts its momentum through $\beta_1$ and $\beta_2$, along with adaptive step-sizes. This inspires us to learn from some classical analysis methods for algorithms with momentum and provides some new estimations to fit in with the adaptive property.

## 3 Convergence of Adam with smooth objective functions

In this section, we assume that the objective function $f$ is $L$-smooth satisfying that for any $\boldsymbol{x}, \boldsymbol{y} \in \mathbb{R}^d$,

$$\|\nabla f(\boldsymbol{y}) - \nabla f(\boldsymbol{x})\| \leq L\|\boldsymbol{y} - \boldsymbol{x}\|. \tag{5}$$

We then show that Adam has the following high probability results.

**Theorem 3.1.** *Let $T \geq 1$ and $\{\boldsymbol{x}_s\}_{s \in [T]}$ be the sequence generated by Algorithm 1. If Assumptions (A1)-(A3) hold, and the hyper-parameters satisfy that*

$$0 \leq \beta_1 < \beta_2 < 1, \quad \beta_2 = 1 - c/T, \quad \eta = C_0\sqrt{1-\beta_2}, \quad \epsilon = \epsilon_0\sqrt{1-\beta_2}, \tag{6}$$

*for some constants $c, C_0 > 0$ and $\epsilon_0 > 0$, then for any given $\delta \in (0, 1/2)$, it holds that with probability at least $1 - 2\delta$,*

$$\frac{1}{T}\sum_{s=1}^{T}\|\nabla f(\boldsymbol{x}_s)\|^2 \leq \mathcal{O}\left\{G^2\left(\sqrt{\frac{\sigma_0^2 + \sigma_1^2 G^p + G^2}{T}} + \frac{\epsilon_0}{T}\right)\log\left(\frac{T}{\delta}\right)\right\},$$

*where $G^2$ is defined by the following order with respect to $T, \epsilon_0, \delta$:[2]*

$$G^2 \sim \mathcal{O}\left(\log^{\frac{3}{2}\max\{2, \frac{4}{4-p}\}}\left(\frac{T}{\epsilon_0\delta}\right)\right). \tag{7}$$

Theorem 3.1 provides the nearly optimal convergence rate $\mathcal{O}\left(\text{poly}(\log T)/\sqrt{T}\right)$ to find a stationary point when setting the parameter probably: $\beta_2 = 1 - \mathcal{O}(1/T)$. It's worth noting that the setting requires $\beta_2$ to be closed enough to 1 when $T$ is sufficiently large, which roughly aligns with the typical setting in [23, 58, 10, 40]. For a more detailed comparison of our results to existing works, including assumptions, convergence rate, and dependency, we refer readers to Table 1.

**Adam without corrective terms and RMSProp.** We also consider a simplified version of Adam that drops two corrective terms as shown in Algorithm 2 in Appendix. Algorithm 2 with $\beta_1 = 0$ can be directly reduced to RMSProp [38]. More importantly, the following result shows that the convergence rate of Algorithm 2 is adaptive to the noise level.

**Theorem 3.2** (informal version of Theorem C.2). *Let $T \geq 1$ and $\{\boldsymbol{x}_s\}_{s \in [T]}$ be generated by Algorithm 2 covering RMSProp. Following the assumptions and setup in Theorem 3.1, with probability at least $1 - 2\delta$, $\sum_{s=1}^{T}\|\nabla f(\boldsymbol{x}_s)\|^2/T \lesssim \tilde{\mathcal{O}}(1/T + \sigma_0/\sqrt{T})$.*

## 4 Convergence of Adam with generalized smooth objective functions

In this section, we study the convergence behavior of Adam in the generalized smooth case. We first provide some necessary introduction to the generalized smooth condition.

---

[2]The detailed expression of $G^2$ could be found in (53) from Appendix.

## 4.1 Generalized smoothness

We consider the following $(L_0, L_q)$-smoothness condition: there exist constants $q \in [0, 2)$ and $L_0, L_q > 0$, satisfying that for any $\boldsymbol{x}, \boldsymbol{y} \in \mathbb{R}^d$ with $\|\boldsymbol{x} - \boldsymbol{y}\| \leq 1/L_q$,

$$\|\nabla f(\boldsymbol{y}) - \nabla f(\boldsymbol{x})\| \leq (L_0 + L_q \|\nabla f(\boldsymbol{x})\|^q) \|\boldsymbol{x} - \boldsymbol{y}\|. \tag{8}$$

The generalized smooth condition was originally put forward by [52] for any twice differentiable function $f$ satisfying that

$$\|\nabla^2 f(\boldsymbol{x})\| \leq L_0 + L_1 \|\nabla f(\boldsymbol{x})\|. \tag{9}$$

It has been proved that a lot of objective functions in experimental areas satisfy (9) but out of $L$-smoothness range, especially in training large language models, see e.g., Figure 1 in [52] and [8].

To better understand the theoretical significance of the generalized smoothness, [50] provided an alternative form in (8) with $q = 1$, only requiring $f$ to be differentiable. They showed that (8) is sufficient to elucidate the convergence of gradient-clipping algorithms.

There are three key reasons for opting for (8). Firstly, considering our access is limited to first-order stochastic gradients, it's logical to only assume that $f$ is differentiable. Second, as pointed out by Lemma A.2 in [50] and Proposition 1 in [13], (8) and (9) are equivalent up to constant factors when $f$ is twice differentiable considering $q = 1$. Thus, (8) covers a broader range of functions than (9). Finally, it's easy to verify that (8) is strictly weaker than $L$-smoothness. A concrete example is that the simple function $f(x) = x^4, x \in \mathbb{R}$ does not satisfy any global $L$-smoothness but (8). Moreover, the expanded range of $q$ to $[0, 2)$ is necessary as all univariate rational functions $P(x)/Q(x)$, where $P, Q$ are polynomials and double exponential functions $a^{(b^x)}$ with $a, b > 1$ are $(L_0, L_q)$-smooth with $1 < q < 2$ (see [25, Proposition 3.4]). We refer interested readers to see [52, 50, 13, 25] for more discussions of concrete examples of generalized smoothness.

## 4.2 Convergence result

We then provide the high probability convergence result of Adam with $(L_0, L_q)$-smoothness condition as follows.

**Theorem 4.1.** *Let $T \geq 1$ and $\delta \in (0, 1/2)$. Suppose that $\{\boldsymbol{x}_s\}_{s \in [T]}$ is a sequence generated by Algorithm 1, $f$ is $(L_0, L_q)$-smooth satisfying (8), Assumptions (A1)-(A3) hold, and the parameters satisfy*

$$0 \leq \beta_1 < \beta_2 < 1, \quad \beta_2 = 1 - c/T, \quad \epsilon = \epsilon_0 \sqrt{1 - \beta_2}, \quad \eta = \tilde{C}_0 \sqrt{1 - \beta_2},$$

$$\tilde{C}_0 \leq \min \left\{ E_0, \frac{E_0}{\mathcal{H}}, \frac{E_0}{\mathcal{L}}, \sqrt{\frac{\beta_2 (1 - \beta_1)^2 (1 - \beta_1/\beta_2)}{4 L_q^2 d}} \right\}, \tag{10}$$

*where $c, \epsilon_0, E_0, \tilde{C}_0 > 0$ are constants, $\hat{H}$ is controlled by $\mathcal{O}\left( \log\left( \frac{T}{\epsilon_0 \delta} \right) \right)$ [3], and $H, \mathcal{H}, \mathcal{L}$ are defined as*

$$H := L_0/L_q + \left( 4 L_q \hat{H} \right)^q + \left( 4 L_q \hat{H} \right)^{\frac{q}{2-q}} + \left( 4 L_0 \hat{H} \right)^{\frac{q}{2}} + 4 L_q \hat{H} + \left( 4 L_q \hat{H} \right)^{\frac{1}{2-q}} + \sqrt{4 L_0 \hat{H}},$$

$$\mathcal{H} := \sqrt{2 (\sigma_0^2 + \sigma_1^2 H^p + H^2) \log\left( \frac{eT}{\delta} \right)}, \quad \mathcal{L} := L_0 + L_q \left( H^q + H + \frac{L_0}{L_q} \right)^q. \tag{11}$$

*Then it holds that with probability at least $1 - 2\delta$,*

$$\frac{1}{T} \sum_{s=1}^{T} \|\nabla f(\boldsymbol{x}_s)\|^2 \leq \mathcal{O} \left\{ \frac{\hat{H}}{\tilde{C}_0} \left( \sqrt{\frac{\sigma_0^2 + \sigma_1^2 H^p + H^2}{T}} + \frac{\epsilon_0}{T} \right) \log\left( \frac{T}{\delta} \right) \right\}. \tag{12}$$

Note that in the above theorem, the order of $\log T$ in $\hat{H}$ and the final convergence bound is better than the one in Theorem 3.1 under the same noise assumption. This better dependency comes from the expense of using problem parameters to tune step-size $\tilde{C}_0$. Since $\hat{H}$ is logarithm order

---

[3] The specific definition of $\hat{H}$ can be found in (115) from Appendix.

of $T$, $H, \mathcal{H}, \mathcal{L}$ are both polynomial logarithm order of $T$ and the final convergence rate in (12) is $\mathcal{O}(\text{poly}(\log T)/\sqrt{T})$ order. Note that $\tilde{C}_0 \leq \mathcal{O}(1/\text{poly}(\log T))$ from (10) when $T \gg d$. Hence, when $T$ is large enough, a possible optimal setting is that $\eta = c_1/(\sqrt{T}\text{poly}(\log T))$ for some constant $c_1 > 0$, which roughly matches the typical setting as mentioned before.

Similarly, we also obtain a convergence bound that is adaptive to the noise level for Adam without corrective terms and RMSProp under generalized smoothness.

**Theorem 4.2** (informal version of Theorem E.1). *Let $T \geq 1$ and $\{\boldsymbol{x}_s\}_{s \in [T]}$ be generated by Algorithm 2 covering RMSProp. Following the assumptions and the same order of hyper-parameters in Theorem 4.1, with probability at least $1 - 2\delta$, $\sum_{s=1}^{T} \|\nabla f(\boldsymbol{x}_s)\|^2/T \lesssim \tilde{\mathcal{O}}(1/T + \sigma_0/\sqrt{T})$.*

## 5   Related works

There is a large amount of works on stochastic approximations (or online learning algorithms) and adaptive variants, e.g., [5, 36, 48, 29, 12, 4, 6, 27, 55] and the references therein. In this section, we will discuss the most related works and make a comparison with our main results.

### 5.1   Convergence with affine variance noise and its variants

We mainly list previous literature considering (2) over non-convex smooth scenario. [3] provided an asymptotic convergence result for SGD with (2). In terms of non-asymptotic results, [4] proved the convergence of SGD, illustrating that the analysis was non-essentially different from the bounded noise case from [17].

In the adaptive methods field, [14] studied convergence of AdaGrad-Norm with (2), pointing out that the analysis is more challenging than the bounded noise and bounded gradient case in [44]. They provided a convergence rate of $\tilde{\mathcal{O}}(1/\sqrt{T})$ without knowledge of problem parameters, and further improved the bound adapting to the noise level: when $\sigma_1 \sim \mathcal{O}(1/\sqrt{T})$,

$$\frac{1}{T} \sum_{t=1}^{T} \mathbb{E}\|\nabla f(\boldsymbol{x}_t)\|^2 \leq \tilde{\mathcal{O}}\left(\frac{\sigma_0}{\sqrt{T}} + \frac{1}{T}\right). \tag{13}$$

(13) matches exactly with SGD's case [4], showing a fast rate of $\tilde{\mathcal{O}}(1/T)$ when $\sigma_0$ is sufficiently low. Later, [41] proposed a deep analysis framework obtaining (13) with a tighter dependency to $T$ and not requiring any restriction over $\sigma_1$. They further obtained the same rate for AdaGrad under a stronger coordinate-wise version of (2): for all $i \in [d]$,

$$\mathbb{E}_{\boldsymbol{z}}|\boldsymbol{g}(\boldsymbol{x}, \boldsymbol{z})_i - \nabla f(\boldsymbol{x})_i|^2 \leq \sigma_0^2 + \sigma_1^2|\nabla f(\boldsymbol{x})_i|^2. \tag{14}$$

[2] obtained a probabilistic convergence rate for AdaGrad-Norm with (3) using a novel induction argument to estimate the function value gap without any requirement over $\sigma_1$ as well.

In the analysis of Adam, a line of works [35, 53, 42] considered Adam for finite-sum objective functions under different regimes while possibly incorporating natural random shuffling technique. They could ensure that this variant converged to a bounded region where

$$\min_{t \in [T]} \mathbb{E}\left[\min\{\|\nabla f(\boldsymbol{x}_t)\|, \|\nabla f(\boldsymbol{x}_t)\|^2\}\right] \lesssim \frac{\log T}{\sqrt{T}} + C_1\sigma_0 \tag{15}$$

under the affine growth condition which is equivalent to (2). Though not explicitly concluded, when setting $\beta_2 = 1 - \mathcal{O}(1/T)$, [53]'s work can also ensure a convergence rate of order $\tilde{\mathcal{O}}(1/\sqrt{T})$ under certain settings. Besides, both [21] and [18] provided convergence bounds allowing for large heavy-ball momentum parameter that aligns more closely with practical settings. However, they relied on the assumption for step-sizes where $C_l \leq \|\frac{1}{\sqrt{\boldsymbol{v}_t} + \boldsymbol{\epsilon}_t}\|_\infty \leq C_u, \forall t \in [T]$. [40] and [20] used distinct methods to derive convergence bounds in expectation and high probability respectively, without relying on bounded gradients. Both studies achieved a convergence rate of the form in (13) for Adam ignoring the corrective terms. [20] further achieved a $\tilde{\mathcal{O}}(1/\sqrt{T})$ rate for Adam. However, the two works only studied coordinate-wise affine variance noise.

In this paper, we derive a stronger high probability convergence rate for Adam with original corrective terms, relying on an almost surely noise assumption. The noise model is general enough to

cover bounded noise, sub-Gaussian noise, and (coordinate-wise) affine variance noise. Although we consider a stronger almost surely assumption, our probabilistic convergence result is also stronger than the expected convergence.

## 5.2 Convergence with generalized smoothness

The generalized smooth condition was first proposed for twice differentiable functions by [52] (see (9)) to explain the acceleration mechanism of gradient-clipping. This assumption was extensively confirmed in experiments of large-scale language models [52]. Later, [50] further relaxed it to a more general form in (8) allowing for first-order differentiable functions. Subsequently, a series of works [31, 54, 33] studied different algorithms' convergence under this condition.

In the field of adaptive methods, [13] provided a convergence bound for AdaGrad-Norm assuming (2) and (8) with $q = 1$, albeit requiring $\sigma_1 < 1$. Based on the same conditions, [41] improved the convergence rate to the form in (13) without restriction on $\sigma_1$. [42] explored how Adam without corrective terms behaves under generalized smoothness with $q = 1$ and (2). However, they could only assert convergence to a bounded region as shown in (15). [8] showed that an Adam-type algorithm converges to a stationary point under a stronger coordinate-wise generalized smooth condition. Recently, [25] provided a novel framework to derive high probability convergence bound for Adam under the generalized smooth and sub-Gaussian noise case.

In this paper, we consider a more general noise setup and investigate Adam's convergence under the generalized smooth landscape. We prove that Adam is powerful enough to find a stationary point with properly tuned step-sizes even under these relaxed assumptions. Moreover, the convergence rate is not harmed by the relaxation of noise and smoothness, matching the optimal $\mathcal{O}(1/\sqrt{T})$ rate up to logarithm factors.

## 5.3 Convergence of Adam

Adam was first proposed by [23] with empirical studies and theoretical results on online convex learning. The original proof of convergence in [23] was later shown by [32] to contain gaps. [32] and the subsequent work [43] also showed that for a range of momentum parameters chosen independently with the problem instance, Adam does not necessarily converge even for convex objectives. Many works have focused on its convergence behavior in non-convex smooth fields. A series of works studied Adam ignoring corrective terms, all requiring a uniform bound for gradients' norm. Among these works, [49] demonstrated that Adam can converge within a specific region if step-sizes and decay parameters are determined properly by the smooth parameter. [9] proposed a convergence result to a stationary point and required all stochastic gradients must keep the same sign. To circumvent this requirement, [58] introduced a convergence bound only requiring hyper-parameters to satisfy specific conditions. [10] conducted a simple proof and further improved the dependency on the heavy-ball momentum parameter. Recently, [56] introduced Nesterov-like acceleration into Adam and AdamW [28] indicating their superiority in convergence over the non-accelerated versions. For Adam-related works under (2) or generalized smoothness, we refer readers to Sections 5.1 and 5.2.

We also want to highlight that a series of works [24, 45, 51] investigated the geometry of Adam from an $l_\infty$-norm perspective. [24] and [45] studied the geometry of Adam by regarding it as a variant of SignSGD and [51] showed that full-batch Adam converges towards a linear classifier that achieves the maximum $l_\infty$-margin when the training data are linearly separable.

## 6 Proof sketch under the smooth case

In this section, we provide a proof sketch of Theorem 3.1 with some insights and proof novelty. Our proof borrows some ideas from [44, 10, 14, 2, 40, 20]. The detailed proof is in Appendix B.

**Preliminary.** To start with, we let the stochastic gradient $\boldsymbol{g}_s = (g_{s,i})_i$, the true gradient $\nabla f(\boldsymbol{x}_s) = \bar{\boldsymbol{g}}_s = (\bar{g}_{s,i})_i$ and $\boldsymbol{\xi}_s = (\xi_{s,i})_i = \boldsymbol{g}_s - \bar{\boldsymbol{g}}_s$. We also let $\epsilon_s = \epsilon\sqrt{1 - \beta_2^s}$ and thus $\boldsymbol{\epsilon}_s = \epsilon_s \mathbf{1}_d$. For any positive integer $T$ and $\delta \in (0, 1)$, we define $\mathcal{M}_T = \sqrt{\log(\mathrm{e}T/\delta)}$. We denote the adaptive part of the step-size as

$$\boldsymbol{b}_s := \sqrt{\boldsymbol{v}_s} + \boldsymbol{\epsilon}_s = \sqrt{\beta_2 \boldsymbol{v}_{s-1} + (1 - \beta_2)\boldsymbol{g}_s^2} + \boldsymbol{\epsilon}_s. \tag{16}$$

We define two auxiliary sequences $\{\boldsymbol{p}_s\}_{s\geq 1}$ and $\{\boldsymbol{y}_s\}_{s\geq 1}$,

$$\boldsymbol{p}_1 = \boldsymbol{0}_d, \quad \boldsymbol{y}_1 = \boldsymbol{x}_1, \quad \boldsymbol{p}_s = \frac{\beta_1}{1-\beta_1}(\boldsymbol{x}_s - \boldsymbol{x}_{s-1}), \boldsymbol{y}_s = \boldsymbol{p}_s + \boldsymbol{x}_s, \forall s \geq 2. \qquad (17)$$

We follow from [16, 47] which was used to prove the convergence of SGD with momentum and later applied to handle many variants of momentum-based algorithms. Recalling the iteration of $\boldsymbol{x}_s$ in (4), we reveal that $\boldsymbol{y}_s$ satisfies

$$\boldsymbol{y}_{s+1} = \boldsymbol{y}_s - \eta_s \cdot \frac{\boldsymbol{g}_s}{\boldsymbol{b}_s} + \frac{\beta_1}{1-\beta_1}\left(\frac{\eta_s \boldsymbol{b}_{s-1}}{\eta_{s-1}\boldsymbol{b}_s} - \boldsymbol{1}_d\right) \odot (\boldsymbol{x}_s - \boldsymbol{x}_{s-1}). \qquad (18)$$

In addition, given $T \geq 1$, we define, $\forall s \in [T]$,

$$G_s = \max_{j\in[s]}\|\bar{\boldsymbol{g}}_j\|, \mathcal{G}_T(s) = \mathcal{M}_T\sqrt{2\sigma_0^2 + 2\sigma_1^2 G_s^p + 2G_s^2}, \mathcal{G}_T = \mathcal{M}_T\sqrt{2\sigma_0^2 + 2\sigma_1^2 G^p + 2G^2}, \quad (19)$$

where $G$ is as in Theorem 3.1. Both $G_s$ and $\mathcal{G}_T(s)$ will serve as upper bounds for gradients' norm before time $s$. We will verify their importance in the later argument.

**Starting from the descent lemma.** We fix the horizon $T$ and start from the standard descent lemma of $L$-smoothness. Then, for any given $t \in [T]$, combining with (18) and summing over $s \in [t]$,

$$f(\boldsymbol{y}_{t+1}) \leq f(\boldsymbol{x}_1) + \underbrace{\sum_{s=1}^{t} -\eta_s \left\langle \nabla f(\boldsymbol{y}_s), \frac{\boldsymbol{g}_s}{\boldsymbol{b}_s}\right\rangle}_{\mathbf{A}} + \underbrace{\frac{\beta_1}{1-\beta_1}\sum_{s=1}^{t}\langle \Delta_s \odot (\boldsymbol{x}_s - \boldsymbol{x}_{s-1}), \nabla f(\boldsymbol{y}_s)\rangle}_{\mathbf{B}}$$

$$+ \underbrace{\frac{L}{2}\sum_{s=1}^{t}\left\|\eta_s \cdot \frac{\boldsymbol{g}_s}{\boldsymbol{b}_s} - \frac{\beta_1}{1-\beta_1}(\Delta_s \odot (\boldsymbol{x}_s - \boldsymbol{x}_{s-1}))\right\|^2}_{\mathbf{C}}, \qquad (20)$$

where we let $\Delta_s = \frac{\eta_s \boldsymbol{b}_{s-1}}{\eta_{s-1}\boldsymbol{b}_s} - \boldsymbol{1}_d$ and use $\boldsymbol{y}_1 = \boldsymbol{x}_1$ from (17). In what follows, we will estimate $\mathbf{A}$, $\mathbf{B}$, and $\mathbf{C}$ respectively.

**Probabilistic estimations.** To proceed with the analysis, we next introduce two probabilistic estimations showing that the norm of the noises and a related summation of martingale difference sequence could be well controlled with high probability. We show that with probability at least $1 - 2\delta$, the following two inequalities hold simultaneously for all $t \in [T]$:

$$\|\boldsymbol{\xi}_t\|^2 \leq \mathcal{M}_T^2\left(\sigma_0^2 + \sigma_1^2\|\bar{\boldsymbol{g}}_t\|^p\right), \quad \text{and} \qquad (21)$$

$$-\sum_{s=1}^{t}\eta_s\left\langle \bar{\boldsymbol{g}}_s, \frac{\boldsymbol{\xi}_s}{\boldsymbol{a}_s}\right\rangle \leq \frac{\mathcal{G}_T(t)}{4\mathcal{G}_T}\sum_{s=1}^{t}\eta_s\left\|\frac{\bar{\boldsymbol{g}}_s}{\sqrt{\boldsymbol{a}_s}}\right\|^2 + D_1\mathcal{G}_T, \qquad (22)$$

where $D_1$ is a constant defined in Lemma B.7 and $\boldsymbol{a}_s$ will be introduced later. In what follows, we always assume that (21) and (22) hold for all $t \in [T]$ and carry out our subsequent analysis with some deterministic estimations.

**Estimating A.** We first decompose $\mathbf{A}$ as

$$\mathbf{A} = \underbrace{\sum_{s=1}^{t} -\eta_s\left\langle \bar{\boldsymbol{g}}_s, \frac{\boldsymbol{g}_s}{\boldsymbol{b}_s}\right\rangle}_{\mathbf{A.1}} + \underbrace{\sum_{s=1}^{t}\eta_s\left\langle \bar{\boldsymbol{g}}_s - \nabla f(\boldsymbol{y}_s), \frac{\boldsymbol{g}_s}{\boldsymbol{b}_s}\right\rangle}_{\mathbf{A.2}}.$$

Due to the correlation of the stochastic gradient $\boldsymbol{g}_s$ and the step-size $\eta_s/\boldsymbol{b}_s$, the estimating of $\mathbf{A.1}$ is challenging, as also noted in the analysis for other adaptive gradient methods, e.g., [44, 10, 14, 2, 40, 20]. To break this correlation, the so-called proxy step-size technique is introduced and variants of proxy step-size have been introduced in the related literature. However, to our best knowledge, none

of these proxy step-sizes could be used in our analysis for Adam considering potential unbounded gradients under the noise model in Assumption (A3). In this paper, we construct a proxy step-size $\eta_s/\boldsymbol{a}_s$, with $\boldsymbol{a}_s$ relying on $\mathcal{G}_T(s)$ in (19), defined as for any $s \in [T]$,

$$\boldsymbol{a}_s = \sqrt{\beta_2 \boldsymbol{v}_{s-1} + (1-\beta_2)\left(\mathcal{G}_T(s)\mathbf{1}_d\right)^2} + \boldsymbol{\epsilon}_s. \tag{23}$$

With the so-called proxy step-size technique over $\eta_s/\boldsymbol{a}_s$ and $\boldsymbol{\xi}_s = \boldsymbol{g}_s - \bar{\boldsymbol{g}}_s$, we decompose **A.1** as

$$\mathbf{A.1} = -\sum_{s=1}^{t} \eta_s \left\|\frac{\bar{\boldsymbol{g}}_s}{\sqrt{\boldsymbol{a}_s}}\right\|^2 \underbrace{-\sum_{s=1}^{t} \eta_s \left\langle \bar{\boldsymbol{g}}_s, \frac{\boldsymbol{\xi}_s}{\boldsymbol{a}_s}\right\rangle}_{\mathbf{A.1.1}} + \underbrace{\sum_{s=1}^{t} \eta_s \left\langle \bar{\boldsymbol{g}}_s, \left(\frac{1}{\boldsymbol{a}_s} - \frac{1}{\boldsymbol{b}_s}\right)\boldsymbol{g}_s\right\rangle}_{\mathbf{A.1.2}}.$$

In the above decomposition, the first term serves as a descent term. **A.1.1** is now a summation of a martingale difference sequence which could be estimated by (22). **A.1.2** is regarded as an error term when introducing $\boldsymbol{a}_s$. However, due to the delicate construction of $\boldsymbol{a}_s$, the definition of local gradients' bound $\mathcal{G}_T(t)$, and using some basic inequalities, we show that

$$\mathbf{A.1.2} \leq \frac{1}{4}\sum_{s=1}^{t} \eta_s \left\|\frac{\bar{\boldsymbol{g}}_s}{\sqrt{\boldsymbol{a}_s}}\right\|^2 + \frac{\eta\mathcal{G}_T(t)\sqrt{1-\beta_2}}{1-\beta_1}\sum_{s=1}^{t} \left\|\frac{\boldsymbol{g}_s}{\boldsymbol{b}_s}\right\|^2.$$

The first RHS term can be eliminated with the descent term while the summation of the last term can be bounded by

$$\sum_{s=1}^{t} \left\|\frac{\boldsymbol{g}_s}{\boldsymbol{b}_s}\right\|^2 \vee \sum_{s=1}^{t} \left\|\frac{\boldsymbol{m}_s}{\boldsymbol{b}_s}\right\|^2 \vee \sum_{s=1}^{t} \left\|\frac{\boldsymbol{m}_s}{\boldsymbol{b}_{s+1}}\right\|^2 \vee \sum_{s=1}^{t} \left\|\frac{\hat{\boldsymbol{m}}_s}{\boldsymbol{b}_s}\right\| \lesssim \frac{d}{1-\beta_2}\log\left(\frac{T}{\beta_2^T}\right), \tag{24}$$

due to the step-size's adaptivity, the iterative relationship of the algorithm, the smoothness of the objective function, as well as (21). Here, $\hat{\boldsymbol{m}}_s = \frac{\boldsymbol{m}_s}{1-\beta_1^s}$.

**Estimating B and C.** The key to estimate **B** is to decompose **B** as

$$\mathbf{B} = \underbrace{\frac{\beta_1}{1-\beta_1}\sum_{s=1}^{t} \langle \Delta_s \odot (\boldsymbol{x}_s - \boldsymbol{x}_{s-1}), \bar{\boldsymbol{g}}_s\rangle}_{\mathbf{B.1}} + \underbrace{\frac{\beta_1}{1-\beta_1}\sum_{s=1}^{t} \langle \Delta_s \odot (\boldsymbol{x}_s - \boldsymbol{x}_{s-1}), \nabla f(\boldsymbol{y}_s) - \bar{\boldsymbol{g}}_s\rangle}_{\mathbf{B.2}}.$$

To estimate **B.1**, we use the updated rule and further decompose $\Delta_s \odot (\boldsymbol{x}_s - \boldsymbol{x}_{s-1})$ as the following terms

$$-\left(\frac{\eta_s}{\boldsymbol{b}_s} - \frac{\eta_s}{\boldsymbol{a}_s}\right)\odot\boldsymbol{m}_{s-1} - \left(\frac{\eta_s}{\boldsymbol{a}_s} - \frac{\eta_s}{\boldsymbol{b}_{s-1}}\right)\odot\boldsymbol{m}_{s-1} - (\eta_s - \eta_{s-1})\frac{\boldsymbol{m}_{s-1}}{\boldsymbol{b}_{s-1}},$$

and upper bound the three related inner products. Using some basic inequalities, the smoothness, (24), and some delicate computations, one can estimate the three related inner products, **B.2** and **C**, and thus get that

$$\mathbf{B} + \mathbf{C} \leq \frac{1}{4}\sum_{s=1}^{t} \eta_s \left\|\frac{\bar{\boldsymbol{g}}_s}{\sqrt{\boldsymbol{a}_s}}\right\|^2 + (b_1\mathcal{G}_T(t) + b_2)\log\left(\frac{T}{\beta_2^T}\right),$$

where $b_1$ and $b_2$ are positive constants determined by $\beta_1, \beta_2, d, L, \eta$.

**Bounding gradients through induction.** The last challenge comes from the potential unbounded gradients' norm. Plugging the above estimations into (20), we obtain that

$$f(\boldsymbol{y}_{t+1}) \leq f(\boldsymbol{x}_1) + \left(\frac{\mathcal{G}_T(t)}{4\mathcal{G}_T} - \frac{1}{2}\right)\sum_{s=1}^{t} \eta_s \left\|\frac{\bar{\boldsymbol{g}}_s}{\sqrt{\boldsymbol{a}_s}}\right\|^2 + c_1\mathcal{G}_T + (c_2\mathcal{G}_T(t) + c_3)\log\left(\frac{T}{\beta_2^T}\right), \tag{25}$$

where $c_1, c_2, c_3$ are constants determined by $\beta_1, \beta_2, d, L, \eta$. Then, we will first show that $G_1 \leq G$ and suppose that for some $t \in [T]$,

$$G_s \leq G, \quad \forall s \in [t] \quad \text{thus} \quad \mathcal{G}_T(s) \leq \mathcal{G}_T, \quad \forall s \in [t]. \tag{26}$$

It's then clear to reveal from (25) and the induction assumption that $f(\boldsymbol{y}_{t+1})$ is restricted by the first-order of $\mathcal{G}_T$. Moreover, $f(\boldsymbol{y}_{t+1}) - f^*$ could be served as the upper bound of $\|\bar{\boldsymbol{g}}_{t+1}\|^2$ since

$$\|\bar{\boldsymbol{g}}_{t+1}\|^2 \leq 2\|\nabla f(\boldsymbol{y}_{t+1})\|^2 + 2\|\bar{\boldsymbol{g}}_{t+1} - \nabla f(\boldsymbol{y}_{t+1})\|^2 \leq 4L(f(\boldsymbol{y}_{t+1}) - f^*) + 2\|\bar{\boldsymbol{g}}_{t+1} - \nabla f(\boldsymbol{y}_{t+1})\|^2, \tag{27}$$

where we use a standard result $\|\nabla f(\boldsymbol{x})\|^2 \leq 2L(f(\boldsymbol{x}) - f^*)$ in smooth-based optimization. We also use the smoothness to control $\|\bar{\boldsymbol{g}}_{t+1} - \nabla f(\boldsymbol{y}_{t+1})\|^2$ and combine with (26) and (27) to derive that

$$\|\bar{\boldsymbol{g}}_{t+1}\|^2 \leq \tilde{d}_1 + \tilde{d}_2(\sigma_1 G^{p/2} + G),$$

where $\tilde{d}_1, \tilde{d}_2$ are constants that are also determined by hyper-parameters and restricted by $\mathcal{O}(\log T - T\log\beta_2)$ with respect to $T$. Then, using Young's inequality,

$$\|\bar{\boldsymbol{g}}_{t+1}\|^2 \leq \frac{G^2}{2} + \tilde{d}_1 + \frac{4-p}{4} \cdot p^{\frac{p}{4-p}} \left(\sigma_1 \tilde{d}_2\right)^{\frac{4}{4-p}} + \left(\tilde{d}_2\right)^2.$$

Thus, combining with a proper construction $G^2$ (detailed in (53)), we could prove that

$$G^2 = 2\tilde{d}_1 + \frac{4-p}{2} \cdot p^{\frac{p}{4-p}} \left(\sigma_1 \tilde{d}_2\right)^{\frac{4}{4-p}} + 2\left(\tilde{d}_2\right)^2,$$

which leads to $\|\bar{\boldsymbol{g}}_{t+1}\|^2 \leq G^2$. Combining with the induction argument, we deduce that $\|\bar{\boldsymbol{g}}_t\|^2 \leq G^2, \forall t \in [T+1]$.

**Final estimation.** Following the induction step for upper bounding the gradients' norm, we also prove the following result in high probability:

$$L\sum_{s=1}^{T} \frac{\eta_s}{\|\boldsymbol{a}_s\|_\infty} \|\bar{\boldsymbol{g}}_s\|^2 \leq L\sum_{s=1}^{T} \eta_s \left\|\frac{\bar{\boldsymbol{g}}_s}{\sqrt{\boldsymbol{a}_s}}\right\|^2 \leq G^2.$$

We could rely on $\mathcal{G}_T$ to prove that $\|\boldsymbol{a}_s\|_\infty \leq \mathcal{G}_T\sqrt{1-\beta_2^s} + \epsilon_s, \forall s \in [T]$, and then combine with $\eta_s$ in Algorithm 1 to further deduce the desired guarantee for $\sum_{s=1}^{T} \|\bar{\boldsymbol{g}}_s\|^2/T$.

# 7   Conclusion

In this paper, we investigate the convergence of the Adam optimization algorithm on non-convex smooth problems under certain relaxed conditions. We begin by considering a mild noise assumption that encompasses several noise types, particularly the almost surely affine variance noise. Under this noise condition, we demonstrate that Adam can find a stationary point at a rate of $\mathcal{O}(\text{poly}(\log T)/\sqrt{T})$ with high probability. Within our framework, we introduce a novel proxy step-size to manage the entanglement of stochastic gradients and adaptive step-sizes, and we employ a new decomposition method to estimate the errors introduced by the proxy step-size, the momentum, and the corrective terms in Adam.

We also extend our analysis to the convergence of Adam when the objective function is generalized smooth. This relaxed assumption is empirically validated to be more realistic in practical applications. Our results indicate that, with appropriate hyper-parameter tuning, Adam can find a stationary point at the same order of convergence rate as in the smooth case.

**Limitations.** Our study has several limitations that warrant further exploration. First, it would be advantageous to provide experimental results to validate the hyper-parameter settings in our results. Second, the convergence bound is not strictly tight compared to the lower bound, leaving a gap involving logarithmic factors, which may be improved in future work.

## Acknowledgement

This work was supported in part by the NSFC under grant number 12471096, and the National Key Research and Development Program of China under grant number 2021YFA1003500.

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

## A Complementary lemmas

We first provide some necessary technical lemmas as follows.

**Lemma A.1.** *Suppose that $\{\alpha_s\}_{s\geq 1}$ is a non-negative sequence. Given $\beta_2 \in (0, 1]$ and $\varepsilon > 0$, we define $\theta_s = \sum_{j=1}^s \beta_2^{s-j}\alpha_j$. Then, for any $t \geq 1$,*

$$\sum_{s=1}^t \frac{\alpha_j}{\varepsilon + \theta_j} \leq \log\left(1 + \frac{\theta_t}{\varepsilon}\right) - t\log\beta_2.$$

*Proof.* See the proof of Lemma 5.2 in [10]. □

**Lemma A.2.** *Suppose that $\{\alpha_s\}_{s\geq 1}$ is a real number sequence. Given $0 \leq \beta_1 < \beta_2 \leq 1$ and $\varepsilon > 0$, we define $\zeta_s = \sum_{j=1}^s \beta_1^{s-j}\alpha_j$, $\gamma_s = \frac{1}{1-\beta_1^s}\sum_{j=1}^s \beta_1^{s-j}\alpha_j$ and $\theta_s = \sum_{j=1}^s \beta_2^{s-j}\alpha_j^2$, then*

$$\sum_{s=1}^t \frac{\zeta_s^2}{\varepsilon + \theta_s} \leq \frac{1}{(1-\beta_1)(1-\beta_1/\beta_2)}\left(\log\left(1 + \frac{\theta_t}{\varepsilon}\right) - t\log\beta_2\right), \quad \forall t \geq 1,$$

$$\sum_{s=1}^t \frac{\gamma_s^2}{\varepsilon + \theta_s} \leq \frac{1}{(1-\beta_1)^2(1-\beta_1/\beta_2)}\left(\log\left(1 + \frac{\theta_t}{\varepsilon}\right) - t\log\beta_2\right), \quad \forall t \geq 1.$$

*Proof.* The proof for the first inequality can be found in the proof of Lemma A.2 [10]. For the second result, let $\hat{M} = \sum_{j=1}^s \beta_1^{s-j}$. Then using Jensen's inequality, we have

$$\left(\sum_{j=1}^s \beta_1^{s-j}\alpha_j\right)^2 = \left(\hat{M}\sum_{j=1}^s \frac{\beta_1^{s-j}}{\hat{M}}\alpha_j\right)^2 \leq \hat{M}^2\sum_{j=1}^s \frac{\beta_1^{s-j}}{\hat{M}}\alpha_j^2 = \hat{M}\sum_{j=1}^s \beta_1^{s-j}\alpha_j^2. \tag{28}$$

Hence, we further have

$$\frac{\gamma_s^2}{\varepsilon + \theta_s} \leq \frac{\hat{M}}{(1-\beta_1^s)^2}\sum_{j=1}^s \beta_1^{s-j}\frac{\alpha_j^2}{\varepsilon + \theta_s} = \frac{1}{(1-\beta_1)(1-\beta_1^s)}\sum_{j=1}^s \beta_1^{s-j}\frac{\alpha_j^2}{\varepsilon + \theta_s}.$$

Recalling the definition of $\theta_s$, we have $\varepsilon + \theta_s \geq \varepsilon + \beta_2^{s-j}\theta_j \geq \beta_2^{s-j}(\varepsilon + \theta_j)$. Hence, combining with $1 - \beta_1 \leq 1 - \beta_1^s$,

$$\frac{\gamma_s^2}{\varepsilon + \theta_s} \leq \frac{1}{(1-\beta_1)(1-\beta_1^s)}\sum_{j=1}^s \left(\frac{\beta_1}{\beta_2}\right)^{s-j}\frac{\alpha_j^2}{\varepsilon + \theta_j} \leq \frac{1}{(1-\beta_1)^2}\sum_{j=1}^s \left(\frac{\beta_1}{\beta_2}\right)^{s-j}\frac{\alpha_j^2}{\varepsilon + \theta_j}.$$

Summing up both sides over $s \in [t]$, and noting that $\beta_1 < \beta_2$,

$$\sum_{s=1}^t \frac{\gamma_s^2}{\varepsilon + \theta_s} \leq \frac{1}{(1-\beta_1)^2}\sum_{s=1}^t\sum_{j=1}^s \left(\frac{\beta_1}{\beta_2}\right)^{s-j}\frac{\alpha_j^2}{\varepsilon + \theta_j} \leq \frac{1}{(1-\beta_1)^2(1-\beta_1/\beta_2)}\sum_{j=1}^t \frac{\alpha_j^2}{\varepsilon + \theta_j}.$$

Finally applying Lemma A.1, we obtain the desired result. □

Then, we introduce a standard concentration inequality for the martingale difference sequence that is useful for achieving the high probability bounds, see [26] for a proof.

**Lemma A.3.** *Suppose $\{Z_s\}_{s \in [T]}$ is a martingale difference sequence with respect to $\zeta_1, \cdots, \zeta_T$. Assume that for each $s \in [T]$, $\sigma_s$ is a random variable only dependent by $\zeta_1, \cdots, \zeta_{s-1}$ and satisfies that*

$$\mathbb{E}\left[\exp(Z_s^2/\sigma_s^2) \mid \zeta_1, \cdots, \zeta_{s-1}\right] \leq e,$$

*then for any $\lambda > 0$, and for any $\delta \in (0, 1)$, it holds that*

$$\mathbb{P}\left(\sum_{s=1}^{T} Z_s > \frac{1}{\lambda}\log\left(\frac{1}{\delta}\right) + \frac{3}{4}\lambda \sum_{s=1}^{T} \sigma_s^2\right) \leq \delta.$$

## B  Proof of Theorem 3.1

The detailed proof of Theorem 3.1 corresponds to the proof sketch in Section 6.

### B.1  Preliminary

To start with, we introduce the following two notations,

$$\hat{m}_s = \frac{m_s}{1 - \beta_1^s}, \quad \hat{v}_s = \frac{v_s}{1 - \beta_2^s}, \tag{29}$$

which include two corrective terms for $m_s$ and $v_s$. It is easy to see that $\eta_s$ satisfies

$$\eta_s = \frac{\eta\sqrt{1 - \beta_2^s}}{1 - \beta_1^s} \leq \frac{\eta}{1 - \beta_1^s} \leq \frac{\eta}{1 - \beta_1}. \tag{30}$$

We follow all the notations in Section 6, which we present here for the convenience of reading,

$$
\begin{aligned}
&\mathcal{M}_T = \sqrt{\log\left(\frac{eT}{\delta}\right)}, \quad G_s = \max_{j \in [s]}\|\bar{g}_j\|, \\
&\mathcal{G}_T(s) = \mathcal{M}_T\sqrt{2\sigma_0^2 + 2\sigma_1^2 G_s^p + 2G_s^2}, \quad \mathcal{G}_T = \mathcal{M}_T\sqrt{2\sigma_0^2 + 2\sigma_1^2 G^p + 2G^2}, \\
&b_s = \sqrt{\beta_2 v_{s-1} + (1 - \beta_2)g_s^2 + \epsilon_s}, \\
&a_s = \sqrt{\beta_2 v_{s-1} + (1 - \beta_2)\left(\mathcal{G}_T(s)\mathbf{1}_d\right)^2 + \epsilon_s}.
\end{aligned}
\tag{31}
$$

The following lemmas provide some estimations for the algorithm-dependent terms, which play vital roles in the proof of Theorem 3.1. The detailed proofs could be found in Appendix F.1.

**Lemma B.1.** *Let $\eta_s, b_s$ be given in Algorithm 1 and (16), then*

$$\left\|\frac{\eta_s b_{s-1}}{\eta_{s-1}b_s} - \mathbf{1}_d\right\|_\infty \leq \Sigma_{\max} := \frac{1}{\sqrt{\beta_2}}, \quad \forall s \geq 2.$$

The following lemma could be found similarly in [57, Lemma A.2].

**Lemma B.2.** *Let $m_s, b_s$ be given in Algorithm 1 and (16) with $0 \leq \beta_1 < \beta_2 < 1$, respectively. Then,*

$$\left\|\frac{m_s}{b_s}\right\|_\infty \leq \sqrt{\frac{(1 - \beta_1)(1 - \beta_1^s)}{(1 - \beta_2)(1 - \beta_1/\beta_2)}}, \quad \forall s \geq 1.$$

*Consequently, if $f$ is $L$-smooth and we set $\eta = C_0\sqrt{1 - \beta_2}$ for some constant $C_0 > 0$, then*

$$\|\bar{g}_s\| \leq \|\bar{g}_1\| + LC_0 s\sqrt{\frac{d}{1 - \beta_1/\beta_2}}, \quad \forall s \geq 1.$$

The following lemma is necessary for deriving (24) in the proof sketch.

**Lemma B.3.** *Let $\boldsymbol{g}_s, \boldsymbol{m}_s$ be given in Algorithm 1 and $\hat{\boldsymbol{m}}_s, \boldsymbol{b}_s$ be defined in (29) and (16). If $0 \leq \beta_1 < \beta_2 < 1$ and $\mathcal{F}_i(t) = 1 + \frac{1}{\epsilon^2} \sum_{s=1}^{t} g_{s,i}^2$, then for any $t \geq 1$,*

$$\sum_{s=1}^{t} \left\| \frac{\boldsymbol{g}_s}{\boldsymbol{b}_s} \right\|^2 \leq \frac{1}{1-\beta_2} \sum_{i=1}^{d} \log \left( \frac{\mathcal{F}_i(t)}{\beta_2^t} \right),$$

$$\sum_{s=1}^{t} \left\| \frac{\boldsymbol{m}_s}{\boldsymbol{b}_s} \right\|^2 \leq \frac{1-\beta_1}{(1-\beta_2)(1-\beta_1/\beta_2)} \sum_{i=1}^{d} \log \left( \frac{\mathcal{F}_i(t)}{\beta_2^t} \right),$$

$$\sum_{s=1}^{t} \left\| \frac{\boldsymbol{m}_s}{\boldsymbol{b}_{s+1}} \right\|^2 \leq \frac{1-\beta_1}{\beta_2(1-\beta_2)(1-\beta_1/\beta_2)} \sum_{i=1}^{d} \log \left( \frac{\mathcal{F}_i(t)}{\beta_2^t} \right),$$

$$\sum_{s=1}^{t} \left\| \frac{\hat{\boldsymbol{m}}_s}{\boldsymbol{b}_s} \right\| \leq \frac{1}{(1-\beta_2)(1-\beta_1/\beta_2)} \sum_{i=1}^{d} \log \left( \frac{\mathcal{F}_i(t)}{\beta_2^t} \right).$$

The following lemmas are based on the smooth condition.

**Lemma B.4.** *Suppose that $f$ is $L$-smooth and Assumption (A1) holds, then for any $\boldsymbol{x} \in \mathbb{R}^d$,*

$$\|\nabla f(\boldsymbol{x})\|^2 \leq 2L(f(\boldsymbol{x}) - f^*).$$

**Lemma B.5.** *Let $\boldsymbol{x}_s$ be given in Algorithm 1 and $\boldsymbol{y}_s$ be defined in (17). If $f$ is $L$-smooth, $\eta = C_0\sqrt{1-\beta_2}$ and $0 \leq \beta_1 < \beta_2 < 1$, then*

$$\|\nabla f(\boldsymbol{x}_s)\| \leq \|\nabla f(\boldsymbol{y}_s)\| + M, \quad M := \frac{LC_0\sqrt{d}}{(1-\beta_1)\sqrt{1-\beta_1/\beta_2}}, \quad \forall s \geq 1.$$

## B.2 Start point and decomposition

Specifically, we fix the horizon $T$ and start from the descent lemma of $L$-smoothness,

$$f(\boldsymbol{y}_{s+1}) \leq f(\boldsymbol{y}_s) + \langle \nabla f(\boldsymbol{y}_s), \boldsymbol{y}_{s+1} - \boldsymbol{y}_s \rangle + \frac{L}{2} \|\boldsymbol{y}_{s+1} - \boldsymbol{y}_s\|^2, \quad \forall s \in [T]. \tag{32}$$

For any given $t \in [T]$, combining with (18) and (32) and then summing over $s \in [t]$, we obtain the same inequality in (20),

$$f(\boldsymbol{y}_{t+1}) \leq f(\boldsymbol{x}_1) + \underbrace{\sum_{s=1}^{t} -\eta_s \left\langle \nabla f(\boldsymbol{y}_s), \frac{\boldsymbol{g}_s}{\boldsymbol{b}_s} \right\rangle}_{\mathbf{A}} + \underbrace{\frac{\beta_1}{1-\beta_1} \sum_{s=1}^{t} \langle \Delta_s \odot (\boldsymbol{x}_s - \boldsymbol{x}_{s-1}), \nabla f(\boldsymbol{y}_s) \rangle}_{\mathbf{B}}$$

$$+ \underbrace{\frac{L}{2} \sum_{s=1}^{t} \left\| \eta_s \cdot \frac{\boldsymbol{g}_s}{\boldsymbol{b}_s} - \frac{\beta_1}{1-\beta_1} (\Delta_s \odot (\boldsymbol{x}_s - \boldsymbol{x}_{s-1})) \right\|^2}_{\mathbf{C}}, \tag{33}$$

where we use $\Delta_s$ in (20) and $\boldsymbol{y}_1 = \boldsymbol{x}_1$. We then further make a decomposition by introducing $\bar{\boldsymbol{g}}_s$ into **A** and **B**

$$\mathbf{A} = \underbrace{\sum_{s=1}^{t} -\eta_s \left\langle \bar{\boldsymbol{g}}_s, \frac{\boldsymbol{g}_s}{\boldsymbol{b}_s} \right\rangle}_{\mathbf{A.1}} + \underbrace{\sum_{s=1}^{t} \eta_s \left\langle \bar{\boldsymbol{g}}_s - \nabla f(\boldsymbol{y}_s), \frac{\boldsymbol{g}_s}{\boldsymbol{b}_s} \right\rangle}_{\mathbf{A.2}}, \tag{34}$$

and

$$\mathbf{B} = \underbrace{\frac{\beta_1}{1-\beta_1} \sum_{s=1}^{t} \langle \Delta_s \odot (\boldsymbol{x}_s - \boldsymbol{x}_{s-1}), \bar{\boldsymbol{g}}_s \rangle}_{\mathbf{B.1}} + \underbrace{\frac{\beta_1}{1-\beta_1} \sum_{s=1}^{t} \langle \Delta_s \odot (\boldsymbol{x}_s - \boldsymbol{x}_{s-1}), \nabla f(\boldsymbol{y}_s) - \bar{\boldsymbol{g}}_s \rangle}_{\mathbf{B.2}}.$$

$$\tag{35}$$

## B.3 Probabilistic estimations

We will provide two probabilistic inequalities with the detailed proofs given in Appendix F.2. The first one establishes an upper bound for the noise norm, which we have already informally presented in (21).

**Lemma B.6.** *Given $T \geq 1$, suppose that for any $s \in [T]$, $\boldsymbol{\xi}_s = \boldsymbol{g}_s - \bar{\boldsymbol{g}}_s$ satisfies Assumption (A3). Then for any given $\delta \in (0, 1)$, it holds that with probability at least $1 - \delta$,*

$$\|\boldsymbol{\xi}_s\|^2 \leq \mathcal{M}_T^2 \left( \sigma_0^2 + \sigma_1^2 \|\bar{\boldsymbol{g}}_s\|^p \right), \quad \forall s \in [T]. \tag{36}$$

We next provide a probabilistic upper bound as shown in (22) for a summation of the inner product, where we rely on the property of the martingale difference sequence and the proxy step-size $\boldsymbol{a}_s$ in (23).

**Lemma B.7.** *Given $T \geq 1$ and $\delta \in (0, 1)$. If Assumptions (A2) and (A3) hold, then for any $\lambda > 0$, with probability at least $1 - \delta$,*

$$-\sum_{s=1}^{t} \eta_s \left\langle \bar{\boldsymbol{g}}_s, \frac{\boldsymbol{\xi}_s}{\boldsymbol{a}_s} \right\rangle \leq \frac{3\lambda\eta\mathcal{G}_T(t)}{4(1-\beta_1)\sqrt{1-\beta_2}} \sum_{s=1}^{t} \eta_s \left\| \frac{\bar{\boldsymbol{g}}_s}{\sqrt{\boldsymbol{a}_s}} \right\|^2 + \frac{1}{\lambda} \log\left(\frac{T}{\delta}\right), \quad \forall t \in [T]. \tag{37}$$

*As a consequence, when setting $\lambda = (1-\beta_1)\sqrt{1-\beta_2}/(3\eta\mathcal{G}_T)$, it holds that with probability at least $1 - \delta$,*

$$-\sum_{s=1}^{t} \eta_s \left\langle \bar{\boldsymbol{g}}_s, \frac{\boldsymbol{\xi}_s}{\boldsymbol{a}_s} \right\rangle \leq \frac{\mathcal{G}_T(t)}{4\mathcal{G}_T} \sum_{s=1}^{t} \eta_s \left\| \frac{\bar{\boldsymbol{g}}_s}{\sqrt{\boldsymbol{a}_s}} \right\|^2 + D_1\mathcal{G}_T, \quad \forall t \in [T], \tag{38}$$

*where $D_1 = \frac{3\eta}{(1-\beta_1)\sqrt{1-\beta_2}} \log\left(\frac{T}{\delta}\right)$.*

## B.4 Deterministic estimations

In this section, we shall assume that (36) or/and (38) hold whenever the related estimation is needed. Then we obtain the following key lemmas with the detailed proofs given in Appendix F.3.

**Lemma B.8.** *Given $T \geq 1$. If (36) holds, then we have*

$$\max_{j \in [s]} \|\boldsymbol{\xi}_j\| \leq \mathcal{G}_T(s), \quad \max_{j \in [s]} \|\boldsymbol{g}_j\| \leq \mathcal{G}_T(s), \quad \max_{j \in [s]} \|\boldsymbol{v}_j\|_\infty \leq \left(\mathcal{G}_T(s)\right)^2, \quad \forall s \in [T].$$

**Lemma B.9.** *Given $T \geq 1$. If $\boldsymbol{b}_s = (b_{s,i})_i$ and $\boldsymbol{a}_s = (a_{s,i})_i$ follow the definitions in (16) and (23) respectively, and (36) holds, then for all $s \in [T], i \in [d]$,*

$$\left| \frac{1}{a_{s,i}} - \frac{1}{b_{s,i}} \right| \leq \frac{\mathcal{G}_T(s)\sqrt{1-\beta_2}}{a_{s,i}b_{s,i}} \quad \text{and} \quad \left| \frac{1}{a_{s,i}} - \frac{1}{b_{s-1,i}} \right| \leq \frac{(\mathcal{G}_T(s) + \epsilon)\sqrt{1-\beta_2}}{a_{s,i}b_{s-1,i}}.$$

**Lemma B.10.** *Given $T \geq 1$. Under the conditions in Lemma B.3 and Lemma B.5, if (36) holds, then the following inequality holds,*

$$\mathcal{F}_i(t) \leq \mathcal{F}(T), \quad \forall t \in [T], i \in [d],$$

*where $\hat{M} = M(1 - \beta_1)$ and $M$ follows the definition in Lemma B.5, $\mathcal{F}(T)$ is define by*

$$\mathcal{F}(T) := 1 + \frac{2\mathcal{M}_T^2}{\epsilon^2} \left[ \sigma_0^2 T + \sigma_1^2 T \left( \|\bar{\boldsymbol{g}}_1\| + T\hat{M} \right)^p + T \left( \|\bar{\boldsymbol{g}}_1\| + T\hat{M} \right)^2 \right]. \tag{39}$$

We move to estimate all the related terms in Appendix B.2. First, the estimation for **A.1** relies on both the two probabilistic estimations in Appendix B.3.

**Lemma B.11.** *Given $T \geq 1$, suppose that (36) and (38) hold. Then for all $t \in [T]$,*

$$\mathbf{A.1} \leq \left( \frac{\mathcal{G}_T(t)}{4\mathcal{G}} - \frac{3}{4} \right) \sum_{s=1}^{t} \eta_s \left\| \frac{\bar{\boldsymbol{g}}_s}{\sqrt{\boldsymbol{a}_s}} \right\|^2 + D_1\mathcal{G}_T + D_2\mathcal{G}_T(t) \sum_{s=1}^{t} \left\| \frac{\boldsymbol{g}_s}{\boldsymbol{b}_s} \right\|^2, \tag{40}$$

*where $D_1$ is given as in Lemma B.7 and $D_2 = \frac{\eta\sqrt{1-\beta_2}}{1-\beta_1}$.*

We also obtain the following lemma to estimate the adaptive momentum part **B.1**.

**Lemma B.12.** *Given $T \geq 1$, if (36) holds, then for all $t \in [T]$,*

$$\mathbf{B.1} \leq \frac{1}{4} \sum_{s=1}^{t} \eta_s \left\| \frac{\bar{g}_s}{\sqrt{a_s}} \right\|^2 + (D_3 \mathcal{G}_T(t) + D_4) \sum_{s=1}^{t} \left( \left\| \frac{m_{s-1}}{b_s} \right\|^2 + \left\| \frac{m_{s-1}}{b_{s-1}} \right\|^2 \right) + D_5 G_t, \quad (41)$$

*where*

$$D_3 = \frac{2\eta\sqrt{1-\beta_2}}{(1-\beta_1)^3}, \quad D_4 = \epsilon D_3, \quad D_5 = \frac{2\eta\sqrt{d}}{\sqrt{(1-\beta_1)^3(1-\beta_2)(1-\beta_1/\beta_2)}}. \quad (42)$$

**Proposition B.13.** *Given $T \geq 1$. If $f$ is L-smooth, then the following inequality holds,*

$$f(y_{t+1}) \leq f(x_1) + \mathbf{A.1} + \mathbf{B.1} + D_6 \sum_{s=1}^{t-1} \left\| \frac{\hat{m}_s}{b_s} \right\|^2 + D_7 \sum_{s=1}^{t} \left\| \frac{g_s}{b_s} \right\|^2, \quad \forall t \in [T],$$

*where $\Sigma_{\max}$ is as in Lemma B.1 and*

$$D_6 = \frac{L\eta^2(1 + 4\Sigma_{\max}^2)}{2(1-\beta_1)^2}, \quad D_7 = \frac{3L\eta^2}{2(1-\beta_1)^2}. \quad (43)$$

*Proof.* Recalling the decomposition in Appendix B.2. We first estimate **A.2**. Using the smoothness of $f$ and (17), we have

$$\|\nabla f(y_s) - \bar{g}_s\| \leq L\|y_s - x_s\| = \frac{L\beta_1}{1-\beta_1}\|x_s - x_{s-1}\|. \quad (44)$$

Hence, applying Young's inequality, (44) and (30),

$$\eta_s \left\langle \bar{g}_s - \nabla f(y_s), \frac{g_s}{b_s} \right\rangle \leq \eta_s \|\bar{g}_s - \nabla f(y_s)\| \cdot \left\| \frac{g_s}{b_s} \right\|$$

$$\leq \frac{1}{2L}\|\bar{g}_s - \nabla f(y_s)\|^2 + \frac{L\eta_s^2}{2} \left\| \frac{g_s}{b_s} \right\|^2 \leq \frac{L\beta_1^2}{2(1-\beta_1)^2}\|x_s - x_{s-1}\|^2 + \frac{L\eta^2}{2(1-\beta_1)^2} \left\| \frac{g_s}{b_s} \right\|^2. \quad (45)$$

Recalling the updated rule in Algorithm 1 and applying (29) as well as (30),

$$\|x_s - x_{s-1}\|^2 = \eta_{s-1}^2 \left\| \frac{m_{s-1}}{b_{s-1}} \right\|^2 \leq \eta^2 \left\| \frac{\hat{m}_{s-1}}{b_{s-1}} \right\|^2. \quad (46)$$

Therefore, applying (45), (46) and $\beta_1 \in [0,1)$, and then summing over $s \in [t]$

$$\mathbf{A.2} \leq \frac{L\eta^2}{2(1-\beta_1)^2} \sum_{s=1}^{t} \left\| \frac{\hat{m}_{s-1}}{b_{s-1}} \right\|^2 + \frac{L\eta^2}{2(1-\beta_1)^2} \sum_{s=1}^{t} \left\| \frac{g_s}{b_s} \right\|^2. \quad (47)$$

Applying Cauchy-Schwarz inequality, Lemma B.1, and combining with (44), (46), $\Sigma_{\max} \geq 1$, and $\beta_1 \in [0,1)$

$$\mathbf{B.2} \leq \frac{\beta_1}{1-\beta_1} \sum_{s=1}^{t} \|\Delta_s\|_\infty \|x_s - x_{s-1}\| \|\nabla f(y_s) - \bar{g}_s\|$$

$$\leq \frac{L\beta_1^2 \Sigma_{\max}}{(1-\beta_1)^2} \sum_{s=1}^{t} \|x_s - x_{s-1}\|^2 \leq \frac{L\Sigma_{\max}^2 \eta^2}{(1-\beta_1)^2} \sum_{s=1}^{t} \left\| \frac{\hat{m}_{s-1}}{b_{s-1}} \right\|^2, \quad (48)$$

Finally, applying the basic inequality, Lemma B.1 and (46),

$$\mathbf{C} \leq L \sum_{s=1}^{t} \eta_s^2 \left\| \frac{g_s}{b_s} \right\|^2 + \frac{L\beta_1^2}{(1-\beta_1)^2} \sum_{s=1}^{t} \|\Delta_s\|_\infty^2 \|x_s - x_{s-1}\|^2$$

$$\leq \frac{L\eta^2}{(1-\beta_1)^2} \sum_{s=1}^{t} \left\| \frac{g_s}{b_s} \right\|^2 + \frac{L\eta^2 \Sigma_{\max}^2}{(1-\beta_1)^2} \sum_{s=1}^{t} \left\| \frac{\hat{m}_{s-1}}{b_{s-1}} \right\|^2. \quad (49)$$

Recalling the decomposition in (34) and (35), then plugging (47), (48) and (49) into (33), we obtain the desired result. $\qquad \square$

## B.5 Bounding gradients

Based on all the results in Appendix B.3 and Appendix B.4, we are now ready to provide a global upper bound for gradients' norm along the optimization trajectory.

**Proposition B.14.** *Under the same conditions in Theorem 3.1, for any given $\delta \in (0, 1/2)$, it holds that with probability at least $1 - 2\delta$,*

$$\|\bar{\boldsymbol{g}}_t\|^2 \le G_t^2 \le G^2, \quad \|\boldsymbol{g}_t\|^2 \le (\mathcal{G}_T(t))^2 \le \mathcal{G}_T^2, \quad \forall t \in [T+1], \tag{50}$$

*and*

$$\|\bar{\boldsymbol{g}}_{t+1}\|^2 \le G^2 - L \sum_{s=1}^{t} \eta_s \left\| \frac{\bar{\boldsymbol{g}}_s}{\sqrt{\boldsymbol{a}_s}} \right\|^2, \quad \forall t \in [T], \tag{51}$$

*where $G^2$ is as in Theorem 3.1 and $G_t, G, \mathcal{G}_T$ are given by* (19).

*Proof.* Applying Lemma B.6 and Lemma B.7, we know that (36) or (38) hold with probability at least $1 - \delta$. With these two inequalities, we could deduce the desired inequalities (50) and (51). Therefore, (50) and (51) hold with probability at least $1 - 2\delta$. We first plug (40) and (41) into the result in Proposition B.13, which leads to that for all $t \in [T]$,

$$f(\boldsymbol{y}_{t+1}) \le f(\boldsymbol{x}_1) + \left( \frac{\mathcal{G}_T(t)}{4\mathcal{G}_T} - \frac{1}{2} \right) \sum_{s=1}^{t} \eta_s \left\| \frac{\bar{\boldsymbol{g}}_s}{\sqrt{\boldsymbol{a}_s}} \right\|^2 + D_1 \mathcal{G}_T + (D_2 \mathcal{G}_T(t) + D_7) \sum_{s=1}^{t} \left\| \frac{\boldsymbol{g}_s}{\boldsymbol{b}_s} \right\|^2$$

$$+ (D_3 \mathcal{G}_T(t) + D_4) \sum_{s=1}^{t} \left( \left\| \frac{\boldsymbol{m}_{s-1}}{\boldsymbol{b}_s} \right\|^2 + \left\| \frac{\boldsymbol{m}_{s-1}}{\boldsymbol{b}_{s-1}} \right\|^2 \right) + D_5 G_t + D_6 \sum_{s=1}^{t-1} \left\| \frac{\hat{\boldsymbol{m}}_s}{\boldsymbol{b}_s} \right\|^2. \tag{52}$$

Next, we will introduce the induction argument based on (52). We first provide the specific definition of $G^2$ as follows which is a constant determined by the horizon $T$ and other hyper-parameters but not relying on $t$,[4]

$$G^2 := 8L(f(\boldsymbol{x}_1) - f^*) + \frac{48\mathcal{M}_T LC_0\sigma_0}{1 - \beta_1} \log\left(\frac{T}{\delta}\right) + \frac{16\mathcal{M}_T LC_0\sigma_0 d}{1 - \beta_1} \log\left(\frac{\mathcal{F}(T)}{\beta_2^T}\right)$$

$$+ 8 \left( \frac{3LC_0 + 8(\mathcal{M}_T\sigma_0 + \epsilon_0)}{\beta_2} \right) \frac{LC_0 d}{(1 - \beta_1)^2(1 - \beta_1/\beta_2)} \log\left(\frac{\mathcal{F}(T)}{\beta_2^T}\right)$$

$$+ \frac{4 - p}{2} \cdot p^{\frac{p}{4-p}} \left[ \frac{72\mathcal{M}_T L\sigma_1 C_0 d}{\beta_2(1 - \beta_1)^2(1 - \beta_1/\beta_2)} \log\left(\frac{T + \mathcal{F}(T)}{\delta\beta_2^T}\right) \right]^{\frac{4}{4-p}}$$

$$+ 32 \left[ \frac{18\mathcal{M}_T LC_0 d}{\beta_2(1 - \beta_1)^2(1 - \beta_1/\beta_2)} \log\left(\frac{T + \mathcal{F}(T)}{\delta\beta_2^T}\right) \right]^2 + \frac{4L^2 C_0^2 d}{(1 - \beta_1)^2(1 - \beta_1/\beta_2)}. \tag{53}$$

The induction then begins by noting that $G_1^2 = \|\bar{\boldsymbol{g}}_1\|^2 \le 2L(f(\boldsymbol{x}_1) - f^*) \le G^2$ from Lemma B.4 and (53). Then we assume that for some $t \in [T]$,

$$G_s \le G, \quad \forall s \in [t] \quad \text{consequently} \quad \mathcal{G}_T(s) \le \mathcal{G}_T, \quad \forall s \in [t]. \tag{54}$$

Using this induction assumption over (52) and subtracting with $f^*$ on both sides,

$$f(\boldsymbol{y}_{t+1}) - f^* \le f(\boldsymbol{x}_1) - f^* - \frac{1}{4} \sum_{s=1}^{t} \eta_s \left\| \frac{\bar{\boldsymbol{g}}_s}{\sqrt{\boldsymbol{a}_s}} \right\|^2 + D_1 \mathcal{G}_T + (D_2 \mathcal{G}_T + D_7) \sum_{s=1}^{t} \left\| \frac{\boldsymbol{g}_s}{\boldsymbol{b}_s} \right\|^2$$

$$+ (D_3 \mathcal{G}_T + D_4) \sum_{s=1}^{t} \left( \left\| \frac{\boldsymbol{m}_{s-1}}{\boldsymbol{b}_s} \right\|^2 + \left\| \frac{\boldsymbol{m}_{s-1}}{\boldsymbol{b}_{s-1}} \right\|^2 \right) + D_5 G + D_6 \sum_{s=1}^{t-1} \left\| \frac{\hat{\boldsymbol{m}}_s}{\boldsymbol{b}_s} \right\|^2. \tag{55}$$

Further, we combine with Lemma B.3 and Lemma B.10 to estimate the four summations defined in Lemma B.3, and then use $G \le \mathcal{G}_T \le 2\mathcal{M}_T \left( \sigma_0 + \sigma_1 G^{p/2} + G \right)$ to control the RHS of (55),

$$f(\boldsymbol{y}_{t+1}) - f^* \le -\frac{1}{4} \sum_{s=1}^{t} \eta_s \left\| \frac{\bar{\boldsymbol{g}}_s}{\sqrt{\boldsymbol{a}_s}} \right\|^2 + \tilde{D}_1 + \tilde{D}_2 + \tilde{D}_3 \mathcal{H}(G), \tag{56}$$

---

[4]We further deduce (7) in Theorem 3.1 based on (53).

where $\mathcal{H}(G) = \sigma_1 G^{p/2} + G$ and $\tilde{D}_1, \tilde{D}_2, \tilde{D}_3$ are defined as

$$\tilde{D}_1 = f(\boldsymbol{x}_1) - f^* + 2\mathcal{M}_T \sigma_0 D_1,$$

$$\tilde{D}_2 = \left[ \frac{2\mathcal{M}_T \sigma_0 D_2 + D_7}{1 - \beta_2} + \frac{4\left(\mathcal{M}_T \sigma_0 D_3 + D_4\right)(1 - \beta_1)}{\beta_2(1 - \beta_2)(1 - \beta_1/\beta_2)} + \frac{D_6}{(1 - \beta_2)(1 - \beta_1/\beta_2)} \right] d \log\left( \frac{\mathcal{F}(T)}{\beta_2^T} \right),$$

$$\tilde{D}_3 = 2\mathcal{M}_T \left[ D_1 + \left( \frac{D_2 d}{1 - \beta_2} + \frac{2D_3(1 - \beta_1)d}{\beta_2(1 - \beta_2)(1 - \beta_1/\beta_2)} \right) \log\left( \frac{\mathcal{F}(T)}{\beta_2^T} \right) \right] + D_5.$$

Applying Lemma B.5 and Lemma B.4,

$$\|\bar{\boldsymbol{g}}_{t+1}\|^2 \leq 2\|\nabla f(\boldsymbol{y}_{t+1})\|^2 + 2M^2 \leq 4L(f(\boldsymbol{y}_{t+1}) - f^*) + 2M^2. \tag{57}$$

Then combining (56) with (57),

$$\|\bar{\boldsymbol{g}}_{t+1}\|^2 \leq -L \sum_{s=1}^{t} \eta_s \left\| \frac{\bar{\boldsymbol{g}}_s}{\sqrt{\boldsymbol{a}_s}} \right\|^2 + 4L(\tilde{D}_1 + \tilde{D}_2) + 4L\tilde{D}_3 \mathcal{H}(G) + 2M^2.$$

Applying two Young's inequalities where $ab \leq \frac{a^2}{2} + \frac{b^2}{2}$ and $ab^{\frac{p}{2}} \leq \frac{4-p}{4} \cdot a^{\frac{4}{4-p}} + \frac{p}{4} \cdot b^2, \forall a, b \geq 0$,

$$\|\bar{\boldsymbol{g}}_{t+1}\|^2 \leq \frac{G^2}{4} + \frac{G^2}{4} - L \sum_{s=1}^{t} \eta_s \left\| \frac{\bar{\boldsymbol{g}}_s}{\sqrt{\boldsymbol{a}_s}} \right\|^2 + 4L(\tilde{D}_1 + \tilde{D}_2)$$

$$+ 16L^2 \tilde{D}_3^2 + \frac{4-p}{4} \cdot p^{\frac{p}{4-p}} \left( 4L\sigma_1 \tilde{D}_3 \right)^{\frac{4}{4-p}} + 2M^2. \tag{58}$$

Recalling the definitions of $D_i, i \in [7]$ in (38), (40), (42), and (43). With a simple calculation relying on $\eta = C_0\sqrt{1 - \beta_2}, \epsilon \leq \epsilon_0, \Sigma_{\max} \leq 1/\sqrt{\beta_2}$ and $0 \leq \beta_1 < \beta_2 < 1$, we could deduce that $G^2$ given in (53) satisfies

$$G^2 = 8L(\tilde{D}_1 + \tilde{D}_2) + 32L^2 \tilde{D}_3^2 + \frac{4-p}{2} \cdot p^{\frac{p}{4-p}} \left( 4L\sigma_1 \tilde{D}_3 \right)^{\frac{4}{4-p}} + 4M^2. \tag{59}$$

Based on (58) and (59), we then deduce that $\|\bar{\boldsymbol{g}}_{t+1}\|^2 \leq G^2$. Further combining with $G_{t+1}$ in (19) and the induction assumption in (54),

$$G_{t+1} \leq \max\{\|\bar{\boldsymbol{g}}_{t+1}\|, G_t\} \leq G.$$

Hence, the induction is complete and we obtain the desired result in (50). Furthermore, as a consequence of (58), we also prove that (51) holds. $\qquad \square$

### B.6 Proof of the main result

Now we are ready to prove the main convergence result.

*Proof of Theorem 3.1.* We set $t = T$ in (51) to obtain that with probability at least $1 - 2\delta$,

$$L \sum_{s=1}^{T} \frac{\eta_s}{\|\boldsymbol{a}_s\|_\infty} \|\bar{\boldsymbol{g}}_s\|^2 \leq L \sum_{s=1}^{T} \eta_s \left\| \frac{\bar{\boldsymbol{g}}_s}{\sqrt{\boldsymbol{a}_s}} \right\|^2 \leq G^2 - \|\bar{\boldsymbol{g}}_{T+1}\|^2 \leq G^2. \tag{60}$$

Then, in what follows, we will assume that both (50) and (60) hold. Based on these two inequalities, we could derive the final convergence bound. Since (50) and (60) hold with probability at least $1 - 2\delta$, the final convergence bound also holds with probability at least $1 - 2\delta$. Applying $\boldsymbol{a}_s$ in (23) and (50), we have

$$\|\boldsymbol{a}_s\|_\infty = \max_{i \in [d]} \sqrt{\beta_2 v_{s-1,i} + (1 - \beta_2)(\mathcal{G}_T(s))^2} + \epsilon_s$$

$$\leq \max_{i \in [d]} \sqrt{(1 - \beta_2) \left[ \sum_{j=1}^{s-1} \beta_2^{s-j} g_{j,i}^2 + (\mathcal{G}_T(s))^2 \right]} + \epsilon_s$$

$$\leq \sqrt{(1 - \beta_2) \sum_{j=1}^{s} \beta_2^{s-j} \mathcal{G}_T^2} + \epsilon_s = \mathcal{G}_T \sqrt{1 - \beta_2^s} + \epsilon_s, \quad \forall s \in [T]. \tag{61}$$

Then combining with the setting $\eta_s$ and $\epsilon_s$ in (6), we have for any $s \in [T]$,

$$\frac{\eta_s}{\|\boldsymbol{a}_s\|_\infty} \geq \frac{C_0\sqrt{(1-\beta_2^s)(1-\beta_2)}}{\mathcal{G}_T\sqrt{1-\beta_2^s} + \epsilon_0\sqrt{(1-\beta_2^s)(1-\beta_2)}} \cdot \frac{1}{1-\beta_1^s} \geq \frac{C_0\sqrt{1-\beta_2}}{\mathcal{G}_T + \epsilon_0\sqrt{1-\beta_2}}.$$

We therefore combine with (60) to obtain that with probability at least $1 - 2\delta$,

$$\frac{1}{T}\sum_{s=1}^T \|\bar{\boldsymbol{g}}_s\|^2 \leq \frac{G^2}{TLC_0}\left(\frac{\sqrt{2\sigma_0^2 + 2\sigma_1^2 G^p + 2G^2}}{\sqrt{1-\beta_2}} + \epsilon_0\right)\sqrt{\log\left(\frac{\mathrm{e}T}{\delta}\right)}. \tag{62}$$

Since $\beta_2 \in (0,1)$, we have

$$-\log\beta_2 = \log\left(\frac{1}{\beta_2}\right) \leq \frac{1-\beta_2}{\beta_2} = \frac{c}{T\beta_2},$$

where we apply $\log(1/a) \leq (1-a)/a, \forall a \in (0,1)$. With both sides multiplying $T$, we obtain that $\log\left(1/\beta_2^T\right) \leq c/\beta_2$. Then, we further have that when $\beta_2 = 1 - c/T$,

$$\log\left(\frac{T}{\beta_2^T}\right) \leq \log T + \frac{c}{\beta_2}. \tag{63}$$

Since $0 \leq \beta_1 < \beta_2 < 1$, there exists some constants $\varepsilon_1, \varepsilon_2 > 0$ such that

$$\frac{1}{\beta_2} \leq \frac{1}{\varepsilon_1}, \quad \frac{1}{1-\beta_1/\beta_2} \leq \frac{1}{\varepsilon_2}. \tag{64}$$

Therefore combining (63), (64) and (53), we could verify that $G^2 \sim \mathcal{O}\left(\mathrm{poly}(\log T)\right)$ with respect to $T$. Finally, using the convergence result in (62), we obtain the desired result. $\qquad\square$

## C  Convergence of Adam without corrective terms and RMSProp under smoothness

In this section, we will prove the convergence bounds in Theorem 3.2 for Adam without corrective terms and RMSProp, which are adaptive to the noise level $\sigma_0$. We first present the algorithm.

---

**Algorithm 2** Adam without corrective terms

---

**Input:** Horizon $T$, $\boldsymbol{x}_1 \in \mathbb{R}^d$, $\beta_1, \beta_2 \in [0,1)$, $\boldsymbol{m}_0 = \boldsymbol{v}_0 = \boldsymbol{0}_d$, $\eta, \epsilon > 0$, $\boldsymbol{\epsilon} = \epsilon\boldsymbol{1}_d$
**for** $s = 1, \cdots, T$ **do**
    Draw a new sample $\boldsymbol{z}_s$ and generate $\boldsymbol{g}_s = g(\boldsymbol{x}_s, \boldsymbol{z}_s)$;
    $\boldsymbol{m}_s = \beta_1\boldsymbol{m}_{s-1} + (1-\beta_1)\boldsymbol{g}_s$;
    $\boldsymbol{v}_s = \beta_2\boldsymbol{v}_{s-1} + (1-\beta_2)\boldsymbol{g}_s^2$;
    $\boldsymbol{x}_{s+1} = \boldsymbol{x}_s - \eta \cdot \boldsymbol{m}_s/\left(\sqrt{\boldsymbol{v}_s} + \boldsymbol{\epsilon}\right)$;
**end for**

---

*Remark* C.1. Setting $\beta_1 = 0$, Algorithm 2 reduces to RMSProp.

Then, we will state the formal version for Theorem 3.2.

**Theorem C.2** (Adam without corrective terms/RMSProp)**.** *Let $T \geq 1$ and $\{\boldsymbol{x}_s\}_{s\in[T]}$ be the sequence generated by Algorithm 2. If Assumptions (A1)-(A3) hold, and the hyper-parameters are given in (6), then for any given $\delta \in (0, 1/2)$, it holds that with probability at least $1 - 2\delta$,*

$$\frac{1}{T}\sum_{s=1}^T \|\nabla f(\boldsymbol{x}_s)\|^2 \leq \tilde{\mathcal{O}}\left(\frac{\tilde{G}^4(1+\sigma_1^2) + \tilde{G}^2(\sigma_0 + \sigma_1\tilde{G}^{p/2} + \tilde{G} + \epsilon_0)}{T} + \frac{\tilde{G}^2\sigma_0}{\sqrt{T}}\right), \tag{65}$$

*where $\tilde{G}^2$ is defined by the following order with respect to $T, \epsilon_0, \delta$:[5]*

$$\tilde{G}^2 \sim \mathcal{O}\left[\left(\frac{\log^{3/2}\left(T/(\epsilon_0\delta)\right)}{1-\beta_1}\right)^{\max\{2, \frac{4}{4-p}\}}\right]. \tag{66}$$

---

[5]The detailed expression of $\tilde{G}^2$ could be found in (88).

*Remark* C.3. (1). Setting $\beta_1 = 0$, Theorem C.2 then shows that RMSProp with the hyper-parameter setup in (6) can also find a stationary point with the convergence rate given in (65).

(2). The convergence rate in Theorem C.2 is of order $\tilde{\mathcal{O}}\left(1/T + \sigma_0/\sqrt{T}\right)$, which can be accelerated to $\tilde{\mathcal{O}}(1/T)$ rate when $\sigma_0$ is sufficiently low. This form matches the ones for SGD [4] and AdaGrad [2, 19], as well as the lower bound for first-order gradient method [1] on non-convex smooth optimization with affine variance noise up to logarithm factors.

## C.1 Proof details

To start with, we follow the same notations of $\mathcal{M}_T, G_s, \mathcal{G}_T(s)$ in (31) and define

$$\tilde{\boldsymbol{b}}_s = (\tilde{b}_{s,i})_i = \sqrt{\beta_2 \boldsymbol{v}_{s-1} + (1-\beta_2)\boldsymbol{g}_s^2} + \epsilon,$$

$$\tilde{\boldsymbol{a}}_s = (\tilde{a}_{s,i})_i = \sqrt{\beta_2 \boldsymbol{v}_{s-1} + (1-\beta_2)\left(\mathcal{G}_T(s)\mathbf{1}_d\right)^2} + \epsilon. \tag{67}$$

We also follow the definition of $\boldsymbol{y}_s$ in (17) and obtain from Algorithm 2 that

$$\boldsymbol{y}_{s+1} = \boldsymbol{y}_s - \eta \cdot \frac{\boldsymbol{g}_s}{\tilde{\boldsymbol{b}}_s} + \frac{\beta_1}{1-\beta_1}\left(\frac{\tilde{\boldsymbol{b}}_{s-1}}{\tilde{\boldsymbol{b}}_s} - \mathbf{1}_d\right) \odot (\boldsymbol{x}_s - \boldsymbol{x}_{s-1}). \tag{68}$$

Then, we have the following claims.

**Proposition C.4.** *Setting $\eta_s = \eta$ and $\eta = C_0\sqrt{1-\beta_2}$, the results in Lemma B.1, Lemma B.2, Lemma B.3, Lemma B.5, Lemma B.6, Lemma B.8 and Lemma B.10 remain unchanged.*

*Proof.* Using $\eta_s = \eta$ and following the proof for the above lemmas, it's easy to verify that these lemmas still hold. $\qquad\square$

The result in Lemma B.9 can be improved to the following one.

**Lemma C.5.** *Given $T \geq 1$. If $\tilde{\boldsymbol{b}}_s = (\tilde{b}_{s,i})_i$ and $\tilde{\boldsymbol{a}}_s = (\tilde{a}_{s,i})_i$ are defined in (67), and (36) holds, then for all $s \in [T], i \in [d]$,*

$$\left|\frac{1}{\tilde{a}_{s,i}} - \frac{1}{\tilde{b}_{s,i}}\right| \leq \frac{\mathcal{G}_T(s)\sqrt{1-\beta_2}}{\tilde{a}_{s,i}\tilde{b}_{s,i}} \quad and \quad \left|\frac{1}{\tilde{a}_{s,i}} - \frac{1}{\tilde{b}_{s-1,i}}\right| \leq \frac{\mathcal{G}_T(s)\sqrt{1-\beta_2}}{\tilde{a}_{s,i}\tilde{b}_{s-1,i}}.$$

Based on these results, we are able to show that the gradient norm is bounded along the training process generated by Algorithm 2.

**Proposition C.6.** *Under the same conditions in Theorem C.2, for any given $\delta \in (0, 1/2)$, it holds that with probability at least $1 - 2\delta$,*

$$\|\bar{\boldsymbol{g}}_t\|^2 \leq \tilde{G}^2, \quad \|\boldsymbol{g}_t\|^2 \leq \mathcal{G}_T(t) \leq \tilde{\mathcal{G}}_T^2, \quad \forall t \in [T+1], \tag{69}$$

*and*

$$\|\bar{\boldsymbol{g}}_{t+1}\|^2 \leq \tilde{G}^2 - L\eta \sum_{s=1}^{t}\left\|\frac{\bar{\boldsymbol{g}}_s}{\sqrt{\tilde{\boldsymbol{a}}_s}}\right\|^2, \quad \forall t \in [T], \tag{70}$$

*where $\tilde{G}^2, \tilde{\mathcal{G}}_T$ are as in (88).*

*Proof.* Let $\tilde{\Delta}_s = \frac{\tilde{\boldsymbol{b}}_{s-1}}{\tilde{\boldsymbol{b}}_s} - \mathbf{1}_d$. We also start from the descent lemma and and use (68),

$$f(\boldsymbol{y}_{t+1}) \leq f(\boldsymbol{x}_1) + \underbrace{\sum_{s=1}^{t} -\eta\left\langle\nabla f(\boldsymbol{y}_s), \frac{\boldsymbol{g}_s}{\tilde{\boldsymbol{b}}_s}\right\rangle}_{\tilde{A}} + \underbrace{\frac{\beta_1}{1-\beta_1}\sum_{s=1}^{t}\left\langle\tilde{\Delta}_s \odot (\boldsymbol{x}_s - \boldsymbol{x}_{s-1}), \nabla f(\boldsymbol{y}_s)\right\rangle}_{\tilde{B}}$$

$$+ \underbrace{\frac{L}{2}\sum_{s=1}^{t}\left\|\eta \cdot \frac{\boldsymbol{g}_s}{\tilde{\boldsymbol{b}}_s} - \frac{\beta_1}{1-\beta_1}(\tilde{\Delta}_s \odot (\boldsymbol{x}_s - \boldsymbol{x}_{s-1}))\right\|^2}_{\tilde{C}}, \tag{71}$$

The following estimation is based on the probability event in Lemma B.6.

**Bounding $\tilde{\mathbf{A}}$.** We first have

$$\tilde{\mathbf{A}} = \underbrace{-\eta \sum_{s=1}^{t} \left\langle \bar{\boldsymbol{g}}_s, \frac{\boldsymbol{g}_s}{\tilde{\boldsymbol{b}}_s} \right\rangle}_{\tilde{\mathbf{A}}.1} + \underbrace{\eta \sum_{s=1}^{t} \left\langle \bar{\boldsymbol{g}}_s - \nabla f(\boldsymbol{y}_s), \frac{\boldsymbol{g}_s}{\tilde{\boldsymbol{b}}_s} \right\rangle}_{\tilde{\mathbf{A}}.2}, \tag{72}$$

Introducing $\tilde{\boldsymbol{a}}_s$ defined in (67) into $\tilde{\mathbf{A}}.1$

$$\tilde{\mathbf{A}}.1 = -\eta \sum_{s=1}^{t} \left\| \frac{\bar{\boldsymbol{g}}_s}{\sqrt{\tilde{\boldsymbol{a}}_s}} \right\|^2 \underbrace{-\eta \sum_{s=1}^{t} \left\langle \bar{\boldsymbol{g}}_s, \frac{\boldsymbol{\xi}_s}{\tilde{\boldsymbol{a}}_s} \right\rangle}_{\tilde{\mathbf{A}}.1.1} + \underbrace{\eta \sum_{s=1}^{t} \left\langle \bar{\boldsymbol{g}}_s, \left(\frac{1}{\tilde{\boldsymbol{a}}_s} - \frac{1}{\tilde{\boldsymbol{b}}_s}\right) \boldsymbol{g}_s \right\rangle}_{\tilde{\mathbf{A}}.1.2}. \tag{73}$$

Following the proof for Lemma B.7 with $\eta_s = \eta$, we obtain that with probability at least $1 - \delta$,

$$\tilde{\mathbf{A}}.1.1 \leq \frac{3\lambda \eta^2 \mathcal{G}_T(t)}{4\sqrt{1-\beta_2}} \sum_{s=1}^{t} \left\| \frac{\bar{\boldsymbol{g}}_s}{\sqrt{\tilde{\boldsymbol{a}}_s}} \right\|^2 + \frac{1}{\lambda} \log\left(\frac{T}{\delta}\right), \forall t \in [T]. \tag{74}$$

Setting $\lambda = \sqrt{1-\beta_2}/(3\eta \tilde{\mathcal{G}}_T)$, we obtain that with probability at least $1 - \delta$,

$$\tilde{\mathbf{A}}.1.1 \leq \frac{\eta \mathcal{G}_T(t)}{4\tilde{\mathcal{G}}_T} \sum_{s=1}^{t} \left\| \frac{\bar{\boldsymbol{g}}_s}{\sqrt{\tilde{\boldsymbol{a}}_s}} \right\|^2 + \frac{3\eta \tilde{\mathcal{G}}_T}{\sqrt{1-\beta_2}} \log\left(\frac{T}{\delta}\right), \forall t \in [T]. \tag{75}$$

Using Lemma C.5 and following the similar deduction for bounding **(A.1.2)** in Lemma B.11, we have

$$\tilde{\mathbf{A}}.1.2 \leq \frac{\eta}{4} \sum_{s=1}^{t} \left\| \frac{\bar{\boldsymbol{g}}_s}{\sqrt{\tilde{\boldsymbol{a}}_s}} \right\|^2 + \eta\sqrt{1-\beta_2}\mathcal{G}_T(t) \sum_{s=1}^{t} \left\| \frac{\boldsymbol{g}_s}{\tilde{\boldsymbol{b}}_s} \right\|^2. \tag{76}$$

Plugging (75) and (76) into (73), we obtain that

$$\tilde{\mathbf{A}}.1 \leq \left(\frac{\mathcal{G}_T(t)}{4\tilde{\mathcal{G}}_T} - \frac{3}{4}\right) \sum_{s=1}^{t} \eta \left\| \frac{\bar{\boldsymbol{g}}_s}{\sqrt{\tilde{\boldsymbol{a}}_s}} \right\|^2 + \frac{3\eta \tilde{\mathcal{G}}_T}{\sqrt{1-\beta_2}} \log\left(\frac{T}{\delta}\right) + \eta\sqrt{1-\beta_2}\mathcal{G}_T(t) \sum_{s=1}^{t} \left\| \frac{\boldsymbol{g}_s}{\tilde{\boldsymbol{b}}_s} \right\|^2. \tag{77}$$

The estimation for $\tilde{\mathbf{A}}.2$ is similar to (45) and (47). Using (44) and Young's inequality,

$$\eta \left\langle \bar{\boldsymbol{g}}_s - \nabla f(\boldsymbol{y}_s), \frac{\boldsymbol{g}_s}{\tilde{\boldsymbol{b}}_s} \right\rangle \leq \frac{1}{2L} \|\bar{\boldsymbol{g}}_s - \nabla f(\boldsymbol{y}_s)\|^2 + \frac{L\eta^2}{2} \left\| \frac{\boldsymbol{g}_s}{\tilde{\boldsymbol{b}}_s} \right\|^2$$

$$\leq \frac{L}{2(1-\beta_1)^2} \|\boldsymbol{x}_s - \boldsymbol{x}_{s-1}\|^2 + \frac{L\eta^2}{2} \left\| \frac{\boldsymbol{g}_s}{\tilde{\boldsymbol{b}}_s} \right\|^2.$$

Thereby,

$$\tilde{\mathbf{A}}.2 \leq \frac{L\eta^2}{2(1-\beta_1)^2} \sum_{s=1}^{t} \left\| \frac{\boldsymbol{m}_{s-1}}{\tilde{\boldsymbol{b}}_{s-1}} \right\|^2 + \frac{L\eta^2}{2} \sum_{s=1}^{t} \left\| \frac{\boldsymbol{g}_s}{\tilde{\boldsymbol{b}}_s} \right\|^2. \tag{78}$$

**Bounding $\tilde{\mathbf{B}}, \tilde{\mathbf{C}}$.** Following the same decomposition in (35) with $\Delta_s$ replaced by $\tilde{\Delta}_s$,

$$\tilde{\mathbf{B}} = \underbrace{\frac{\beta_1}{1-\beta_1} \sum_{s=1}^{t} \left\langle \tilde{\Delta}_s \odot (\boldsymbol{x}_s - \boldsymbol{x}_{s-1}), \bar{\boldsymbol{g}}_s \right\rangle}_{\tilde{\mathbf{B}}.1} + \underbrace{\frac{\beta_1}{1-\beta_1} \sum_{s=1}^{t} \left\langle \tilde{\Delta}_s \odot (\boldsymbol{x}_s - \boldsymbol{x}_{s-1}), \nabla f(\boldsymbol{y}_s) - \bar{\boldsymbol{g}}_s \right\rangle}_{\tilde{\mathbf{B}}.2}. \tag{79}$$

We define $\tilde{\Sigma} := \frac{\beta_1}{1-\beta_1} \left\langle \tilde{\Delta}_s \odot (\boldsymbol{x}_s - \boldsymbol{x}_{s-1}), \bar{\boldsymbol{g}}_s \right\rangle$. Since $\eta_s = \eta$, we have the following decomposition

$$\tilde{\Sigma} \leq \underbrace{\frac{\eta\beta_1}{1-\beta_1} \cdot \left|\left\langle \left(\frac{1}{\tilde{\boldsymbol{b}}_s} - \frac{1}{\tilde{\boldsymbol{a}}_s}\right) \odot \boldsymbol{m}_{s-1}, \bar{\boldsymbol{g}}_s \right\rangle\right|}_{\tilde{\Sigma}_1} + \underbrace{\frac{\eta\beta_1}{1-\beta_1} \cdot \left|\left\langle \left(\frac{1}{\tilde{\boldsymbol{a}}_s} - \frac{1}{\tilde{\boldsymbol{b}}_{s-1}}\right) \odot \boldsymbol{m}_{s-1}, \bar{\boldsymbol{g}}_s \right\rangle\right|}_{\tilde{\Sigma}_2}. \tag{80}$$

The additional $\Sigma_3$ in (159) vanishes. Then, using Lemma C.5 and Young's inequality

$$\tilde{\Sigma}_1 \le \sum_{i=1}^{d} \frac{\beta_1}{1-\beta_1} \cdot \frac{\eta \mathcal{G}_T(s)\sqrt{1-\beta_2}}{\tilde{a}_{s,i}\tilde{b}_{s,i}} \cdot |\bar{g}_{s,i} m_{s-1,i}|$$

$$\le \sum_{i=1}^{d} \frac{\eta}{8} \cdot \frac{\bar{g}_{s,i}^2}{\tilde{a}_{s,i}} + \frac{2\eta\beta_1^2(1-\beta_2)}{(1-\beta_1)^2} \sum_{i=1}^{d} \frac{(\mathcal{G}_T(s))^2}{\tilde{a}_{s,i}} \cdot \frac{m_{s-1,i}^2}{\tilde{b}_{s,i}^2}$$

$$\le \frac{\eta}{8}\left\| \frac{\bar{g}_s}{\sqrt{\tilde{a}_s}} \right\|^2 + \frac{2\eta\mathcal{G}_T(t)\sqrt{1-\beta_2}}{(1-\beta_1)^2}\left\| \frac{m_{s-1}}{\tilde{b}_s} \right\|^2. \tag{81}$$

Similarly, using Lemma C.5 and Young's inequality

$$\tilde{\Sigma}_2 \le \frac{\eta}{8}\left\| \frac{\bar{g}_s}{\sqrt{\tilde{a}_s}} \right\|^2 + \frac{2\eta\mathcal{G}_T(t)\sqrt{1-\beta_2}}{(1-\beta_1)^2}\left\| \frac{m_{s-1}}{\tilde{b}_{s-1}} \right\|^2. \tag{82}$$

Then, summing up (81) and (82) over $s \in [t]$ and combining with (80),

$$\tilde{\mathbf{B}}.\mathbf{1} \le \frac{\eta}{4}\sum_{s=1}^{t}\left\| \frac{\bar{g}_s}{\sqrt{\tilde{a}_s}} \right\|^2 + \frac{2\eta\mathcal{G}_T(t)\sqrt{1-\beta_2}}{(1-\beta_1)^2}\sum_{s=1}^{t}\left( \left\| \frac{m_{s-1}}{\tilde{b}_s} \right\|^2 + \left\| \frac{m_{s-1}}{\tilde{b}_{s-1}} \right\|^2 \right). \tag{83}$$

Following the similar deduction in (48) and using Lemma B.1, we have

$$\tilde{\mathbf{B}}.\mathbf{2} \le \frac{L\Sigma_{\max}^2\eta^2}{(1-\beta_1)^2}\sum_{s=1}^{t}\left\| \frac{m_{s-1}}{\tilde{b}_{s-1}} \right\|^2 \le \frac{L\eta^2}{\beta_2(1-\beta_1)^2}\sum_{s=1}^{t}\left\| \frac{m_{s-1}}{\tilde{b}_{s-1}} \right\|^2. \tag{84}$$

Using Young's inequality and Lemma B.1,

$$\tilde{\mathbf{C}} \le L\eta^2 \sum_{s=1}^{t}\left\| \frac{g_s}{\tilde{b}_s} \right\|^2 + \frac{L\beta_1^2}{(1-\beta_1)^2}\sum_{s=1}^{t}\left\| \tilde{\Delta}_s \right\|_\infty^2 \| x_s - x_{s-1} \|^2$$

$$\le L\eta^2 \sum_{s=1}^{t}\left\| \frac{g_s}{\tilde{b}_s} \right\|^2 + \frac{L\eta^2\Sigma_{\max}^2}{(1-\beta_1)^2}\sum_{s=1}^{t}\left\| \frac{m_{s-1}}{\tilde{b}_{s-1}} \right\|^2$$

$$\le L\eta^2 \sum_{s=1}^{t}\left\| \frac{g_s}{\tilde{b}_s} \right\|^2 + \frac{L\eta^2}{\beta_2(1-\beta_1)^2}\sum_{s=1}^{t}\left\| \frac{m_{s-1}}{\tilde{b}_{s-1}} \right\|^2. \tag{85}$$

**Putting together.** Noting that the above results are established based on probability events in Lemma B.6 and (75). Hence, plugging (77), (78), (83), (84) and (85) into (71) and using $\beta_2 < 1$, it holds that with probability at least $1 - 2\delta$, for all $t \in [T]$,

$$f(y_{t+1}) \le f(x_1) + \eta\left( \frac{\mathcal{G}_T(t)}{4\tilde{\mathcal{G}}_T} - \frac{1}{2} \right)\sum_{s=1}^{t}\left\| \frac{\bar{g}_s}{\sqrt{\tilde{a}_s}} \right\|^2 + \frac{3\eta\tilde{\mathcal{G}}_T}{\sqrt{1-\beta_2}}\log\left( \frac{T}{\delta} \right)$$

$$+ \left( \frac{2\eta\sqrt{1-\beta_2}\mathcal{G}_T(t)}{(1-\beta_1)^2} + \frac{5L\eta^2}{2\beta_2(1-\beta_1)^2} \right)\sum_{s=1}^{t}\left\| \frac{m_{s-1}}{\tilde{b}_{s-1}} \right\|^2$$

$$+ \frac{2\eta\sqrt{1-\beta_2}\mathcal{G}_T(t)}{(1-\beta_1)^2}\sum_{s=1}^{t}\left\| \frac{m_{s-1}}{\tilde{b}_s} \right\|^2 + \left( \eta\sqrt{1-\beta_2}\mathcal{G}_T(t) + \frac{3L\eta^2}{2} \right)\sum_{s=1}^{t}\left\| \frac{g_s}{\tilde{b}_s} \right\|^2.$$

Then, combining with Lemma B.3 and Lemma B.10 and using $\eta = C_0\sqrt{1-\beta_2}$, we obtain that with probability at least $1 - 2\delta$, for all $t \in [T]$,

$$f(y_{t+1}) \le f(x_1) + \eta\left( \frac{\mathcal{G}_T(t)}{4\tilde{\mathcal{G}}_T} - \frac{1}{2} \right)\sum_{s=1}^{t}\left\| \frac{\bar{g}_s}{\sqrt{\tilde{a}_s}} \right\|^2 + E_1\tilde{\mathcal{G}}_T + E_2\mathcal{G}_T(t) + E_3. \tag{86}$$

where the coefficients are defined as

$$E_1 = 3C_0\log\left( \frac{T}{\delta} \right), E_2 = d\left( C_0 + \frac{4C_0}{\beta_2(1-\beta_1)(1-\beta_1/\beta_2)} \right)\log\left( \frac{\mathcal{F}(T)}{\beta_2^T} \right),$$

$$E_3 = d\left( \frac{5LC_0^2}{2\beta_2(1-\beta_1)(1-\beta_1/\beta_2)} + \frac{3LC_0^2}{2(1-\beta_1/\beta_2)} \right)\log\left( \frac{\mathcal{F}(T)}{\beta_2^T} \right). \tag{87}$$

**An induction argument.** Noting that (86) holds with probability at least $1 - 2\delta$ for all $t \in [T]$, then we will provide an induction to prove the desired result based on (86), which also holds with probability at least $1 - 2\delta$. We first provide the detailed expression of $\tilde{G}^2$ and $\tilde{\mathcal{G}}_T$ as follows:

$$\tilde{G}^2 = 8L(f(\boldsymbol{x}_1) - f^*) + 128L^2\mathcal{M}_T^2(E_1 + E_2)^2 + \frac{(4-p)}{2}p^{\frac{p}{4-p}}\left(8L\sigma_1\mathcal{M}_T(E_1 + E_3)\right)^{\frac{4}{4-p}}$$

$$+ 16L\mathcal{M}_T(E_1 + E_2)\sigma_0 + 8LE_3 + 4M^2,$$

$$\tilde{\mathcal{G}}_T = \mathcal{M}_T\sqrt{2\sigma_0^2 + 2\sigma_1^2\tilde{G}^p + 2\tilde{G}^2}, \tag{88}$$

where $E_1, E_2, E_3$ are defined in (87) and $M$ is as in Lemma B.5. The induction then begins by noting that $G_1^2 = \|\bar{\boldsymbol{g}}_1\|^2 \leq 2L(f(\boldsymbol{x}_1) - f^*) \leq \tilde{G}^2$ from Lemma B.4. Then, we assume that for some $t \in [T]$,

$$G_s \leq \tilde{G}, \quad \forall s \in [t] \quad \text{consequently} \quad \mathcal{G}_T(s) \leq \tilde{\mathcal{G}}_T, \quad \forall s \in [t].$$

Using this induction assumption over (86) and $\tilde{\mathcal{G}}_T \leq 2\mathcal{M}_T\left(\sigma_0 + \sigma_1\tilde{G}^{p/2} + \tilde{G}\right)$,

$$f(\boldsymbol{y}_{t+1}) \leq f(\boldsymbol{x}_1) - \frac{\eta}{4}\sum_{s=1}^{t}\left\|\frac{\bar{\boldsymbol{g}}_s}{\sqrt{\tilde{\boldsymbol{a}}_s}}\right\|^2 + 2\mathcal{M}_T(E_1 + E_2)\left(\sigma_0 + \sigma_1\tilde{G}^{p/2} + \tilde{G}\right) + E_3. \tag{89}$$

Following the same result in (57), we have

$$\|\bar{\boldsymbol{g}}_{t+1}\|^2 \leq 4L(f(\boldsymbol{y}_{t+1}) - f^*) + 2M^2. \tag{90}$$

Combining with (89) and (90),

$$\|\bar{\boldsymbol{g}}_{t+1}\|^2 \leq 4L(f(\boldsymbol{x}_1) - f^*) - L\eta\sum_{s=1}^{t}\left\|\frac{\bar{\boldsymbol{g}}_s}{\sqrt{\tilde{\boldsymbol{a}}_s}}\right\|^2$$

$$+ 8\mathcal{M}_TL(E_1 + E_2)\left(\sigma_0 + \sigma_1\tilde{G}^{p/2} + \tilde{G}\right) + 4LE_3 + 2M^2. \tag{91}$$

Using two Young's inequalities where $ab \leq \frac{a^2}{2} + \frac{b^2}{2}$ and $ab^{\frac{p}{2}} \leq \frac{4-p}{4} \cdot a^{\frac{4}{4-p}} + \frac{p}{4} \cdot b^2, \forall a, b \geq 0$,

$$\|\bar{\boldsymbol{g}}_{t+1}\|^2 \leq \frac{\tilde{G}^2}{2} + 4L(f(\boldsymbol{x}_1) - f^*) - L\eta\sum_{s=1}^{t}\left\|\frac{\bar{\boldsymbol{g}}_s}{\sqrt{\tilde{\boldsymbol{a}}_s}}\right\|^2 + 64L^2\mathcal{M}_T^2(E_1 + E_2)^2$$

$$+ \frac{(4-p)}{4}p^{\frac{p}{4-p}}\left(8L\sigma_1\mathcal{M}_T(E_1 + E_3)\right)^{\frac{4}{4-p}} + 8L\mathcal{M}_T(E_1 + E_2)\sigma_0 + 4LE_3 + 2M^2$$

$$\leq \tilde{G}^2, \tag{92}$$

where the last inequality comes from the definition of $\tilde{G}$ in (88). Hence, the induction is complete and we obtain the desired result in (69). Furthermore, as a consequence of (92), we also prove that (70) holds. $\qquad\square$

## C.2 Proof of the main result

Now we are ready to prove the main convergence result.

*Proof of Theorem C.2.* In what follows, we will assume that Proposition C.6 holds. Hence, the final convergence bound also holds with probability at least $1 - 2\delta$. We set $t = T$ in (70) to obtain that with probability at least $1 - 2\delta$,

$$L\eta\sum_{s=1}^{T}\frac{\|\bar{\boldsymbol{g}}_s\|^2}{\|\tilde{\boldsymbol{a}}_s\|_\infty} \leq L\eta\sum_{s=1}^{T}\left\|\frac{\bar{\boldsymbol{g}}_s}{\sqrt{\tilde{\boldsymbol{a}}_s}}\right\|^2 \leq \tilde{G}^2 - \|\bar{\boldsymbol{g}}_{T+1}\|^2 \leq \tilde{G}^2. \tag{93}$$

Using Lemma B.6, we derive that $\forall s \in [T]$,

$$\|\boldsymbol{g}_s\|^2 \leq 2\|\boldsymbol{\xi}_s\|^2 + 2\|\bar{\boldsymbol{g}}_s\|^2 \leq 2\mathcal{M}_T^2\sigma_0^2 + 2\mathcal{M}_T^2(1 + \sigma_1^2)\|\bar{\boldsymbol{g}}_s\|^2. \tag{94}$$

Applying $\tilde{\boldsymbol{a}}_s$ in (67), Proposition C.6, (94) and $\eta = C_0\sqrt{1-\beta_2}, \epsilon = \epsilon_0\sqrt{1-\beta_2}$, we have for any $s \in [T]$,

$$
\begin{aligned}
\frac{\|\tilde{\boldsymbol{a}}_s\|_\infty}{\eta} &= \frac{1}{\eta}\left(\max_{i\in[d]}\sqrt{\beta_2 v_{s-1,i} + (1-\beta_2)(\mathcal{G}_T(s))^2} + \epsilon\right) \\
&\leq \frac{1}{\eta}\left(\sqrt{(1-\beta_2)\sum_{j=1}^T \beta_2^{s-j}\|\boldsymbol{g}_j\|^2} + \sqrt{1-\beta_2}\tilde{\mathcal{G}}_T + \epsilon\right) \\
&\leq \frac{\mathcal{M}_T}{C_0}\sqrt{2(1+\sigma_1^2)\sum_{j=1}^T \|\bar{\boldsymbol{g}}_j\|^2} + \frac{\mathcal{M}_T\sigma_0\sqrt{2T} + \tilde{\mathcal{G}}_T + \epsilon_0}{C_0}.
\end{aligned}
\tag{95}
$$

We therefore combine with (93) to obtain that

$$
\begin{aligned}
\sum_{s=1}^T \|\bar{\boldsymbol{g}}_s\|^2 &\leq \frac{\tilde{G}^2}{LC_0}\left(\mathcal{M}_T\sqrt{2(1+\sigma_1^2)\sum_{j=1}^T \|\bar{\boldsymbol{g}}_j\|^2} + \mathcal{M}_T\sigma_0\sqrt{2T} + \tilde{\mathcal{G}}_T + \epsilon_0\right) \\
&\leq \frac{1}{2}\sum_{s=1}^T \|\bar{\boldsymbol{g}}_s\|^2 + \frac{\tilde{G}^4\mathcal{M}_T^2(1+\sigma_1^2)}{L^2C_0^2} + \frac{\tilde{G}^2}{LC_0}\left(\mathcal{M}_T\sigma_0\sqrt{2T} + \tilde{\mathcal{G}}_T + \epsilon_0\right),
\end{aligned}
$$

which leads to

$$
\frac{1}{T}\sum_{s=1}^T \|\bar{\boldsymbol{g}}_s\|^2 \leq \frac{1}{T}\left(\frac{2\tilde{G}^4\mathcal{M}_T^2(1+\sigma_1^2)}{L^2C_0^2} + \frac{2\tilde{G}^2(\tilde{\mathcal{G}}_T + \epsilon_0)}{LC_0}\right) + \frac{1}{\sqrt{T}}\cdot\frac{2\sqrt{2}\tilde{G}^2\mathcal{M}_T\sigma_0}{LC_0}.
\tag{96}
$$

Following the result in (63) and (64), we show that when $\beta_2 = 1 - c/T$, $\tilde{G}^2, \tilde{\mathcal{G}}_T \sim \mathcal{O}\left(\mathrm{poly}(\log T)\right)$ with respect to $T$. Finally, using the convergence result in (96), we obtain the desired result. $\qquad\square$

## D   Proof of Theorem 4.1

In this section, we shall follow all the notations defined in Section 6. Further, we will add two non-decreasing sequences $\{\mathcal{L}_s^{(x)}\}_{s\geq 1}$ and $\{\mathcal{L}_s^{(y)}\}_{s\geq 1}$ as follows

$$
\mathcal{L}_s^{(x)} = L_0 + L_q G_s^q, \quad \mathcal{L}_s^{(y)} = L_0 + L_q(G_s + G_s^q + L_0/L_q)^q, \quad \forall s \geq 1.
\tag{97}
$$

### D.1   Preliminary

We first mention that Lemma B.1, Lemma B.2, and Lemma B.3 in Appendix B.1 remain unchanged since they are independent of the smooth condition. Then the first essential challenge is that we need to properly tune $\eta$ to restrict the distance between $\boldsymbol{x}_{s+1}$ and $\boldsymbol{x}_s$, $\boldsymbol{y}_{s+1}$ and $\boldsymbol{y}_s$ within $1/L_q$ for all $s \geq 1$. The following two lemmas then ensure this point. The detailed proofs could be found in Appendix G.

**Lemma D.1.** *Let $\boldsymbol{x}_s, \boldsymbol{y}_s$ be defined in Algorithm 1 and* (17). *If $0 \leq \beta_1 < \beta_2 < 1$, then for any $s \geq 1$,*

$$
\max\{\|\boldsymbol{x}_{s+1} - \boldsymbol{x}_s\|, \|\boldsymbol{y}_s - \boldsymbol{x}_s\|, \|\boldsymbol{y}_{s+1} - \boldsymbol{y}_s\|\} \leq \eta\sqrt{\frac{4d}{\beta_2(1-\beta_1)^2(1-\beta_2)(1-\beta_1/\beta_2)}}.
\tag{98}
$$

*As a consequence, when*

$$
\eta \leq \frac{1}{L_qF}, \quad F := \sqrt{\frac{4d}{\beta_2(1-\beta_1)^2(1-\beta_2)(1-\beta_1/\beta_2)}},
\tag{99}
$$

*then for any $s \geq 1$, all the three gaps in* (98) *are smaller than $1/L_q$.*

**Lemma D.2.** *Let $\eta \leq 1/(L_q F)$ where $F$ is as in Lemma D.1. If $f$ is $(L_0, L_q)$-smooth, then for any $s \geq 1$,*

$$\|\nabla f(\boldsymbol{y}_s)\| \leq L_0/L_q + \|\nabla f(\boldsymbol{x}_s)\|^q + \|\nabla f(\boldsymbol{x}_s)\|,$$
$$\|\nabla f(\boldsymbol{x}_s)\| \leq L_0/L_q + \|\nabla f(\boldsymbol{y}_s)\|^q + \|\nabla f(\boldsymbol{y}_s)\|.$$

*As a consequence, for any $s \geq 1$,*

$$\|\nabla f(\boldsymbol{y}_s) - \nabla f(\boldsymbol{x}_s)\| \leq \mathcal{L}_s^{(x)}\|\boldsymbol{y}_s - \boldsymbol{x}_s\|, \quad \|\nabla f(\boldsymbol{y}_{s+1}) - \nabla f(\boldsymbol{y}_s)\| \leq \mathcal{L}_s^{(y)}\|\boldsymbol{y}_{s+1} - \boldsymbol{y}_s\|, \quad (100)$$

$$f(\boldsymbol{y}_{s+1}) - f(\boldsymbol{y}_s) - \langle \nabla f(\boldsymbol{y}_s), \boldsymbol{y}_{s+1} - \boldsymbol{y}_s \rangle \leq \frac{\mathcal{L}_s^{(y)}}{2}\|\boldsymbol{y}_{s+1} - \boldsymbol{y}_s\|. \quad (101)$$

In the generalized smooth case, Lemma B.4 does not hold. In contrast, we provide a generalized smooth version of [50, Lemma A.5], which establishes a different relationship between the gradient's norm and the function value gap. Noting that when $q = 1$, Lemma D.3 reduces to [50, Lemma A.5].

**Lemma D.3.** *Suppose that $f$ is $(L_0, L_q)$-smooth and Assumption (A1) holds. Then for any $\boldsymbol{x} \in \mathbb{R}^d$,*

$$\|\nabla f(\boldsymbol{x})\| \leq \max \left\{ 4L_q(f(\boldsymbol{x}) - f^*), [4L_q(f(\boldsymbol{x}) - f^*)]^{\frac{1}{2-q}}, \sqrt{4L_0(f(\boldsymbol{x}) - f^*)} \right\}.$$

## D.2   Probabilistic estimations

The probabilistic inequalities in (36) and (37) remain unchanged since they do not rely on any smooth-related conditions. However, we shall rely on a different setting of $\lambda$ in (37) as follows.

**Lemma D.4.** *Given $T \geq 1$ and $\delta \in (0,1)$. Under the same conditions of Lemma B.7, if we set $\lambda = (1 - \beta_1)\sqrt{1 - \beta_2}/(3\eta\mathcal{H})$ where $\mathcal{H}$ is as in (11), then with probability at least $1 - \delta$,*

$$\sum_{s=1}^{t} -\eta_s \left\langle \bar{\boldsymbol{g}}_s, \frac{\boldsymbol{\xi}_s}{\boldsymbol{a}_s} \right\rangle \leq \frac{\mathcal{G}_T(t)}{4\mathcal{H}} \sum_{s=1}^{t} \eta_s \left\| \frac{\bar{\boldsymbol{g}}_s}{\sqrt{\boldsymbol{a}_s}} \right\|^2 + D_1\mathcal{H}, \quad \forall t \in [T], \quad (102)$$

*where $D_1$ is given in Lemma B.7.*

The bounds of the four summations in Lemma B.3 also remain unchanged. However, the upper bound for $\mathcal{F}_i(t)$ should be revised by the following lemma. The detailed proof could be found in Appendix G.

**Lemma D.5.** *Given $T \geq 1$. Under the conditions and notations of Lemma B.3, if $f$ is $(L_0, L_q)$-smooth, $\eta = \tilde{C}_0\sqrt{1 - \beta_2}$, (36) and (99) hold, then the following inequalities hold,*

$$\mathcal{F}_i(t) \leq \mathcal{J}(t), \quad \forall t \in [T], i \in [d], \quad (103)$$

*where $\mathcal{J}(t)$ is defined as*

$$\mathcal{J}(t) := 1 + \frac{2\mathcal{M}_T^2}{\epsilon^2} \left[ \sigma_0^2 t + \sigma_1^2 t \left( \|\bar{\boldsymbol{g}}_1\| + t\tilde{M}_t \right)^p + t \left( \|\bar{\boldsymbol{g}}_1\| + t\tilde{M}_t \right)^2 \right], \quad (104)$$

*and $\tilde{M}_t := \tilde{C}_0 \mathcal{L}_t^{(x)} \sqrt{\frac{d}{1 - \beta_1/\beta_2}}$.*

It's worth noting that $\mathcal{J}(t)$ is still random relying on the random variable $\mathcal{L}_t^{(x)}$.

## D.3   Deterministic estimations

Note that (41) in Appendix B.4 remains unchanged since it's independent from any smooth-related condition. In terms of **A.1**, the only difference is using $\mathcal{H}$ to replace $\mathcal{G}$ in (40) as we choose a different $\lambda$ in (37), leading to

$$\mathbf{A.1} \leq \left( \frac{\mathcal{G}_T(t)}{4\mathcal{H}} - \frac{3}{4} \right) \sum_{s=1}^{t} \eta_s \left\| \frac{\bar{\boldsymbol{g}}_s}{\sqrt{\boldsymbol{a}_s}} \right\|^2 + D_1\mathcal{H} + D_2\mathcal{G}_T(t) \sum_{s=1}^{t} \left\| \frac{\boldsymbol{g}_s}{\boldsymbol{b}_s} \right\|^2. \quad (105)$$

We also establish the following proposition which is a generalized smooth version of Proposition B.13.

**Proposition D.6.** *Given $T \geq 1$. If $f$ is $(L_0, L_q)$-smooth and (99) holds, then*

$$f(\boldsymbol{y}_{t+1}) \leq f(\boldsymbol{x}_1) + \mathbf{A.1} + \mathbf{B.1} + \sum_{s=1}^{t-1} D_6(s) \left\| \frac{\hat{\boldsymbol{m}}_s}{\boldsymbol{b}_s} \right\|^2 + \sum_{s=1}^{t} D_7(s) \left\| \frac{\boldsymbol{g}_s}{\boldsymbol{b}_s} \right\|^2, \quad \forall t \in [T], \quad (106)$$

*where $\Sigma_{\max}$ is as in Lemma B.1 and $D_6(s), D_7(s)$ are defined as,[6]*

$$D_6(s) = \frac{\mathcal{L}_s^{(y)} \eta^2 (1 + 4\Sigma_{\max}^2)}{2(1 - \beta_1)^2}, \quad D_7(s) = \frac{3\mathcal{L}_s^{(y)} \eta^2}{2(1 - \beta_1)^2}.$$

*Proof.* The proof follows some same parts in proving Proposition B.14. We start from the descent lemma (101) in Lemma D.2 and sum over $s \in [t]$ to obtain that

$$f(\boldsymbol{y}_{t+1}) \leq f(\boldsymbol{x}_1) + \sum_{s=1}^{t} \langle \nabla f(\boldsymbol{y}_s), \boldsymbol{y}_{s+1} - \boldsymbol{y}_s \rangle + \sum_{s=1}^{t} \frac{\mathcal{L}_s^{(y)}}{2} \|\boldsymbol{y}_{s+1} - \boldsymbol{y}_s\|^2$$

$$= f(\boldsymbol{x}_1) + \mathbf{A} + \mathbf{B} + \underbrace{\sum_{s=1}^{t} \frac{\mathcal{L}_s^{(y)}}{2} \left\| \eta_s \cdot \frac{\boldsymbol{g}_s}{\boldsymbol{b}_s} - \frac{\beta_1}{1 - \beta_1} \left( \frac{\eta_s \boldsymbol{b}_{s-1}}{\eta_{s-1} \boldsymbol{b}_s} - \mathbf{1} \right) \Sigma_s \odot (\boldsymbol{x}_s - \boldsymbol{x}_{s-1}) \right\|^2}_{\mathbf{C'}},$$

$$(107)$$

where $\mathbf{A}$ and $\mathbf{B}$ follow the same definitions in (33). We also follow the decompositions in (34) and (35). We could also rely on the same analysis for the smooth case in (47) but the smooth parameter is replaced by $\mathcal{L}_s^{(x)}$. Hence, we obtain that

$$\mathbf{A.2} \leq \sum_{s=1}^{t} \frac{\mathcal{L}_s^{(x)} \eta^2}{2(1 - \beta_1)^2} \left\| \frac{\hat{\boldsymbol{m}}_{s-1}}{\boldsymbol{b}_{s-1}} \right\|^2 + \sum_{s=1}^{t} \frac{\mathcal{L}_s^{(x)} \eta^2}{2(1 - \beta_1)^2} \left\| \frac{\boldsymbol{g}_s}{\boldsymbol{b}_s} \right\|^2. \quad (108)$$

Similarly,

$$\mathbf{B.2} \leq \sum_{s=1}^{t} \frac{\Sigma_{\max}^2 \mathcal{L}_s^{(x)} \eta^2}{(1 - \beta_1)^2} \left\| \frac{\hat{\boldsymbol{m}}_{s-1}}{\boldsymbol{b}_{s-1}} \right\|^2, \quad (109)$$

Noting that $\mathbf{C'}$ differs from $\mathbf{C}$ with $L$ replaced by $\mathcal{L}_s^{(y)}$. Hence, relying on a similar analysis in (49), we obtain that

$$\mathbf{C'} \leq \sum_{s=1}^{t} \frac{\mathcal{L}_s^{(y)} \eta^2}{(1 - \beta_1)^2} \left\| \frac{\boldsymbol{g}_s}{\boldsymbol{b}_s} \right\|^2 + \sum_{s=1}^{t} \frac{\Sigma_{\max}^2 \mathcal{L}_s^{(y)} \eta^2}{(1 - \beta_1)^2} \left\| \frac{\hat{\boldsymbol{m}}_{s-1}}{\boldsymbol{b}_{s-1}} \right\|^2. \quad (110)$$

Combining (107) with (108), (109) and (110), and noting that $\mathcal{L}_s^{(x)} \leq \mathcal{L}_s^{(y)}$ from (97), we thereby obtain the desired result. □

### D.4 Bounding gradients

Based on the unchanged parts in Appendix B.3 and Appendix B.4 and the new estimations in (105) and (106), we are now ready to provide the uniform gradients' bound in the following proposition.

**Proposition D.7.** *Under the same conditions in Theorem 4.1, for any given $\delta \in (0, 1/2)$, it holds that with probability at least $1 - 2\delta$,*

$$\|\bar{\boldsymbol{g}}_t\| \leq H, \quad \mathcal{G}_T(t) \leq \mathcal{H}, \quad \mathcal{L}_t^{(x)} \leq \mathcal{L}_t^{(y)} \leq \mathcal{L}, \quad \forall t \in [T + 1], \quad (111)$$

*and*

$$f(\boldsymbol{y}_{t+1}) - f^* \leq -\frac{1}{4} \sum_{s=1}^{t} \eta_s \left\| \frac{\bar{\boldsymbol{g}}_s}{\sqrt{\boldsymbol{a}_s}} \right\|^2 + \hat{H}, \quad \forall t \in [T], \quad (112)$$

*where $H, \mathcal{H}, \mathcal{L}$ are given in (11) and $\hat{H}$ is given in (115).*

---

[6] The notations are different from $D_6$ and $D_7$ defined in (43).

*Proof.* Based on the two inequalities (36) and (102), we could deduce the final results in (111) and (112). Since (36) and (102) hold with probability at least $1 - 2\delta$, we thereby deduce the desired result holding with probability at least $1 - 2\delta$. To start with, we shall verify that (99) always holds. Recalling $\eta$ in (10) and $F$ in Lemma D.1,

$$\eta F = \tilde{C}_0 \sqrt{1 - \beta_2} F \leq \sqrt{\frac{\beta_2 (1 - \beta_1)^2 (1 - \beta_2)(1 - \beta_1/\beta_2)}{4 L_q^2 d}} \cdot F \leq \frac{1}{L_q}.$$

Hence, we make sure that the distance requirement in (8) always holds according to Lemma D.1. Second, plugging (105) and (41) into the result in (106),

$$
\begin{aligned}
f(\boldsymbol{y}_{t+1}) \leq & f(\boldsymbol{x}_1) + \left( \frac{\mathcal{G}_T(t)}{4\mathcal{H}} - \frac{1}{2} \right) \sum_{s=1}^{t} \eta_s \left\| \frac{\bar{\boldsymbol{g}}_s}{\sqrt{\boldsymbol{a}_s}} \right\|^2 + D_1 \mathcal{H} + D_2 \mathcal{G}_T(t) \sum_{s=1}^{t} \left\| \frac{\boldsymbol{g}_s}{\boldsymbol{b}_s} \right\|^2 \\
& + \sum_{s=1}^{t} D_7(s) \left\| \frac{\boldsymbol{g}_s}{\boldsymbol{b}_s} \right\|^2 + (D_3 \mathcal{G}_T(t) + D_4) \sum_{s=1}^{t} \left( \left\| \frac{\boldsymbol{m}_{s-1}}{\boldsymbol{b}_s} \right\|^2 + \left\| \frac{\boldsymbol{m}_{s-1}}{\boldsymbol{b}_{s-1}} \right\|^2 \right) \\
& + D_5 G_t + \sum_{s=1}^{t-1} D_6(s) \left\| \frac{\hat{\boldsymbol{m}}_s}{\boldsymbol{b}_s} \right\|^2.
\end{aligned}
\tag{113}
$$

We still rely on an induction argument to deduce the result. First, we provide the detail expressions of $\hat{H}, H$ as follows which is determined by hyper-parameters $\beta_1, \beta_2$ and constants $E_0, d, T, \delta, \mathcal{M}_T$,

$$
\begin{aligned}
\hat{H} := & f(\boldsymbol{x}_1) - f^* + \frac{3 E_0 \mathcal{M}_T}{1 - \beta_1} \log\left( \frac{T}{\delta} \right) + \frac{E_0 \mathcal{M}_T d}{1 - \beta_1} \log\left( \frac{\tilde{\mathcal{J}}(T)}{\beta_2^T} \right) \\
& + \frac{4 E_0 (\mathcal{M}_T + \epsilon) d}{\beta_2 (1 - \beta_1)^2 (1 - \beta_1/\beta_2)} \log\left( \frac{\tilde{\mathcal{J}}(T)}{\beta_2^T} \right) + \frac{2 E_0 d}{\sqrt{(1 - \beta_1)^3 (1 - \beta_1/\beta_2)}} \\
& + \frac{3 E_0^2 d}{2(1 - \beta_1)^2} \log\left( \frac{\tilde{\mathcal{J}}(T)}{\beta_2^T} \right) + \frac{5 E_0^2 d}{2 \beta_2 (1 - \beta_1)^2 (1 - \beta_1/\beta_2)} \log\left( \frac{\tilde{\mathcal{J}}(T)}{\beta_2^T} \right),
\end{aligned}
\tag{114}
$$

$$
H := L_0/L_q + \left( 4 L_q \hat{H} \right)^q + \left( 4 L_q \hat{H} \right)^{\frac{q}{2-q}} + \left( 4 L_0 \hat{H} \right)^{\frac{q}{2}} + 4 L_q \hat{H} + \left( 4 L_q \hat{H} \right)^{\frac{1}{2-q}} + \sqrt{4 L_0 \hat{H}}.
\tag{115}
$$

where $E_0 > 0$ is a constant and $\tilde{\mathcal{J}}(T)$ is a polynomial of $T$ given as

$$
\tilde{\mathcal{J}}(T) := 1 + \frac{2 \mathcal{M}_T^2}{\epsilon^2} \left[ \sigma_0^2 T + \sigma_1^2 T \left( \|\bar{\boldsymbol{g}}_1\| + T\tilde{M} \right)^p + T \left( \|\bar{\boldsymbol{g}}_1\| + T\tilde{M} \right)^2 \right],
\tag{116}
$$

and $\tilde{M} := E_0 \sqrt{\frac{d}{1 - \beta_1/\beta_2}}$. The induction then begins by noting that from Lemma D.3 and $H$ in (115),

$$
G_1 = \|\bar{\boldsymbol{g}}_1\| \leq 4 L_q (f(\boldsymbol{x}_1) - f^*) + (4 L_q (f(\boldsymbol{x}_1) - f^*))^{\frac{1}{2-q}} + \sqrt{4 L_0 (f(\boldsymbol{x}_1) - f^*)} \leq H.
$$

Suppose that for some $t \in [T]$,

$$
G_s \leq H, \quad \forall s \in [t].
\tag{117}
$$

Consequently, recalling $\mathcal{G}_T(s)$ in (19), $\mathcal{L}_s^{(x)}, \mathcal{L}_s^{(y)}$ in (97) and $\mathcal{H}, \mathcal{L}$ in (11),

$$
\mathcal{G}_T(s) \leq \mathcal{H}, \quad \mathcal{L}_s^{(x)} \leq \mathcal{L}_s^{(y)} \leq \mathcal{L}, \quad \forall s \in [t].
\tag{118}
$$

We thus apply (118) to (113),

$$
\begin{aligned}
f(\boldsymbol{y}_{t+1}) \leq & f(\boldsymbol{x}_1) - \frac{1}{4} \sum_{s=1}^{t} \eta_s \left\| \frac{\bar{\boldsymbol{g}}_s}{\sqrt{\boldsymbol{a}_s}} \right\|^2 + D_1 \mathcal{H} + D_2 \mathcal{H} \sum_{s=1}^{t} \left\| \frac{\boldsymbol{g}_s}{\boldsymbol{b}_s} \right\|^2 + \sum_{s=1}^{t} D_7(s) \left\| \frac{\boldsymbol{g}_s}{\boldsymbol{b}_s} \right\|^2 \\
& + (D_3 \mathcal{H} + D_4) \sum_{s=1}^{t} \left( \left\| \frac{\boldsymbol{m}_{s-1}}{\boldsymbol{b}_s} \right\|^2 + \left\| \frac{\boldsymbol{m}_{s-1}}{\boldsymbol{b}_{s-1}} \right\|^2 \right) + D_5 H + \sum_{s=1}^{t-1} D_6(s) \left\| \frac{\hat{\boldsymbol{m}}_s}{\boldsymbol{b}_s} \right\|^2.
\end{aligned}
\tag{119}
$$

Further recalling the setting of $\tilde{C}_0$ in (10), with a simple calculation it holds that,

$$\tilde{C}_0 H \le \tilde{C}_0 \mathcal{H} \le E_0, \quad \tilde{C}_0 \mathcal{L} \le E_0, \quad \tilde{C}_0^2 \mathcal{L} \le E_0^2, \quad \tilde{C}_0 \epsilon_0 \le E_0 \epsilon_0. \tag{120}$$

Therefore, combining with (118), (120) and $\tilde{M}_t$ in (104), we could use the deterministic polynomial $\tilde{\mathcal{J}}(t)$ to further control $\mathcal{J}(t)$ in (104),

$$\tilde{M}_t \le \tilde{C}_0 \mathcal{L} \sqrt{\frac{d}{1 - \beta_1/\beta_2}} \le E_0 \sqrt{\frac{d}{1 - \beta_1/\beta_2}} = \tilde{M}, \quad \mathcal{J}(t) \le \tilde{\mathcal{J}}(t) \le \tilde{\mathcal{J}}(T),$$

$$\log\left(\frac{\mathcal{F}_i(t)}{\beta_2^t}\right) \le \log\left(\frac{\mathcal{J}(t)}{\beta_2^t}\right) \le \log\left(\frac{\tilde{\mathcal{J}}(T)}{\beta_2^T}\right), \quad \forall t \le T, i \in [d].$$

Then, we could use $\tilde{\mathcal{J}}(T)$ to control the four summations in Lemma B.3 which emerge in (119). In addition, we rely on $\eta = \tilde{C}_0\sqrt{1 - \beta_2}$ and the induction assumptions of (117) and (118) to further upper bound the RHS of (119), leading to

$$
\begin{aligned}
f(\boldsymbol{y}_{t+1}) - f^* \le\,& f(\boldsymbol{x}_1) - f^* - \frac{1}{4}\sum_{s=1}^{t} \eta_s \left\|\frac{\bar{\boldsymbol{g}}_s}{\sqrt{\boldsymbol{a}_s}}\right\|^2 + \frac{3\tilde{C}_0\mathcal{H}}{1 - \beta_1}\log\left(\frac{T}{\delta}\right) + \frac{\tilde{C}_0\mathcal{H}d}{1 - \beta_1}\log\left(\frac{\tilde{\mathcal{J}}(T)}{\beta_2^T}\right) \\
&+ \frac{4\tilde{C}_0(\mathcal{H} + \epsilon_0)d}{\beta_2(1 - \beta_1)^2(1 - \beta_1/\beta_2)}\log\left(\frac{\tilde{\mathcal{J}}(T)}{\beta_2^T}\right) + \frac{2\tilde{C}_0 Hd}{\sqrt{(1 - \beta_1)^3(1 - \beta_1/\beta_2)}} \\
&+ \frac{3\tilde{C}_0^2\mathcal{L}d}{2(1 - \beta_1)^2}\log\left(\frac{\tilde{\mathcal{J}}(T)}{\beta_2^T}\right) + \frac{5\tilde{C}_0^2\mathcal{L}d}{2\beta_2(1 - \beta_1)^2(1 - \beta_1/\beta_2)}\log\left(\frac{\tilde{\mathcal{J}}(T)}{\beta_2^T}\right).
\end{aligned}
\tag{121}
$$

Then, combining with (120) and the definition of $\hat{H}$ in (114), we obtain that

$$\Delta_{t+1} := f(\boldsymbol{y}_{t+1}) - f^* \le -\frac{1}{4}\sum_{s=1}^{t}\eta_s\left\|\frac{\bar{\boldsymbol{g}}_s}{\sqrt{\boldsymbol{a}_s}}\right\|^2 + \hat{H} \le \hat{H}. \tag{122}$$

Then, further using Lemma D.2, Lemma D.3 and $H$ in (115),

$$
\begin{aligned}
\|\bar{\boldsymbol{g}}_{t+1}\| &\le L_0/L_q + \|\nabla f(\boldsymbol{y}_{t+1})\|^q + \|\nabla f(\boldsymbol{y}_{t+1})\| \\
&\le L_0/L_q + (4L_q\Delta_{t+1})^q + (4L_q\Delta_{t+1})^{\frac{q}{2-q}} \\
&\quad + (4L_0\Delta_{t+1})^{\frac{q}{2}} + 4L_q\Delta_{t+1} + (4L_q\Delta_{t+1})^{\frac{1}{2-q}} + \sqrt{4L_0\Delta_{t+1}} \le H.
\end{aligned}
$$

We then deduce that $G_{t+1} = \max\{G_t, \|\bar{\boldsymbol{g}}_{t+1}\|\} \le H$. The induction is then complete and we obtain the desired result in (111). Finally, as an intermediate result of the proof, we obtain that (112) holds as well.

$\square$

## D.5 Proof of the main result

*Proof of Theorem 4.1.* The proof for the final convergence rate follows a similar idea and some same estimations in the proof of Theorem 3.1. Setting $t = T$ in (112), it holds that with probability at least $1 - 2\delta$,

$$\frac{1}{4}\sum_{s=1}^{t}\frac{\eta_s}{\|\boldsymbol{a}_s\|_\infty}\cdot\|\bar{\boldsymbol{g}}_s\|^2 \le \frac{1}{4}\sum_{s=1}^{t}\eta_s\left\|\frac{\bar{\boldsymbol{g}}_s}{\sqrt{\boldsymbol{a}_s}}\right\|^2 \le \hat{H}. \tag{123}$$

Then, in what follows, we would assume that (111) and (123) always hold. Relying on the two inequalities, we thereby deduce the final convergence result. Furthermore, since (111) and (123) hold with probability at least $1 - 2\delta$, the final convergence result also holds with probability at least $1 - 2\delta$. Using (111) and following the same analysis in (61),

$$\|\boldsymbol{a}_s\|_\infty \le \max_{i\in[d]}\sqrt{(1 - \beta_2)\left(\sum_{j=1}^{s-1}\beta_2^{s-j}g_{j,i}^2 + (\mathcal{G}_T(j))^2\right)} + \epsilon_s \le (\mathcal{H} + \epsilon)\sqrt{1 - \beta_2^s}, \quad \forall s \in [T].$$

Combining with the parameter setting in (10),

$$\frac{\eta_s}{\|\boldsymbol{a}_s\|_\infty} \geq \frac{\eta\sqrt{1-\beta_2^s}}{(1-\beta_1^s)\|\boldsymbol{a}_s\|_\infty} \geq \frac{\tilde{C}_0\sqrt{1-\beta_2}}{\mathcal{H}+\epsilon_0\sqrt{1-\beta_2}}.$$

We then combine with (123) and $\mathcal{H}$ in (11) to obtain that with probability at least $1-2\delta$,

$$\frac{1}{T}\sum_{s=1}^{T}\|\bar{\boldsymbol{g}}_s\|^2 \leq \frac{4\hat{H}}{T\tilde{C}_0}\left(\frac{\sqrt{2(\sigma_0^2+\sigma_1^2 H^p+H^2)}}{\sqrt{1-\beta_2}}+\epsilon_0\right)\sqrt{\log\left(\frac{\mathrm{e}T}{\delta}\right)}.$$

Finally, following the same deduction in (63) and (64), we could derive that $\hat{H} \sim \mathcal{O}\left(\log^2\left(\frac{T}{\epsilon_0\delta}\right)\right)$ from (114) and the desired results in Theorem 4.1. □

# E  Convergence of Adam without corrective terms and RMSProp under generalized smoothness

In this section, we will provide the detailed result and proof for the convergence of Algorithm 2 under generalized smoothness. First, we will provide the formal version of Theorem 4.2.

**Theorem E.1.** *Let $T \geq 1$ and $\delta \in (0, 1/2)$. Suppose that $\{\boldsymbol{x}_s\}_{s\in[T]}$ is a sequence generated by Algorithm 2, $f$ is $(L_0, L_q)$-smooth satisfying (8), Assumptions (A1)-(A3) hold, and*

$$0 \leq \beta_1 < \beta_2 < 1, \quad \beta_2 = 1-c/T, \quad \epsilon = \epsilon_0\sqrt{1-\beta_2}, \quad \eta = \tilde{C}_0\sqrt{1-\beta_2},$$

$$\tilde{C}_0 \leq \min\left\{E_0, \frac{E_0}{\tilde{\mathcal{H}}}, \frac{E_0}{\tilde{\mathcal{L}}}, \sqrt{\frac{\beta_2(1-\beta_1)^2(1-\beta_1/\beta_2)}{4L_q^2 d}}\right\}, \tag{124}$$

*where $c, \epsilon_0, E_0, \tilde{C}_0 > 0$ are constants, $\bar{H}$ is controlled by $\mathcal{O}\left(\log\left(\frac{T}{\epsilon_0\delta}\right)\right)$ [7], and $\tilde{H}, \tilde{\mathcal{H}}, \tilde{\mathcal{L}}$ are defined as*

$$\tilde{H} := L_0/L_q + \left(4L_q\bar{H}\right)^q + \left(4L_q\bar{H}\right)^{\frac{q}{2-q}} + \left(4L_0\bar{H}\right)^{\frac{q}{2}} + 4L_q\bar{H} + \left(4L_q\bar{H}\right)^{\frac{1}{2-q}} + \sqrt{4L_0\bar{H}},$$

$$\tilde{\mathcal{H}} := \sqrt{2(\sigma_0^2+\sigma_1^2\tilde{H}^p+\tilde{H}^2)\log\left(\frac{\mathrm{e}T}{\delta}\right)}, \quad \tilde{\mathcal{L}} := L_0 + L_q\left(\tilde{H}^q + \tilde{H} + \frac{L_0}{L_q}\right)^q. \tag{125}$$

*Then, it holds that with probability at least $1-2\delta$,*

$$\frac{1}{T}\sum_{s=1}^{T}\|\nabla f(\boldsymbol{x}_s)\|^2 \leq \tilde{\mathcal{O}}\left\{\frac{\bar{H}^2(1+\sigma_1^2)+\bar{H}\left(\tilde{H}+\epsilon_0\right)}{T}+\frac{\bar{H}\sigma_0}{\sqrt{T}}\right\}. \tag{126}$$

*Remark* E.2. (1). The convergence rate in Theorem C.2 is of order $\tilde{\mathcal{O}}\left(1/T+\sigma_0/\sqrt{T}\right)$, which can be accelerated to $\tilde{\mathcal{O}}(1/T)$ rate when $\sigma_0$ is sufficiently low.
(2). Setting $\beta_1 = 0$, Theorem E.1 then shows that RMSProp with the hyper-parameter setup in (124) can also find a stationary point with the convergence rate of $\tilde{\mathcal{O}}\left(1/T+\sigma_0/\sqrt{T}\right)$ order.

## E.1  Proof detail

To start with, we follow all the notations defined in Appendix C and Appendix D, particularly $G_s, \mathcal{G}_T(s)$ in (31) and $\mathcal{L}_s^{(x)}, \mathcal{L}_s^{(y)}$ in (97). Then, we have the following claims.

**Proposition E.3.** *Setting $\eta_s = \eta$ and $\eta = \tilde{C}_0\sqrt{1-\beta_2}$, the results in Lemma D.1, Lemma D.2 and Lemma D.3 remain unchanged.*

*Proof.* Using $\eta_s = \eta$ and following the proof for the above lemmas, it's easy to verify that these lemmas still hold. □

---

[7]The specific definition of $\bar{H}$ can be found in (138).

Based on these results, we are able to show that the gradient norm is bounded along the training process generated by Algorithm 2 under generalized smoothness.

**Proposition E.4.** *Under the same conditions in Theorem C.2, for any given $\delta \in (0, 1/2)$, it holds that with probability at least $1 - 2\delta$,*

$$\|\bar{\boldsymbol{g}}_t\| \le \tilde{H}, \quad \mathcal{G}_T(t) \le \tilde{\mathcal{H}}, \quad \mathcal{L}_t^{(x)} \le \mathcal{L}_t^{(y)} \le \tilde{\mathcal{L}}, \quad \forall t \in [T+1], \tag{127}$$

*and*

$$\Delta_{t+1} \le -\frac{\eta}{4} \sum_{s=1}^{t} \left\| \frac{\bar{\boldsymbol{g}}_s}{\sqrt{\tilde{\boldsymbol{a}}_s}} \right\|^2 + \bar{H}, \quad \forall t \in [T], \tag{128}$$

*Proof.* To start with, we shall verify that (99) always holds. Recalling $\eta$ in (124) and $F$ in Lemma D.1,

$$\eta F = \tilde{C}_0 \sqrt{1 - \beta_2} F \le \sqrt{\frac{\beta_2 (1 - \beta_1)^2 (1 - \beta_2)(1 - \beta_1/\beta_2)}{4 L_q^2 d}} \cdot F \le \frac{1}{L_q}.$$

Hence, we make sure that the distance requirement in (8) always holds according to Lemma D.1. Then, using the descent lemma in (101) and (68), it leads to for any $t \ge 1$,

$$f(\boldsymbol{y}_{t+1}) \le f(\boldsymbol{x}_1) + \underbrace{\sum_{s=1}^{t} -\eta \left\langle \nabla f(\boldsymbol{y}_s), \frac{\boldsymbol{g}_s}{\tilde{\boldsymbol{b}}_s} \right\rangle}_{\tilde{\mathbf{A}}} + \underbrace{\frac{\beta_1}{1 - \beta_1} \sum_{s=1}^{t} \left\langle \tilde{\Delta}_s \odot (\boldsymbol{x}_s - \boldsymbol{x}_{s-1}), \nabla f(\boldsymbol{y}_s) \right\rangle}_{\tilde{\mathbf{B}}}$$

$$+ \underbrace{\frac{\mathcal{L}_s^{(y)}}{2} \sum_{s=1}^{t} \left\| \eta \cdot \frac{\boldsymbol{g}_s}{\tilde{\boldsymbol{b}}_s} - \frac{\beta_1}{1 - \beta_1} (\tilde{\Delta}_s \odot (\boldsymbol{x}_s - \boldsymbol{x}_{s-1})) \right\|^2}_{\tilde{\mathbf{C}}'}, \tag{129}$$

The following estimation is based on the probability event in Lemma B.6.

**Bounding $\tilde{\mathbf{A}}$.** We first have the same decomposition as in (72). Then, setting $\lambda = \sqrt{1 - \beta_2}/(3\eta\tilde{\mathcal{H}})$ in (74), we obtain that with probability at least $1 - \delta$,

$$\tilde{\mathbf{A}}.\mathbf{1}.\mathbf{1} \le \frac{\eta \mathcal{G}_T(t)}{4\tilde{\mathcal{H}}} \sum_{s=1}^{t} \left\| \frac{\bar{\boldsymbol{g}}_s}{\sqrt{\tilde{\boldsymbol{a}}_s}} \right\|^2 + \frac{3\eta\tilde{\mathcal{H}}}{\sqrt{1 - \beta_2}} \log \left( \frac{T}{\delta} \right), \forall t \in [T]. \tag{130}$$

Using Lemma C.5 and following the similar deduction for bounding $(\tilde{\mathbf{A}}.\mathbf{1}.\mathbf{2})$ in (76), we have

$$\tilde{\mathbf{A}}.\mathbf{1}.\mathbf{2} \le \frac{\eta}{4} \sum_{s=1}^{t} \left\| \frac{\bar{\boldsymbol{g}}_s}{\sqrt{\tilde{\boldsymbol{a}}_s}} \right\|^2 + \eta\sqrt{1 - \beta_2} \mathcal{G}_T(t) \sum_{s=1}^{t} \left\| \frac{\boldsymbol{g}_s}{\tilde{\boldsymbol{b}}_s} \right\|^2. \tag{131}$$

Combining with (130) and (131), we obtain that

$$\tilde{\mathbf{A}}.\mathbf{1} \le \left( \frac{\mathcal{G}_T(t)}{4\tilde{\mathcal{H}}} - \frac{3}{4} \right) \sum_{s=1}^{t} \eta \left\| \frac{\bar{\boldsymbol{g}}_s}{\sqrt{\tilde{\boldsymbol{a}}_s}} \right\|^2 + \frac{3\eta\tilde{\mathcal{H}}}{\sqrt{1 - \beta_2}} \log \left( \frac{T}{\delta} \right) + \eta\sqrt{1 - \beta_2} \mathcal{G}_T(t) \sum_{s=1}^{t} \left\| \frac{\boldsymbol{g}_s}{\tilde{\boldsymbol{b}}_s} \right\|^2. \tag{132}$$

The estimation for $\tilde{\mathbf{A}}.\mathbf{2}$ is similar to (108) and (78). With $L$ replaced by $\mathcal{L}_s^{(x)}$ in (78),

$$\tilde{\mathbf{A}}.\mathbf{2} \le \sum_{s=1}^{t} \frac{\mathcal{L}_s^{(x)} \eta^2}{2(1 - \beta_1)^2} \left\| \frac{\boldsymbol{m}_{s-1}}{\tilde{\boldsymbol{b}}_{s-1}} \right\|^2 + \sum_{s=1}^{t} \frac{\mathcal{L}_s^{(x)} \eta^2}{2} \left\| \frac{\boldsymbol{g}_s}{\tilde{\boldsymbol{b}}_s} \right\|^2. \tag{133}$$

**Bounding $\tilde{\mathbf{B}}, \tilde{\mathbf{C}}'$.** Following the same decomposition in (79) and (80), we first note that the estimation for $\tilde{\mathbf{B}}.\mathbf{1}$ in (83) remains unchangedL

$$\tilde{\mathbf{B}}.\mathbf{1} \le \frac{\eta}{4} \sum_{s=1}^{t} \left\| \frac{\bar{\boldsymbol{g}}_s}{\sqrt{\tilde{\boldsymbol{a}}_s}} \right\|^2 + \frac{2\eta \mathcal{G}_T(t)\sqrt{1-\beta_2}}{(1-\beta_1)^2} \sum_{s=1}^{t} \left( \left\| \frac{\boldsymbol{m}_{s-1}}{\tilde{\boldsymbol{b}}_s} \right\|^2 + \left\| \frac{\boldsymbol{m}_{s-1}}{\tilde{\boldsymbol{b}}_{s-1}} \right\|^2 \right). \tag{134}$$

Following the similar deduction in (84) and using Lemma B.1, we can replace $L$ with $\mathcal{L}_s^{(x)}$ to obtain that

$$\tilde{\mathbf{B}}.\mathbf{2} \le \sum_{s=1}^{t} \frac{\Sigma_{\max}^2 \mathcal{L}_s^{(x)} \eta^2}{(1-\beta_1)^2} \left\| \frac{\hat{\boldsymbol{m}}_{s-1}}{\boldsymbol{b}_{s-1}} \right\|^2 \le \sum_{s=1}^{t} \frac{\mathcal{L}_s^{(x)} \eta^2}{\beta_2 (1-\beta_1)^2} \left\| \frac{\hat{\boldsymbol{m}}_{s-1}}{\tilde{\boldsymbol{b}}_{s-1}} \right\|^2. \tag{135}$$

Following the similar deduction in (110), we indeed replace $L$ with $\mathcal{L}_s^{(y)}$ in (85)

$$\tilde{\mathbf{C}}' \le \eta^2 \sum_{s=1}^{t} \mathcal{L}_s^{(y)} \left\| \frac{\boldsymbol{g}_s}{\tilde{\boldsymbol{b}}_s} \right\|^2 + \frac{\eta^2}{\beta_2 (1-\beta_1)^2} \sum_{s=1}^{t} \mathcal{L}_s^{(y)} \left\| \frac{\boldsymbol{m}_{s-1}}{\tilde{\boldsymbol{b}}_{s-1}} \right\|^2. \tag{136}$$

**Putting together.** Noting that the above results are established based on probability events in Lemma B.6 and (130). Hence, plugging (132), (133), (134), (135) and (136) into (129) and using $\beta_2 < 1$ and $\mathcal{L}_s^{(x)} \le \mathcal{L}_s^{(y)}$ from (97), it holds that with probability at least $1 - 2\delta$, for all $t \in [T]$,

$$\begin{aligned} f(\boldsymbol{y}_{t+1}) &\le f(\boldsymbol{x}_1) + \eta \left( \frac{\mathcal{G}_T(t)}{4\tilde{\mathcal{H}}} - \frac{1}{2} \right) \sum_{s=1}^{t} \left\| \frac{\bar{\boldsymbol{g}}_s}{\sqrt{\tilde{\boldsymbol{a}}_s}} \right\|^2 + \frac{3\eta \tilde{\mathcal{H}}}{\sqrt{1-\beta_2}} \log \left( \frac{T}{\delta} \right) \\ &+ \sum_{s=1}^{t} \left( \frac{2\eta\sqrt{1-\beta_2}\mathcal{G}_T(t)}{(1-\beta_1)^2} + \frac{5\mathcal{L}_s^{(y)} \eta^2}{2\beta_2 (1-\beta_1)^2} \right) \left\| \frac{\boldsymbol{m}_{s-1}}{\tilde{\boldsymbol{b}}_{s-1}} \right\|^2 \\ &+ \frac{2\eta\sqrt{1-\beta_2}\mathcal{G}_T(t)}{(1-\beta_1)^2} \sum_{s=1}^{t} \left\| \frac{\boldsymbol{m}_{s-1}}{\tilde{\boldsymbol{b}}_s} \right\|^2 + \sum_{s=1}^{t} \left( \eta\sqrt{1-\beta_2}\mathcal{G}_T(t) + \frac{3\mathcal{L}_s^{(y)} \eta^2}{2} \right) \left\| \frac{\boldsymbol{g}_s}{\tilde{\boldsymbol{b}}_s} \right\|^2. \end{aligned} \tag{137}$$

**An induction argument.** We first provide the detailed expression of $\bar{H}$:

$$\begin{aligned} \bar{H} &:= f(\boldsymbol{x}_1) - f^* + 3E_0 \log \left( \frac{T}{\delta} \right) \\ &+ \left( \frac{2E_0}{(1-\beta_1)(1-\beta_1/\beta_2)} + \frac{5E_0^2}{2\beta_2 (1-\beta_1)(1-\beta_1/\beta_2)} \right) d \log \left( \frac{\mathcal{I}(T)}{\beta_2^T} \right) \\ &+ \frac{2E_0 d}{\beta_2 (1-\beta_1)(1-\beta_1/\beta_2)} \log \left( \frac{\mathcal{I}(T)}{\beta_2^T} \right) + \left( E_0 + \frac{3E_0^2}{2} \right) \frac{d}{1-\beta_1/\beta_2} \log \left( \frac{\mathcal{I}(T)}{\beta_2^T} \right) \\ \mathcal{I}(t) &= 1 + \frac{2\mathcal{M}_T^2}{\epsilon^2} \left[ \sigma_0^2 t + \sigma_1^2 t \left( \|\bar{\boldsymbol{g}}_1\| + t\tilde{M}' \right)^p + t \left( \|\bar{\boldsymbol{g}}_1\| + t\tilde{M}' \right)^2 \right], \quad \tilde{M}' = E_0 \sqrt{\frac{d}{1-\beta_1/\beta_2}}. \end{aligned} \tag{138}$$

The induction then begins by noting that from Lemma D.3 and $\tilde{H}$ in (125),

$$G_1 = \|\bar{\boldsymbol{g}}_1\| \le 4L_q(f(\boldsymbol{x}_1) - f^*) + (4L_q(f(\boldsymbol{x}_1) - f^*))^{\frac{1}{2-q}} + \sqrt{4L_0(f(\boldsymbol{x}_1) - f^*)} \le \tilde{H}.$$

Suppose that for some $t \in [T]$,

$$G_s \le \tilde{H}, \quad \forall s \in [t]. \tag{139}$$

Consequently, recalling $\mathcal{G}_T(s)$ in (19), $\mathcal{L}_s^{(x)}, \mathcal{L}_s^{(y)}$ in (97) and $\mathcal{H}, \mathcal{L}$ in (11),

$$\mathcal{G}_T(s) \le \tilde{\mathcal{H}}, \quad \mathcal{L}_s^{(x)} \le \mathcal{L}_s^{(y)} \le \tilde{\mathcal{L}}, \quad \forall s \in [t]. \tag{140}$$

Further recalling the setting of $\tilde{C}_0$ in (10), with a simple calculation it holds that,

$$\tilde{C}_0 H \le \tilde{C}_0 \tilde{\mathcal{H}} \le E_0, \quad \tilde{C}_0 \tilde{\mathcal{L}} \le E_0, \quad \tilde{C}_0^2 \tilde{\mathcal{L}} \le E_0^2, \quad \tilde{C}_0 \epsilon_0 \le E_0 \epsilon_0. \tag{141}$$

Using (140), (141), $\mathcal{J}(t)$, $\tilde{M}_t'$ defined in Lemma D.5 and $\mathcal{I}(t)$, $\tilde{M}'$ defined in (138),

$$\tilde{M}_t \leq \tilde{M}', \quad \log\left(\frac{\mathcal{F}_i(t)}{\beta_2^t}\right) \leq \log\left(\frac{\mathcal{J}(t)}{\beta_2^t}\right) \leq \log\left(\frac{\tilde{\mathcal{I}}(T)}{\beta_2^T}\right), \quad \forall t \leq T, i \in [d]$$

Using Lemma B.3, Lemma B.10 and Lemma D.5, we have

$$\sum_{s=1}^t \left\|\frac{\boldsymbol{m}_{s-1}}{\tilde{\boldsymbol{b}}_{s-1}}\right\|^2 \leq \frac{d(1-\beta_1)}{(1-\beta_2)(1-\beta_1/\beta_2)} \log\left(\frac{\mathcal{I}(T)}{\beta_2^T}\right),$$

$$\sum_{s=1}^t \left\|\frac{\boldsymbol{m}_{s-1}}{\tilde{\boldsymbol{b}}_s}\right\|^2 \leq \frac{d(1-\beta_1)}{\beta_2(1-\beta_2)(1-\beta_1/\beta_2)} \log\left(\frac{\mathcal{I}(T)}{\beta_2^T}\right),$$

$$\sum_{s=1}^t \left\|\frac{\boldsymbol{g}_s}{\tilde{\boldsymbol{b}}_s}\right\|^2 \leq \frac{d}{(1-\beta_2)(1-\beta_1/\beta_2)} \log\left(\frac{\mathcal{I}(T)}{\beta_2^T}\right). \tag{142}$$

Then, plugging (140) and (142) into (137), and using $\eta = \tilde{C}_0\sqrt{1-\beta_2}$ and (141),

$$\Delta_{t+1} = f(\boldsymbol{y}_{t+1}) - f^* \leq \Delta_1 - \frac{\eta}{4}\sum_{s=1}^t \left\|\frac{\bar{\boldsymbol{g}}_s}{\sqrt{\tilde{\boldsymbol{a}}_s}}\right\|^2 + 3E_0\log\left(\frac{T}{\delta}\right)$$

$$+ \left(\frac{2E_0}{(1-\beta_1)(1-\beta_1/\beta_2)} + \frac{5E_0^2}{2\beta_2(1-\beta_1)(1-\beta_1/\beta_2)}\right)d\log\left(\frac{\mathcal{I}(T)}{\beta_2^T}\right)$$

$$+ \frac{2E_0 d}{\beta_2(1-\beta_1)(1-\beta_1/\beta_2)}\log\left(\frac{\mathcal{I}(T)}{\beta_2^T}\right) + \left(E_0 + \frac{3E_0^2}{2}\right)\frac{d}{1-\beta_1/\beta_2}\log\left(\frac{\mathcal{I}(T)}{\beta_2^T}\right) \leq \bar{H}, \tag{143}$$

where we use $\bar{H}$ defined in (138). Further using Lemma D.2, Lemma D.3 and $\tilde{H}$ in (125),

$$\|\bar{\boldsymbol{g}}_{t+1}\| \leq L_0/L_q + (4L_q\Delta_{t+1})^q + (4L_q\Delta_{t+1})^{\frac{q}{2-q}}$$

$$+ (4L_0\Delta_{t+1})^{\frac{q}{2}} + 4L_q\Delta_{t+1} + (4L_q\Delta_{t+1})^{\frac{1}{2-q}} + \sqrt{4L_0\Delta_{t+1}} \leq \tilde{H}.$$

We then deduce that $G_{t+1} = \max\{G_t, \|\bar{\boldsymbol{g}}_{t+1}\|\} \leq \tilde{H}$. The induction is then complete and we obtain the desired result in (127). Finally, as an intermediate result of the proof, we obtain that (143) holds as well. $\qquad\square$

## E.2 Proof of the main result

Now we are ready to prove the main convergence result.

*Proof of Theorem C.2.* In what follows, we will assume that Proposition E.4 holds. Hence, the final convergence bound also holds with probability at least $1 - 2\delta$. We set $t = T$ in (127),

$$\frac{\eta}{4}\sum_{s=1}^T \frac{\|\bar{\boldsymbol{g}}_s\|^2}{\|\tilde{\boldsymbol{a}}_s\|_\infty} \leq \frac{\eta}{4}\sum_{s=1}^T \left\|\frac{\bar{\boldsymbol{g}}_s}{\sqrt{\tilde{\boldsymbol{a}}_s}}\right\|^2 \leq \bar{H} - \Delta_{t+1} \leq \bar{H}. \tag{144}$$

Following the similar deduction in (95), with $\eta = \tilde{C}_0\sqrt{1-\beta_2}, \epsilon = \epsilon_0\sqrt{1-\beta_2}$, we have for any $s \in [T]$,

$$\frac{\|\tilde{\boldsymbol{a}}_s\|_\infty}{\eta} \leq \frac{\mathcal{M}_T}{C_0}\sqrt{2(1+\sigma_1^2)\sum_{j=1}^T \|\bar{\boldsymbol{g}}_j\|^2} + \frac{\mathcal{M}_T\sigma_0\sqrt{2T} + \tilde{\mathcal{H}} + \epsilon_0}{C_0}. \tag{145}$$

We therefore combine with (144) and (145) to obtain that

$$\sum_{s=1}^T \|\bar{\boldsymbol{g}}_s\|^2 \leq \frac{4\bar{H}}{\tilde{C}_0}\left(\mathcal{M}_T\sqrt{2(1+\sigma_1^2)\sum_{j=1}^T \|\bar{\boldsymbol{g}}_j\|^2} + \mathcal{M}_T\sigma_0\sqrt{2T} + \tilde{\mathcal{H}} + \epsilon_0\right)$$

$$\leq \frac{1}{2}\sum_{s=1}^T \|\bar{\boldsymbol{g}}_s\|^2 + \frac{8\bar{H}^2\mathcal{M}_T(1+\sigma_1^2)}{\tilde{C}_0^2} + \frac{4\bar{H}}{\tilde{C}_0}\left(\mathcal{M}_T\sigma_0\sqrt{2T} + \tilde{\mathcal{H}} + \epsilon_0\right), \tag{146}$$

which leads to

$$\frac{1}{T}\sum_{s=1}^{T}\|\bar{\boldsymbol{g}}_s\|^2 \leq \frac{1}{\tilde{C}_0 T}\left(16\bar{H}^2\mathcal{M}_T(1+\sigma_1^2)+8\bar{H}(\tilde{\mathcal{H}}+\epsilon_0)\right)+\frac{8\sqrt{2}\bar{H}\mathcal{M}_T\sigma_0}{\tilde{C}_0\sqrt{T}}. \tag{147}$$

Following the result in (63) and (64), we show that when $\beta_2 = 1 - c/T$, $\bar{H}, \tilde{\mathcal{H}} \sim \mathcal{O}\left(\text{poly}(\log T)\right)$ with respect to $T$. Finally, using the convergence result in (147), we obtain the desired result. $\quad\square$

# F  Omitted proof in Appendix B

## F.1  Omitted proof in Appendix B.1

*Proof of Lemma B.1.* We fix arbitrary $i \in [d]$ and have the following two cases. When $\frac{\eta_s b_{s-1,i}}{\eta_{s-1}b_{s,i}} < 1$, we have

$$\left|\frac{\eta_s b_{s-1,i}}{\eta_{s-1}b_{s,i}}-1\right| = 1 - \frac{\eta_s b_{s-1,i}}{\eta_{s-1}b_{s,i}} < 1.$$

When $\frac{\eta_s b_{s-1,i}}{\eta_{s-1}b_{s,i}} \geq 1$, let $r = \beta_2^{s-1}$. Since $0 < 1 - \beta_1^{s-1} < 1 - \beta_1^s, \forall s \geq 2$, then we have

$$\frac{\eta_s}{\eta_{s-1}} = \sqrt{\frac{1-\beta_2^s}{1-\beta_2^{s-1}}\cdot\frac{1-\beta_1^{s-1}}{1-\beta_1^s}} \leq \sqrt{1+\frac{\beta_2^{s-1}(1-\beta_2)}{1-\beta_2^{s-1}}} = \sqrt{1+(1-\beta_2)\cdot\frac{r}{1-r}}.$$

Since $h(r) = r/(1-r)$ is increasing as $r$ grows and $r$ takes the maximum value when $s = 2$. Hence, it holds that

$$\frac{\eta_s}{\eta_{s-1}} \leq \sqrt{1+(1-\beta_2)\cdot\frac{\beta_2}{1-\beta_2}} = \sqrt{1+\beta_2}. \tag{148}$$

Then, since $\epsilon_{s-1} \leq \epsilon_s$, we further have

$$\frac{b_{s-1,i}}{b_{s,i}} = \frac{\epsilon_{s-1}+\sqrt{v_{s-1,i}}}{\epsilon_s+\sqrt{\beta_2 v_{s-1,i}+(1-\beta_2)g_{s,i}^2}} \leq \frac{\epsilon_s+\sqrt{v_{s-1,i}}}{\epsilon_s+\sqrt{\beta_2 v_{s-1,i}}} \leq \frac{1}{\sqrt{\beta_2}}. \tag{149}$$

Combining with (148) and (149), we have

$$\left|\frac{\eta_s b_{s-1,i}}{\eta_{s-1}b_{s,i}}-1\right| = \frac{\eta_s b_{s-1,i}}{\eta_{s-1}b_{s,i}}-1 \leq \sqrt{\frac{1+\beta_2}{\beta_2}}-1.$$

Combining the two cases and noting that the bound holds for any $i \in [d]$, we then obtain the desired result. $\quad\square$

*Proof of Lemma B.2.* Denoting $\tilde{M} = \sum_{j=1}^{s-1}\beta_1^{s-1-j}$ and applying (28) with $\hat{M}$ and $\alpha_j$ replaced by $\tilde{M}$ and $g_{j,i}$ respectively,

$$\left(\sum_{j=1}^{s-1}\beta_1^{s-1-j}g_{j,i}\right)^2 \leq \tilde{M}\cdot\sum_{j=1}^{s-1}\beta_1^{s-1-j}g_{j,i}^2. \tag{150}$$

Hence, combining with the definition of $b_{s,i}$ in (16), we further have for any $i \in [d]$ and $s \geq 2$,

$$\left|\frac{m_{s-1,i}}{b_{s-1,i}}\right| \leq \left|\frac{m_{s-1,i}}{\sqrt{v_{s-1,i}}}\right| = \sqrt{\frac{(1-\beta_1)^2\left(\sum_{j=1}^{s-1}\beta_1^{s-1-j}g_{j,i}\right)^2}{(1-\beta_2)\sum_{j=1}^{s-1}\beta_2^{s-1-j}g_{j,i}^2}}$$

$$\leq \frac{1-\beta_1}{\sqrt{1-\beta_2}}\sqrt{\tilde{M}\cdot\frac{\sum_{j=1}^{s-1}\beta_1^{s-1-j}g_{j,i}^2}{\sum_{j=1}^{s-1}\beta_2^{s-1-j}g_{j,i}^2}} \leq \frac{1-\beta_1}{\sqrt{1-\beta_2}}\sqrt{\tilde{M}\cdot\sum_{j=1}^{s-1}\left(\frac{\beta_1}{\beta_2}\right)^{s-1-j}}$$

$$= \frac{1-\beta_1}{\sqrt{1-\beta_2}}\sqrt{\frac{1-\beta_1^{s-1}}{1-\beta_1}\cdot\frac{1-(\beta_1/\beta_2)^{s-1}}{1-\beta_1/\beta_2}} \leq \sqrt{\frac{(1-\beta_1)(1-\beta_1^{s-1})}{(1-\beta_2)(1-\beta_1/\beta_2)}},$$

where the last inequality applies $\beta_1 < \beta_2$. We thus prove the first result. To prove the second result, from the smoothness of $f$,

$$\|\bar{g}_s\| \le \|\bar{g}_{s-1}\| + \|\bar{g}_s - \bar{g}_{s-1}\| \le \|\bar{g}_{s-1}\| + L\|x_s - x_{s-1}\|. \tag{151}$$

Combining with (30) and $\eta = C_0\sqrt{1-\beta_2}$,

$$\|x_s - x_{s-1}\|_\infty \le \eta_{s-1}\left\|\frac{m_{s-1}}{b_{s-1}}\right\|_\infty \le \eta\sqrt{\frac{1}{(1-\beta_2)(1-\beta_1/\beta_2)}} = C_0\sqrt{\frac{1}{1-\beta_1/\beta_2}}. \tag{152}$$

Using $\|x_s - x_{s-1}\| \le \sqrt{d}\|x_s - x_{s-1}\|_\infty$ and (151),

$$\|\bar{g}_s\| \le \|\bar{g}_{s-1}\| + LC_0\sqrt{\frac{d}{1-\beta_1/\beta_2}} \le \|\bar{g}_1\| + LC_0 s\sqrt{\frac{d}{1-\beta_1/\beta_2}}.$$

$\square$

*Proof of Lemma B.3.* Recalling the updated rule and the definition of $b_{s,i}$ in (16), using $\epsilon_s^2 = \epsilon^2(1 - \beta_2^s) \ge \epsilon^2(1-\beta_2)$,

$$b_{s,i}^2 \ge v_{s,i}^2 + \epsilon_s^2 \ge (1-\beta_2)\left(\sum_{j=1}^s \beta_2^{s-j}g_{j,i}^2 + \epsilon^2\right), \quad \text{and} \quad m_{s,i} = (1-\beta_1)\sum_{j=1}^s \beta_1^{s-j}g_{j,i}. \tag{153}$$

**Proof for the first summation.** Using (153), for any $i \in [d]$,

$$\sum_{s=1}^t \frac{g_{s,i}^2}{b_{s,i}^2} \le \frac{1}{1-\beta_2}\sum_{s=1}^t \frac{g_{s,i}^2}{\epsilon^2 + \sum_{j=1}^s \beta_2^{s-j}g_{j,i}^2}.$$

Applying Lemma A.1 and recalling the definition of $\mathcal{F}_i(t)$,

$$\sum_{s=1}^t \frac{g_{s,i}^2}{b_{s,i}^2} \le \frac{1}{1-\beta_2}\left[\log\left(1 + \frac{1}{\epsilon^2}\sum_{s=1}^t \beta_2^{t-s}g_{s,i}^2\right) - t\log\beta_2\right] \le \frac{1}{1-\beta_2}\log\left(\frac{\mathcal{F}_i(t)}{\beta_2^t}\right).$$

Summing over $i \in [d]$, we obtain the first desired result.

**Proof for the second summation.** Following from (153),

$$\sum_{s=1}^t \frac{m_{s,i}^2}{b_{s,i}^2} \le \frac{(1-\beta_1)^2}{1-\beta_2} \cdot \sum_{s=1}^t \frac{\left(\sum_{j=1}^s \beta_1^{s-j}g_{j,i}\right)^2}{\epsilon^2 + \sum_{j=1}^s \beta_2^{s-j}g_{j,i}^2}.$$

Applying Lemma A.2 and $\beta_2 \le 1$,

$$\sum_{s=1}^t \frac{m_{s,i}^2}{b_{s,i}^2} \le \frac{(1-\beta_1)^2}{1-\beta_2} \cdot \frac{1}{(1-\beta_1)(1-\beta_1/\beta_2)}\left[\log\left(1 + \frac{1}{\epsilon^2}\sum_{s=1}^t \beta_2^{t-s}g_{s,i}^2\right) - t\log\beta_2\right]$$

$$= \frac{1-\beta_1}{(1-\beta_2)(1-\beta_1/\beta_2)}\log\left(\frac{\mathcal{F}_i(t)}{\beta_2^t}\right).$$

Summing over $i \in [d]$, we obtain the second desired result.

**Proof for the third summation.** Following from (153),

$$\sum_{s=1}^t \frac{m_{s,i}^2}{b_{s+1,i}^2} \le \sum_{s=1}^t \frac{\left[(1-\beta_1)\sum_{j=1}^s \beta_1^{s-j}g_{j,i}\right]^2}{\epsilon^2(1-\beta_2) + (1-\beta_2)\sum_{j=1}^{s+1}\beta_2^{s+1-j}g_{j,i}^2}$$

$$\le \sum_{s=1}^t \frac{(1-\beta_1)^2\left(\sum_{j=1}^s \beta_1^{s-j}g_{j,i}\right)^2}{\epsilon^2(1-\beta_2) + (1-\beta_2)\beta_2\sum_{j=1}^s \beta_2^{s-j}g_{j,i}^2}.$$

Applying Lemma A.2, and using $\beta_2 \leq 1$,

$$
\begin{aligned}
\sum_{s=1}^{t} \frac{m_{s,i}^2}{b_{s+1,i}^2} &\leq \frac{(1-\beta_1)^2}{(1-\beta_2)\beta_2} \cdot \sum_{s=1}^{t} \frac{\left(\sum_{j=1}^{s} \beta_1^{s-j} g_{j,i}\right)^2}{\frac{\epsilon^2}{\beta_2} + \sum_{j=1}^{s} \beta_2^{s-j} g_{j,i}^2} \\
&\leq \frac{(1-\beta_1)^2}{(1-\beta_2)\beta_2} \cdot \frac{1}{(1-\beta_1)(1-\beta_1/\beta_2)} \left[\log\left(1 + \frac{\beta_2}{\epsilon^2}\sum_{s=1}^{t} \beta_2^{t-s} g_{s,i}^2\right) - t\log\beta_2\right] \\
&\leq \frac{1-\beta_1}{\beta_2(1-\beta_2)(1-\beta_1/\beta_2)} \log\left(\frac{\mathcal{F}_i(t)}{\beta_2^t}\right).
\end{aligned}
$$

Summing over $i \in [d]$, we obtain the third desired result.

**Proof for the fourth summation.** Following the definition of $\hat{m}_{s,i}$ from (29), and combining with (153),

$$
\sum_{s=1}^{t} \frac{\hat{m}_{s,i}^2}{b_{s,i}^2} \leq \frac{(1-\beta_1)^2}{1-\beta_2} \cdot \sum_{s=1}^{t} \frac{\left(\frac{1}{1-\beta_1^s}\sum_{j=1}^{s} \beta_1^{s-j} g_{j,i}\right)^2}{\epsilon^2 + \sum_{j=1}^{s} \beta_2^{s-j} g_{j,i}^2}.
$$

Applying Lemma A.2 and using $\beta_2 \leq 1$,

$$
\begin{aligned}
\sum_{s=1}^{t} \frac{\hat{m}_{s,i}^2}{b_{s,i}^2} &\leq \frac{(1-\beta_1)^2}{1-\beta_2} \cdot \frac{1}{(1-\beta_1)^2(1-\beta_1/\beta_2)} \left[\log\left(1 + \frac{1}{\epsilon^2}\sum_{s=1}^{t} \beta_2^{t-s} g_{s,i}^2\right) - t\log\beta_2\right] \\
&\leq \frac{1}{(1-\beta_2)(1-\beta_1/\beta_2)} \log\left(\frac{\mathcal{F}_i(t)}{\beta_2^t}\right).
\end{aligned}
$$

Summing over $i \in [d]$, we obtain the fourth desired result. $\qquad\square$

*Proof of Lemma B.4.* Let $\hat{x} = x - \frac{1}{L}\nabla f(x)$. Then using the descent lemma of smoothness,

$$
f(\hat{x}) \leq f(x) + \langle \nabla f(x), \hat{x} - x\rangle + \frac{L}{2}\|\hat{x} - x\|^2 \leq f(x) - \frac{1}{2L}\|\nabla f(x)\|^2.
$$

Re-arranging the order, and noting that $f(\hat{x}) \geq f^*$,

$$
\|\nabla f(x)\|^2 \leq 2L(f(x) - f(\hat{x})) \leq 2L(f(x) - f^*).
$$

$\qquad\square$

*Proof of Lemma B.5.* Applying the norm inequality and the smoothness of $f$,

$$
\|\nabla f(x_s)\| \leq \|\nabla f(y_s)\| + \|\nabla f(x_s) - \nabla f(y_s)\| \leq \|\nabla f(y_s)\| + L\|y_s - x_s\|.
$$

Combining with the definition of $y_s$ in (17) and (152), and using $\beta_1 \in [0, 1)$, we obtain the desired result that

$$
\|\nabla f(x_s)\| \leq \|\nabla f(y_s)\| + \frac{L\beta_1}{1-\beta_1}\|x_s - x_{s-1}\| \leq \|\nabla f(y_s)\| + \frac{LC_0\sqrt{d}}{(1-\beta_1)\sqrt{1-\beta_1/\beta_2}}.
$$

$\qquad\square$

### F.2 Omitted proof in Appendix B.3

*Proof of Lemma B.6.* Let us denote $\gamma_s = \frac{\|\boldsymbol{\xi}_s\|^2}{\sigma_0^2 + \sigma_1^2 \|\bar{g}_s\|^p}, \forall s \in [T]$. Then from Assumption (A3), we first have $\mathbb{E}_{\boldsymbol{z}_s}[\exp(\gamma_s)] \leq \exp(1)$. Taking full expectation,

$$
\mathbb{E}[\exp(\gamma_s)] \leq \exp(1).
$$

By Markov's inequality, for any $A \in \mathbb{R}$,

$$\mathbb{P}\left(\max_{s \in [T]} \gamma_s \geq A\right) = \mathbb{P}\left(\exp\left(\max_{s \in [T]} \gamma_s\right) \geq \exp(A)\right) \leq \exp(-A)\mathbb{E}\left[\exp\left(\max_{s \in [T]} \gamma_s\right)\right]$$

$$\leq \exp(-A)\mathbb{E}\left[\sum_{s=1}^{T} \exp\left(\gamma_s\right)\right] \leq \exp(-A)T\exp(1),$$

which leads to that with probability at least $1 - \delta$,

$$\|\boldsymbol{\xi}_s\|^2 \leq \log\left(\frac{\mathrm{e}T}{\delta}\right)\left(\sigma_0^2 + \sigma_1^2 \|\bar{\boldsymbol{g}}_s\|^p\right), \quad \forall s \in [T].$$

$\square$

*Proof of Lemma B.7.* Recalling the definitions of $\boldsymbol{a}_s$ in (23) and $\boldsymbol{\epsilon}_s$ in Algorithm 1, we have for any $s \in [T], i \in [d]$,

$$\frac{1}{a_{s,i}} \leq \frac{1}{\mathcal{G}_T(s)\sqrt{1-\beta_2} + \epsilon\sqrt{1-\beta_2^s}} \leq \frac{1}{(\mathcal{G}_T(s) + \epsilon)\sqrt{1-\beta_2}}$$

$$\leq \frac{1}{\mathcal{G}_T(s)\sqrt{1-\beta_2}} \leq \frac{1}{\sqrt{\sigma_0^2 + \sigma_1^2\|\bar{\boldsymbol{g}}_s\|^p}\sqrt{1-\beta_2}}. \tag{154}$$

Then given any $i \in [d]$, we set

$$X_s = -\eta_s\left\langle \bar{\boldsymbol{g}}_s, \frac{\boldsymbol{\xi}_s}{\boldsymbol{a}_s}\right\rangle, \omega_s = \eta_s\left\|\frac{\bar{\boldsymbol{g}}_s}{\boldsymbol{a}_s}\right\|\sqrt{\sigma_0^2 + \sigma_1^2\|\bar{\boldsymbol{g}}_s\|^p}, \quad \forall s \in [T].$$

Noting that $\bar{\boldsymbol{g}}_s, \boldsymbol{a}_s$ and $\eta_s$ are random variables dependent by $\boldsymbol{z}_1, \cdots, \boldsymbol{z}_{s-1}$ and $\boldsymbol{\xi}_s$ is only dependent on $\boldsymbol{z}_s$. We then verify that $X_s$ is a martingale difference sequence since

$$\mathbb{E}\left[X_s \mid \boldsymbol{z}_1, \cdots, \boldsymbol{z}_{s-1}\right] = \mathbb{E}_{\boldsymbol{z}_s}\left[-\eta_s\left\langle \bar{\boldsymbol{g}}_s, \frac{\boldsymbol{\xi}_s}{\boldsymbol{a}_s}\right\rangle\right] = -\eta_s\left\langle \bar{\boldsymbol{g}}_s, \frac{\mathbb{E}_{\boldsymbol{z}_s}[\boldsymbol{\xi}_s]}{\boldsymbol{a}_s}\right\rangle = 0.$$

Noting that $\omega_s$ is a random variable only dependent by $\boldsymbol{z}_1, \cdots, \boldsymbol{z}_{s-1}$ and applying Assumption (A3) and Cauchy-Schwarz inequality, we have

$$\mathbb{E}\left[\exp\left(\frac{X_s^2}{\omega_s^2}\right) \mid \boldsymbol{z}_1, \cdots, \boldsymbol{z}_{s-1}\right] \leq \mathbb{E}\left[\exp\left(\frac{\boldsymbol{\xi}_s^2}{\sigma_0^2 + \sigma_1^2\|\bar{\boldsymbol{g}}_s\|^p}\right) \mid \boldsymbol{z}_1, \cdots, \boldsymbol{z}_{s-1}\right]$$

$$\leq \mathbb{E}_{\boldsymbol{z}_s}\left[\exp\left(\frac{\|\boldsymbol{\xi}_s\|^2}{\sigma_0^2 + \sigma_1^2\|\bar{\boldsymbol{g}}_s\|^p}\right)\right] \leq \exp(1), \quad \forall s \in [T].$$

Applying Lemma A.3 and (154), we have that for any $\lambda > 0$, with probability at least $1 - \delta$,

$$\sum_{s=1}^{t} X_s \leq \frac{3\lambda}{4}\sum_{s=1}^{t}\omega_s^2 + \frac{1}{\lambda}\log\left(\frac{1}{\delta}\right)$$

$$\leq \frac{3\lambda}{4\sqrt{1-\beta_2}}\sum_{s=1}^{t}\eta_s^2\left\|\frac{\bar{\boldsymbol{g}}_s}{\boldsymbol{a}_s}\right\|^2(\sigma_0^2 + \sigma_1^2\|\bar{\boldsymbol{g}}_s\|^p) + \frac{1}{\lambda}\log\left(\frac{1}{\delta}\right)$$

$$\leq \frac{3\lambda}{4\sqrt{1-\beta_2}}\sum_{s=1}^{t}\eta_s^2\left\|\frac{\bar{\boldsymbol{g}}_s}{\sqrt{\boldsymbol{a}_s}}\right\|^2\sqrt{\sigma_0^2 + \sigma_1^2\|\bar{\boldsymbol{g}}_s\|^p} + \frac{1}{\lambda}\log\left(\frac{1}{\delta}\right). \tag{155}$$

Note that for any $t \in [T]$, (155) holds with probability at least $1 - \delta$. Then for any fixed $\lambda > 0$, we could re-scale $\delta$ to obtain that with probability at least $1 - \delta$, for all $t \in [T]$,

$$\sum_{s=1}^{t} X_s \leq \frac{3\lambda}{4\sqrt{1-\beta_2}}\sum_{s=1}^{t}\eta_s^2\left\|\frac{\bar{\boldsymbol{g}}_s}{\sqrt{\boldsymbol{a}_s}}\right\|^2\sqrt{\sigma_0^2 + \sigma_1^2\|\bar{\boldsymbol{g}}_s\|^p} + \frac{1}{\lambda}\log\left(\frac{T}{\delta}\right).$$

Using $\sqrt{\sigma_0^2 + \sigma_1^2\|\bar{\boldsymbol{g}}_s\|^p} \leq \mathcal{G}_T(t), s \leq t$ from (19), together with (30), we have that with probability at least $1 - \delta$,

$$-\sum_{s=1}^{t}\eta_s\left\langle \bar{\boldsymbol{g}}_s, \frac{\boldsymbol{\xi}_s}{\boldsymbol{a}_s}\right\rangle \leq \frac{3\lambda\eta\mathcal{G}_T(t)}{4(1-\beta_1)\sqrt{1-\beta_2}}\sum_{s=1}^{t}\eta_s\left\|\frac{\bar{\boldsymbol{g}}_s}{\sqrt{\boldsymbol{a}_s}}\right\|^2 + \frac{1}{\lambda}\log\left(\frac{T}{\delta}\right), \quad \forall t \in [T].$$

Finally setting $\lambda = (1-\beta_1)\sqrt{1-\beta_2}/(3\eta\mathcal{G}_T)$, we then have the desired result in (38). $\square$

### F.3 Omitted proof of Appendix B.4

*Proof of Lemma B.8.* First directly applying (36) and $G_s$ in (19), for any $j \in [s]$,

$$\|\boldsymbol{\xi}_j\| \leq \mathcal{M}_T \sqrt{\sigma_0^2 + \sigma_1^2 \|\bar{\boldsymbol{g}}_j\|^p} \leq \mathcal{M}_T \sqrt{\sigma_0^2 + \sigma_1^2 G_j^p} \leq \mathcal{M}_T \sqrt{\sigma_0^2 + \sigma_1^2 G_s^p} \leq \mathcal{G}_T(s).$$

Applying the basic inequality, (36) and $\mathcal{M}_T \geq 1$, for any $j \in [s]$,

$$\|\boldsymbol{g}_j\|^2 \leq 2\|\bar{\boldsymbol{g}}_j\|^2 + 2\|\boldsymbol{\xi}_j\|^2 \leq 2\mathcal{M}_T^2 \left(\sigma_0^2 + \sigma_1^2\|\bar{\boldsymbol{g}}_j\|^p + \|\bar{\boldsymbol{g}}_j\|^2\right) \leq (\mathcal{G}_T(s))^2.$$

Finally, we would use an induction argument to prove the last result. Given any $i \in [d]$, noting that $v_{1,i} = (1 - \beta_2)g_{1,i}^2 \leq (\mathcal{G}_T(s))^2$. Suppose that for some $s' \in [s]$, $v_{j,i} \leq (\mathcal{G}_T(s))^2, \forall j \in [s']$,

$$v_{s'+1,i} = \beta_2 v_{s',i} + (1 - \beta_2)g_{s',i}^2 \leq \beta_2(\mathcal{G}_T(s))^2 + (1 - \beta_2)(\mathcal{G}_T(s))^2 \leq (\mathcal{G}_T(s))^2.$$

We then obtain that $v_{j,i} \leq (\mathcal{G}_T(s))^2, \forall j \in [s]$. Noting that the above inequality holds for all $i \in [d]$, we therefore obtain the desired result. $\square$

*Proof of Lemma B.9.* Recalling the definition of $b_{s,i}$ in (16) and letting $a_{s,i} = \sqrt{\tilde{v}_{s,i}} + \epsilon_s$ in (23),

$$\left|\frac{1}{a_{s,i}} - \frac{1}{b_{s,i}}\right| = \frac{\left|\sqrt{v_{s,i}} - \sqrt{\tilde{v}_{s,i}}\right|}{a_{s,i}b_{s,i}} = \frac{1 - \beta_2}{a_{s,i}b_{s,i}} \frac{\left|g_{s,i}^2 - (\mathcal{G}_T(s))^2\right|}{\sqrt{v_{s,i}} + \sqrt{\tilde{v}_{s,i}}}$$

$$\leq \frac{1 - \beta_2}{a_{s,i}b_{s,i}} \cdot \frac{(\mathcal{G}_T(s))^2}{\sqrt{v_{s,i}} + \sqrt{\beta_2 v_{s-1,i} + (1 - \beta_2)(\mathcal{G}_T(s))^2}} \leq \frac{\mathcal{G}_T(s)\sqrt{1 - \beta_2}}{a_{s,i}b_{s,i}},$$

where we apply $g_{s,i}^2 \leq \|\boldsymbol{g}_s\|^2 \leq (\mathcal{G}_T(s))^2$ from Lemma B.8 in the first inequality since (36) holds. The second result also follows from the same analysis. We first combine with $\epsilon_s = \epsilon\sqrt{1 - \beta_2^s}$ to obtain that

$$|\epsilon_s - \epsilon_{s-1}| \leq \epsilon\left(\sqrt{1 - \beta_2^s} - \sqrt{1 - \beta_2^{s-1}}\right) \leq \epsilon\sqrt{\beta_2^{s-1}(1 - \beta_2)} \leq \epsilon\sqrt{1 - \beta_2}, \tag{156}$$

where we apply $\sqrt{a} - \sqrt{b} \leq \sqrt{a - b}, \forall 0 \leq b \leq a$. Applying the definition of $b_{s-1,i}$ and $a_{s,i}$,

$$\left|\frac{1}{b_{s-1,i}} - \frac{1}{a_{s,i}}\right| = \frac{\left|\sqrt{\tilde{v}_{s,i}} - \sqrt{v_{s-1,i}} + (\epsilon_s - \epsilon_{s-1})\right|}{b_{s-1,i}a_{s,i}}$$

$$\leq \frac{1}{b_{s-1,i}a_{s,i}} \frac{(1 - \beta_2)\left|(\mathcal{G}_T(s))^2 - v_{s-1,i}\right|}{\sqrt{\tilde{v}_{s,i}} + \sqrt{v_{s-1,i}}} + \frac{|\epsilon_s - \epsilon_{s-1}|}{b_{s-1,i}a_{s,i}}$$

$$\leq \frac{1}{b_{s-1,i}a_{s,i}} \cdot \frac{(1 - \beta_2)(\mathcal{G}_T(s))^2}{\sqrt{\tilde{v}_{s,i}} + \sqrt{v_{s-1,i}}} + \frac{\epsilon\sqrt{1 - \beta_2}}{b_{s-1,i}a_{s,i}} \leq \frac{(\mathcal{G}_T(s) + \epsilon)\sqrt{1 - \beta_2}}{b_{s-1,i}a_{s,i}}.$$

where the second inequality applies $v_{s-1,i} \leq (\mathcal{G}_T(s))^2$ in Lemma B.8 and the last inequality comes from $\sqrt{1 - \beta_2}\mathcal{G}_T(s) \leq \tilde{v}_{s,i}$. $\square$

*Proof of Lemma B.10.* Applying the basic inequality and (36), for all $t \in [T], i \in [d]$,

$$\sum_{s=1}^{t} g_{s,i}^2 \leq \sum_{s=1}^{t} \|\boldsymbol{g}_s\|^2 \leq 2\sum_{s=1}^{t}\left(\|\bar{\boldsymbol{g}}_s\|^2 + \|\boldsymbol{\xi}_s\|^2\right) \leq 2\mathcal{M}_T^2\left(\sigma_0^2 t + \sigma_1^2\sum_{s=1}^{t}\|\bar{\boldsymbol{g}}_s\|^p + \sum_{s=1}^{t}\|\bar{\boldsymbol{g}}_s\|^2\right).$$
$$\tag{157}$$

Combining with Lemma B.2, we have

$$\sum_{s=1}^{t}\|\bar{\boldsymbol{g}}_s\|^p \leq \sum_{s=1}^{t}\left(\|\bar{\boldsymbol{g}}_1\| + \frac{LC_0\sqrt{d}s}{\sqrt{1 - \beta_1/\beta_2}}\right)^p \leq t \cdot \left(\|\bar{\boldsymbol{g}}_1\| + \frac{LC_0\sqrt{d}t}{\sqrt{1 - \beta_1/\beta_2}}\right)^p$$

$$\sum_{s=1}^{t}\|\bar{\boldsymbol{g}}_s\|^2 \leq t \cdot \left(\|\bar{\boldsymbol{g}}_1\| + \frac{LC_0\sqrt{d}t}{\sqrt{1 - \beta_1/\beta_2}}\right)^2.$$

Further applying the definition of $\mathcal{F}_i(t)$ in Lemma B.3, it leads to $\mathcal{F}_i(t) \leq \mathcal{F}(t), \forall i \in [d]$. Finally, since $\mathcal{F}(t)$ is increasing with $t$, we obtain the desired result. $\square$

*Proof of Lemma B.11.* First, we have the following decomposition,

$$\mathbf{A.1} = -\sum_{s=1}^{t}\eta_s\left\|\frac{\bar{\boldsymbol{g}}_s}{\sqrt{\boldsymbol{a}_s}}\right\|^2 \underbrace{-\sum_{s=1}^{t}\eta_s\left\langle\bar{\boldsymbol{g}}_s,\frac{\boldsymbol{\xi}_s}{\boldsymbol{a}_s}\right\rangle}_{\mathbf{A.1.1}} + \underbrace{\sum_{s=1}^{t}\eta_s\left\langle\bar{\boldsymbol{g}}_s,\left(\frac{1}{\boldsymbol{a}_s}-\frac{1}{\boldsymbol{b}_s}\right)\boldsymbol{g}_s\right\rangle}_{\mathbf{A.1.2}}. \qquad (158)$$

Since (36) holds, we could apply Cauchy-Schwarz inequality, Lemma B.9, and $\mathcal{G}_T(s) \leq \mathcal{G}_T(t), \forall s \leq t$ from (19) to obtain that for all $t \in [T]$,

$$\mathbf{A.1.2} \leq \sum_{i=1}^{d}\sum_{s=1}^{t}\eta_s\left|\frac{1}{a_{s,i}}-\frac{1}{b_{s,i}}\right|\cdot|\bar{g}_{s,i}g_{s,i}| \leq \sum_{i=1}^{d}\sum_{s=1}^{t}\eta_s\cdot\frac{\mathcal{G}_T(s)\sqrt{1-\beta_2}}{a_{s,i}b_{s,i}}\cdot|\bar{g}_{s,i}g_{s,i}|$$

$$\leq \frac{1}{4}\sum_{i=1}^{d}\sum_{s=1}^{t}\frac{\eta_s\bar{g}_{s,i}^2}{a_{s,i}} + (1-\beta_2)\sum_{i=1}^{d}\sum_{s=1}^{t}\frac{(\mathcal{G}_T(s))^2}{a_{s,i}}\cdot\frac{\eta_s g_{s,i}^2}{b_{s,i}^2}$$

$$\overset{(30),(154)}{\leq} \frac{1}{4}\sum_{s=1}^{t}\eta_s\left\|\frac{\bar{\boldsymbol{g}}_s}{\sqrt{\boldsymbol{a}_s}}\right\|^2 + \frac{\eta\mathcal{G}_T(t)\sqrt{1-\beta_2}}{1-\beta_1}\sum_{s=1}^{t}\left\|\frac{\boldsymbol{g}_s}{\boldsymbol{b}_s}\right\|^2.$$

Finally, combining with (38) for estimating **A.1.1**, we deduce the desired result in (40). $\qquad\square$

*Proof of Lemma B.12.* Let us denote $\Sigma := \frac{\beta_1}{1-\beta_1}\langle\Delta_s\odot(\boldsymbol{x}_s-\boldsymbol{x}_{s-1}),\bar{\boldsymbol{g}}_s\rangle$ where $\Delta_s$ is defined in (20). We have

$$\Sigma \leq \frac{\beta_1}{1-\beta_1}\cdot\left|\left\langle\Delta_s\odot\frac{\eta_{s-1}\boldsymbol{m}_{s-1}}{\boldsymbol{b}_{s-1}},\bar{\boldsymbol{g}}_s\right\rangle\right| = \frac{\beta_1}{1-\beta_1}\cdot\left|\left\langle\left(\frac{\eta_s}{\boldsymbol{b}_s}-\frac{\eta_{s-1}}{\boldsymbol{b}_{s-1}}\right)\odot\boldsymbol{m}_{s-1},\bar{\boldsymbol{g}}_s\right\rangle\right|$$

$$\leq \underbrace{\frac{\beta_1}{1-\beta_1}\cdot\left|\left\langle\left(\frac{\eta_s}{\boldsymbol{b}_s}-\frac{\eta_s}{\boldsymbol{a}_s}\right)\odot\boldsymbol{m}_{s-1},\bar{\boldsymbol{g}}_s\right\rangle\right|}_{\Sigma_1} + \underbrace{\frac{\beta_1}{1-\beta_1}\cdot\left|\left\langle\left(\frac{\eta_s}{\boldsymbol{a}_s}-\frac{\eta_s}{\boldsymbol{b}_{s-1}}\right)\odot\boldsymbol{m}_{s-1},\bar{\boldsymbol{g}}_s\right\rangle\right|}_{\Sigma_2}$$

$$+ \underbrace{\frac{\beta_1}{1-\beta_1}\cdot\left|(\eta_{s-1}-\eta_s)\left\langle\frac{\boldsymbol{m}_{s-1}}{\boldsymbol{b}_{s-1}},\bar{\boldsymbol{g}}_s\right\rangle\right|}_{\Sigma_3}. \qquad (159)$$

Since (36) holds, we could apply Lemma B.9 and Young's inequality and then use (154), (30), $\beta_1 \in [0,1)$ and $\mathcal{G}_T(s) \leq \mathcal{G}_T(t) \leq \mathcal{G}_T(t) + \epsilon, \forall s \leq t$,

$$\Sigma_1 \leq \sum_{i=1}^{d}\frac{\beta_1}{1-\beta_1}\cdot\frac{\mathcal{G}_T(s)\eta_s\sqrt{1-\beta_2}}{a_{s,i}b_{s,i}}\cdot|\bar{g}_{s,i}m_{s-1,i}|$$

$$\leq \sum_{i=1}^{d}\frac{\eta_s}{8}\cdot\frac{\bar{g}_{s,i}^2}{a_{s,i}} + \frac{2\eta_s\beta_1^2(1-\beta_2)}{(1-\beta_1)^2}\sum_{i=1}^{d}\frac{(\mathcal{G}_T(s))^2}{a_{s,i}}\cdot\frac{m_{s-1,i}^2}{b_{s,i}^2}$$

$$\leq \frac{\eta_s}{8}\left\|\frac{\bar{\boldsymbol{g}}_s}{\sqrt{\boldsymbol{a}_s}}\right\|^2 + \frac{2(\mathcal{G}_T(t)+\epsilon)\eta\sqrt{1-\beta_2}}{(1-\beta_1)^3}\left\|\frac{\boldsymbol{m}_{s-1}}{\boldsymbol{b}_s}\right\|^2. \qquad (160)$$

Using the similar analysis for $\Sigma_1$, we also have

$$\Sigma_2 \leq \sum_{i=1}^{d}\frac{\eta_s\beta_1}{1-\beta_1}\frac{\sqrt{1-\beta_2}}{a_{s,i}b_{s-1,i}}\cdot(\mathcal{G}_T(s)+\epsilon)\cdot|\bar{g}_{s,i}\cdot m_{s-1,i}|$$

$$\leq \frac{\eta_s}{8}\left\|\frac{\bar{\boldsymbol{g}}_s}{\sqrt{\boldsymbol{a}_s}}\right\|^2 + \frac{2(\mathcal{G}_T(t)+\epsilon)\eta\sqrt{1-\beta_2}}{(1-\beta_1)^3}\left\|\frac{\boldsymbol{m}_{s-1}}{\boldsymbol{b}_{s-1}}\right\|^2. \qquad (161)$$

Then we move to bound the summation of $\Sigma_3$ over $s \in \{2, \cdots, t\}$ since $\boldsymbol{m}_0 = 0$. Recalling $\eta_s$ in (30), we have the following decomposition,

$$\Sigma_3 \leq \underbrace{\frac{\eta\beta_1\sqrt{1-\beta_2^s}}{1-\beta_1}\left|\left(\frac{1}{1-\beta_1^{s-1}} - \frac{1}{1-\beta_1^s}\right)\left\langle \bar{\boldsymbol{g}}_s, \frac{\boldsymbol{m}_{s-1}}{\boldsymbol{b}_{s-1}}\right\rangle\right|}_{\Sigma_{3.1}}$$

$$+ \underbrace{\frac{\eta\beta_1}{(1-\beta_1)(1-\beta_1^{s-1})}\left|\left(\sqrt{1-\beta_2^{s-1}} - \sqrt{1-\beta_2^s}\right)\left\langle \bar{\boldsymbol{g}}_s, \frac{\boldsymbol{m}_{s-1}}{\boldsymbol{b}_{s-1}}\right\rangle\right|}_{\Sigma_{3.2}}. \tag{162}$$

Noting that $\|\bar{\boldsymbol{g}}_s\| \leq G_s \leq G_t, \forall s \leq t$. Then further applying Cauchy-Schwarz inequality and Lemma B.2,

$$\sqrt{1-\beta_2^s}\left|\left\langle \bar{\boldsymbol{g}}_s, \frac{\boldsymbol{m}_{s-1}}{\boldsymbol{b}_{s-1}}\right\rangle\right| \leq \sqrt{1-\beta_2^s}\|\bar{\boldsymbol{g}}_s\|\left\|\frac{\boldsymbol{m}_{s-1}}{\boldsymbol{b}_{s-1}}\right\| \leq \sqrt{d}G_t\sqrt{\frac{(1-\beta_1)(1-\beta_1^{s-1})}{(1-\beta_2)(1-\beta_1/\beta_2)}}.$$

Hence, summing $\Sigma_{3.1}$ up over $s \in [t]$, applying $\beta_1 \in (0,1)$ and noting that $\Sigma_{3.1}$ vanishes when $s = 1$,

$$\sum_{s=1}^t \Sigma_{3.1} \leq \frac{\sqrt{d}\eta G_t}{1-\beta_1} \cdot \sqrt{\frac{1-\beta_1}{(1-\beta_2)(1-\beta_1/\beta_2)}}\sum_{s=2}^t\left(\frac{1}{1-\beta_1^{s-1}} - \frac{1}{1-\beta_1^s}\right)$$

$$\leq \frac{\sqrt{d}\eta G_t}{\sqrt{(1-\beta_1)^3(1-\beta_2)(1-\beta_1/\beta_2)}}. \tag{163}$$

Similarly, using $\|\bar{\boldsymbol{g}}_s\| \leq G_s \leq G_t, \forall s \leq t$ and $1 - \beta_1^{s-1} \geq 1 - \beta_1$,

$$\frac{1}{1-\beta_1^{s-1}}\left|\left\langle \bar{\boldsymbol{g}}_s, \frac{\boldsymbol{m}_{s-1}}{\boldsymbol{b}_{s-1}}\right\rangle\right| \leq \frac{1}{1-\beta_1^{s-1}}\|\bar{\boldsymbol{g}}_s\|\left\|\frac{\boldsymbol{m}_{s-1}}{\boldsymbol{b}_{s-1}}\right\| \leq \sqrt{d}G_t\sqrt{\frac{1}{(1-\beta_2)(1-\beta_1/\beta_2)}}.$$

Hence, summing $\Sigma_{3.2}$ up over $s \in [t]$ and still applying $\beta_1 \in [0,1)$,

$$\sum_{s=1}^t \Sigma_{3.2} \leq \frac{\sqrt{d}\eta G_t}{1-\beta_1} \cdot \sqrt{\frac{1}{(1-\beta_2)(1-\beta_1/\beta_2)}}\sum_{s=2}^t\left(\sqrt{1-\beta_2^s} - \sqrt{1-\beta_2^{s-1}}\right)$$

$$\leq \frac{\sqrt{d}\eta G_t}{(1-\beta_1)\sqrt{(1-\beta_2)(1-\beta_1/\beta_2)}} \leq \frac{\sqrt{d}\eta G_t}{\sqrt{(1-\beta_1)^3(1-\beta_2)(1-\beta_1/\beta_2)}}. \tag{164}$$

Combining with (162), (163) and (164), we obtain an upper bound for $\sum_{s=1}^t \Sigma_3$. Summing (159), (160) and (161) up over $s \in [t]$, and combining with the estimation for $\sum_{s=1}^t \Sigma_3$, we obtain the desired inequality in (41). □

## G  Omitted proof in Appendix D

*Proof of Lemma D.1.* Recalling in (152), we have already shown that

$$\|\boldsymbol{x}_{s+1} - \boldsymbol{x}_s\| \leq \sqrt{d}\|\boldsymbol{x}_{s+1} - \boldsymbol{x}_s\|_\infty \leq \eta\sqrt{\frac{d}{(1-\beta_2)(1-\beta_1/\beta_2)}}, \quad \forall s \geq 1. \tag{165}$$

Applying the definition of $\boldsymbol{y}_s$ in (17), an intermediate result in (165) and $\beta_1 \in [0,1)$,[8]

$$\|\boldsymbol{y}_s - \boldsymbol{x}_s\| = \frac{\beta_1}{1-\beta_1}\|\boldsymbol{x}_s - \boldsymbol{x}_{s-1}\| \leq \frac{\eta}{1-\beta_1}\sqrt{\frac{d}{(1-\beta_2)(1-\beta_1/\beta_2)}}, \quad \forall s \geq 1. \tag{166}$$

---

[8]The inequality still holds for $s = 1$ since $\boldsymbol{x}_1 = \boldsymbol{y}_1$.

Recalling the iteration of $\boldsymbol{y}_s$ in (18) and then using Young's inequality

$$\|\boldsymbol{y}_{s+1} - \boldsymbol{y}_s\|^2 \leq \underbrace{2\eta_s^2 \left\|\frac{\boldsymbol{g}_s}{\boldsymbol{b}_s}\right\|^2}_{(*)} + \underbrace{\frac{2\beta_1^2}{(1-\beta_1)^2} \left\|\frac{\eta_s \boldsymbol{b}_{s-1}}{\eta_{s-1}\boldsymbol{b}_s} - \mathbf{1}\right\|_\infty^2 \|\boldsymbol{x}_s - \boldsymbol{x}_{s-1}\|^2}_{(**)}.$$

Noting that $g_{s,i}/b_{s,i} \leq 1/\sqrt{1-\beta_2}$ from (16), we then combine with (30) to have

$$(*) \leq 2\eta_s^2 \cdot \frac{d}{1-\beta_2} \leq \frac{2\eta^2 d}{(1-\beta_1)^2(1-\beta_2)}.$$

Applying Lemma B.1 where $\Sigma_{\max}^2 \leq 1/\beta_2$ and (165),

$$(**) \leq \frac{2\eta^2 \beta_1^2 \Sigma_{\max}^2 d}{(1-\beta_1)^2(1-\beta_2)(1-\beta_1/\beta_2)} \leq \frac{2\eta^2 d}{\beta_2(1-\beta_1)^2(1-\beta_2)(1-\beta_1/\beta_2)}.$$

Summing up two estimations and using $0 \leq \beta_1 < \beta_2 < 1$, we finally have

$$\|\boldsymbol{y}_{s+1} - \boldsymbol{y}_s\| \leq \eta\sqrt{\frac{4d}{\beta_2(1-\beta_1)^2(1-\beta_2)(1-\beta_1/\beta_2)}}. \tag{167}$$

Combining with (165), (166) and (167), and using $0 \leq \beta_1 < \beta_2 < 1$, we then deduce a uniform bound for all the three gaps. $\qquad\square$

*Proof of Lemma D.2.* Under the same conditions in Lemma D.1, we have

$$\|\boldsymbol{y}_s - \boldsymbol{x}_s\| \leq \frac{1}{L_q}, \quad \|\boldsymbol{y}_{s+1} - \boldsymbol{y}_s\| \leq \frac{1}{L_q}.$$

Then, using the generalized smoothness in (8),

$$\begin{aligned}
\|\nabla f(\boldsymbol{y}_s)\| &\leq \|\nabla f(\boldsymbol{x}_s)\| + \|\nabla f(\boldsymbol{y}_s) - \nabla f(\boldsymbol{x}_s)\| \\
&\leq \|\nabla f(\boldsymbol{x}_s)\| + (L_0 + L_q\|\nabla f(\boldsymbol{x}_s)\|^q)\|\boldsymbol{y}_s - \boldsymbol{x}_s\| \\
&\leq \|\nabla f(\boldsymbol{x}_s)\| + \|\nabla f(\boldsymbol{x}_s)\|^q + L_0/L_q.
\end{aligned}$$

We could use a similar argument to deduce the bound for $\|\nabla f(\boldsymbol{x}_s)\|$. Further, combining with $\mathcal{L}_s^{(x)}$ and $\mathcal{L}_s^{(y)}$ in (97), we could bound the generalized smooth parameters as

$$\begin{aligned}
L_0 + L_q\|\nabla f(\boldsymbol{x}_s)\|^q &\leq L_0 + L_q G_s^q = \mathcal{L}_s^{(x)}, \\
L_0 + L_q\|\nabla f(\boldsymbol{y}_s)\|^q &\leq L_0 + L_q(\|\nabla f(\boldsymbol{x}_s)\| + \|\nabla f(\boldsymbol{x}_s)\|^q + L_0/L_q)^q = \mathcal{L}_s^{(y)}. \tag{168}
\end{aligned}$$

We could then deduce the first two inequalities in (100). Finally, (101) could be deduced by using the same argument in the proof of [50, Lemma A.3]. $\qquad\square$

*Proof of Lemma D.3.* Given any $\boldsymbol{x} \in \mathbb{R}^d$, we let

$$\tau = \frac{1}{L_0 + L_q \max\{\|\nabla f(\boldsymbol{x})\|^q, \|\nabla f(\boldsymbol{x})\|\}}, \quad \hat{\boldsymbol{x}} = \boldsymbol{x} - \tau\nabla f(\boldsymbol{x}).$$

From the definition of $\tau$, we could easily verify that $\|\hat{\boldsymbol{x}} - \boldsymbol{x}\| = \tau\|\nabla f(\boldsymbol{x})\| \leq 1/L_q$. Since $f$ is $(L_0, L_q)$-smooth, we could thereby use the descent lemma in [50, Lemma A.3] such that

$$\begin{aligned}
f(\hat{\boldsymbol{x}}) &\leq f(\boldsymbol{x}) + \langle\nabla f(\boldsymbol{x}), \hat{\boldsymbol{x}} - \boldsymbol{x}\rangle + \frac{L_0 + L_q\|\nabla f(\boldsymbol{x})\|^q}{2}\|\hat{\boldsymbol{x}} - \boldsymbol{x}\|^2 \\
&= f(\boldsymbol{x}) - \tau\|\nabla f(\boldsymbol{x})\|^2 + \frac{(L_0 + L_q\|\nabla f(\boldsymbol{x})\|^q)\tau^2}{2}\|\nabla f(\boldsymbol{x})\|^2 \leq f(\boldsymbol{x}) - \frac{\tau}{2}\|\nabla f(\boldsymbol{x})\|^2.
\end{aligned}$$

Since $f(\hat{\boldsymbol{x}}) \geq f^*$, when $\|\nabla f(\boldsymbol{x})\| = 0$, the desired result is trivial. Let us suppose $\|\nabla f(\boldsymbol{x})\| > 0$.

**Case 1** $\|\nabla f(\boldsymbol{x})\|^q > \|\nabla f(\boldsymbol{x})\|$

$$\frac{\tau}{2}\|\nabla f(\boldsymbol{x})\|^2 = \frac{\|\nabla f(\boldsymbol{x})\|^{2-q}}{2L_0/\|\nabla f(\boldsymbol{x})\|^q + 2L_q} \leq f(\boldsymbol{x}) - f(\hat{\boldsymbol{x}}) \leq f(\boldsymbol{x}) - f^*.$$

When $\|\nabla f(\boldsymbol{x})\|^q < L_0/L_q$, it leads to

$$\frac{\|\nabla f(\boldsymbol{x})\|^2}{4L_0} = \frac{\|\nabla f(\boldsymbol{x})\|^{2-q}}{4L_0/\|\nabla f(\boldsymbol{x})\|^q} \leq \frac{\|\nabla f(\boldsymbol{x})\|^{2-q}}{2L_0/\|\nabla f(\boldsymbol{x})\|^q + 2L_q} \leq f(\boldsymbol{x}) - f^*.$$

When $\|\nabla f(\boldsymbol{x})\|^q \geq L_0/L_q$, it leads to

$$\frac{\|\nabla f(\boldsymbol{x})\|^{2-q}}{4L_q} \leq \frac{\|\nabla f(\boldsymbol{x})\|^{2-q}}{2L_0/\|\nabla f(\boldsymbol{x})\|^q + 2L_q} \leq f(\boldsymbol{x}) - f^*.$$

We then deduce that

$$\|\nabla f(\boldsymbol{x})\| \leq \max\left\{ [4L_q(f(\boldsymbol{x}) - f^*)]^{\frac{1}{2-q}}, \sqrt{4L_0(f(\boldsymbol{x}) - f^*)} \right\}. \tag{169}$$

**Case 2** $\|\nabla f(\boldsymbol{x})\|^q \leq \|\nabla f(\boldsymbol{x})\|$ We could rely on the similar analysis to obtain that[9]

$$\|\nabla f(\boldsymbol{x})\| \leq \max\left\{ 4L_q(f(\boldsymbol{x}) - f^*), \sqrt{4L_0(f(\boldsymbol{x}) - f^*)} \right\}. \tag{170}$$

Combining (169) and (170), we then deduce the desired result. $\qquad\square$

*Proof of Lemma D.5.* Recalling (152), we then obtained that when $\eta = \tilde{C}_0\sqrt{1-\beta_2}$,

$$\|\boldsymbol{x}_s - \boldsymbol{x}_{s-1}\| \leq \sqrt{d}\|\boldsymbol{x}_s - \boldsymbol{x}_{s-1}\|_\infty \leq \tilde{C}_0\sqrt{\frac{d}{1-\beta_1/\beta_2}}. \tag{171}$$

Noting that when (99) holds, we have

$$\|\bar{\boldsymbol{g}}_s\| \leq \|\bar{\boldsymbol{g}}_{s-1}\| + \|\bar{\boldsymbol{g}}_s - \bar{\boldsymbol{g}}_{s-1}\| \leq \|\bar{\boldsymbol{g}}_{s-1}\| + (L_0 + L_q\|\bar{\boldsymbol{g}}_{s-1}\|^q)\|\boldsymbol{x}_s - \boldsymbol{x}_{s-1}\|$$

$$\leq \|\bar{\boldsymbol{g}}_{s-1}\| + \tilde{C}_0\mathcal{L}_{s-1}^{(x)}\sqrt{\frac{d}{1-\beta_1/\beta_2}} \leq \|\bar{\boldsymbol{g}}_1\| + \tilde{C}_0\sqrt{\frac{d}{1-\beta_1/\beta_2}}\sum_{j=1}^{s-1}\mathcal{L}_j^{(x)}.$$

Using $\mathcal{L}_j^{(x)} \leq \mathcal{L}_t^{(x)}, \forall j \leq t$, we have

$$\sum_{s=1}^t \|\bar{\boldsymbol{g}}_s\|^p \leq \sum_{s=1}^t \left( \|\bar{\boldsymbol{g}}_1\| + \tilde{C}_0\sqrt{\frac{d}{1-\beta_1/\beta_2}}(s-1)\mathcal{L}_t^{(x)} \right)^p \leq t\left( \|\bar{\boldsymbol{g}}_1\| + t\tilde{C}_0\mathcal{L}_t^{(x)}\sqrt{\frac{d}{1-\beta_1/\beta_2}} \right)^p.$$

Similarly, we also have

$$\sum_{s=1}^t \|\bar{\boldsymbol{g}}_s\|^2 \leq t\left( \|\bar{\boldsymbol{g}}_1\| + t\tilde{C}_0\mathcal{L}_t^{(x)}\sqrt{\frac{d}{1-\beta_1/\beta_2}} \right)^2.$$

Further combining with $\mathcal{F}_i(t)$ in Lemma B.3 and $\mathcal{J}(t)$ in (104),

$$\mathcal{F}_i(t) \leq 1 + \frac{1}{\epsilon^2}\sum_{s=1}^t \|\boldsymbol{g}_s\|^2 \leq \mathcal{J}(t), \quad \forall t \in [T], i \in [d].$$

$\qquad\square$

---

[9]We refer readers to see [52, Lemma A.5] for a detailed proof under this case.

