# OpenReview forum: "On Convergence of Adam for Stochastic Optimization under Relaxed Assumptions"
_NeurIPS.cc/2024/Conference — NeurIPS 2024 poster_

### Official Review · Reviewer_hewP · 2024-06-18

**Soundness:** 3
**Presentation:** 3
**Contribution:** 2
**Rating:** 7
**Confidence:** 5

**Summary:**

This paper provides two probabilistic convergence rates for Adam with generalized affine variance noise under smoothness and generalized smooth condition, respectively, which achieves comparable results to many prior results.

**Strengths:**

Please see the above Summary.

**Weaknesses:**

1. I suggest that authors should provide detailed formulas for some notations including $\textbf{g}_t, g(\textbf{x}), \nabla f(\textbf{x})$. Is $\nabla f(\textbf{x})$ the gradient with the form of expectation.

2. In line 118, the reference [10] is cited repeatedly.

3. Section 5 is used to discuss the most related works and make comparisons with the main results in this paper. However, authors only discuss the most related works without any comparison with their main results.

4. As mentioned in 1., if $\nabla f(\textbf{x})$ is the gradient with the form of expectation, the two results (Theorems 3.1 and 4.1) in this paper are not fully high probability since $\frac{1}{T} \sum_{t=1}^T ||\nabla f(x_t)||^2$ is equivalent to $\frac{1}{T} \sum_{t=1}^T ||E_{z_i}[\nabla f(x_t,z_i)]||^2$ ($z_i$ is the training data defined by me) which is smaller than $\frac{1}{T} \sum_{t=1}^T E_{z_i}[||\nabla f(x_t,z_i)||^2]$. In other words, the results with the form $\frac{1}{T} \sum_{t=1}^T E_{z_i}[||\nabla f(x_t,z_i)||^2]$ can directly derive the corresponding results with the form $\frac{1}{T} \sum_{t=1}^T ||E_{z_i}[\nabla f(x_t,z_i)]||^2$, to say nothing of the one with additional high probability. Therefore, from my perspective, high probability is not an advantage for this paper, but rather weakens $\frac{1}{T} \sum_{t=1}^T ||E_{z_i}[\nabla f(x_t,z_i)]||^2$.

**Questions:**

1. What is the symbol $\xi$ in Equation (1)? It should be explained in the main text.

2. What are the meanings of “right parameter” mentioned in line 4 (Abstract) and “problem parameter” in line 19?

3. Although, in line 139, authors denote “it’s easy to verify that (8) is strictly weaker than L-smoothness” and provide a concrete example $f(x)=x^4$, can detailed proof be given to verify this argument?

4. In Table 1, authors make comparisons with some prior work. However, the forms of some results (such as [33, 40, 49]) are not in the same form as the average form $\frac{1}{T} \sum_{t=1}^T ||\nabla f(x_t)||^2$ in this paper. Therefore, are these comparisons appropriate?

**Limitations:**

The contribution of this paper is a bit weak. Although authors weaken some assumptions, such as noise and smoothness, the form of their results is also weakened. Therefore, I don’t ensure the weakened assumptions are their advantages.

---

> ### Author Rebuttal · Authors · 2024-08-06
>
> We thanks a lot for the reviewer's effort invested on our paper. Below are our responses to the major concern.
>
> **Response to Weaknesses 1-3: Thank you for the suggestions on the presentation. We will revise as follows accordingly.**
>
> 1. In Line 14, we will replace '$g_t$' with '$g_t = \frac{\partial f_\xi(x,\xi_t)} {\partial x}|_{x=x_t}$'.\
> In Line 16, we will replace '$g(x)$ is unbiased' with '$g(x,\xi)$ is an unbiased estimate of $\nabla f(x)$ such that $E [g(x,\xi)|x] = \nabla f(x)$'. \
> In Eq. (2), we will replace '$E[\|g(x) - \nabla f(x)||_2^2]$' with '$E[\|g(x,\xi) - \nabla f(x)||_2^2|x]$'.  \
> In general,  $\nabla f(x)$ does not need to be of the expectation form, while for Eq. (1) ,
>
> $$
> \ \  \ \ \  \ \ \ \nabla f(x) =  E \left[\frac{\partial f_{\xi}(x,\xi)}{\partial x}\mid x \right]
> $$
>
> 2. In Line 118, we will replace '[22, 10, 53, 10, 38]' with '[22, 10, 53, 38]'.
>
> 3. To accomplish Table 1, we could add some detailed comparisons on convergence for Adam as follows.  \
> In Line 189, we will add 'In comparisons with these works, our hyper-parameter setup could be relatively simpler than [49],  while our results do not need the assumption on bounded gradients from [18, 20]. Moreover, we study Adam with two corrective terms which were overlooked in these works.' \
> In Line 203, we will add 'In comparisons, our hyper-parameter setup on $\beta_2$ is simpler than [40] while involving the natural corrective terms. Moreover, we consider a milder noise assumption than the sub-Gaussian noise assumption in [23], with different hyper-parameter setups on $\beta_1$ and $\beta_2$.'
>
> **Response to Weakness 4: We clarify as follows.**
> - **High-probability convergence can ensure expected convergence:** \
> Let $\Sigma =  \sqrt{T} \times \frac{1}{T}\sum_{t=1}^T||\nabla f(x_t)||^2.$
> Our result shows that $$Prob\left(\Sigma \gtrsim { \log{1\over \delta}}\right) \leq \delta.$$
> Then for some constants $c, C$,
> $$ E[\Sigma] =  \int_{0}^\infty Prob(\Sigma \geq y) dy \leq \int_{0}^\infty e^{-cy} dy \simeq C.$$
> - *$  \frac{1}{T}\sum_{t=1}^T E_{z_i}||\nabla f(x_t,z_i)||^2$ (you mentioned) is the average of **conditional** expectations, while $\frac{1}{T}\sum_{t=1}^TE ||\nabla f(x_t)||^2$ is the average of **full/total** expectations.* \
> Previous expected convergence results are on $\frac{1}{T}\sum_{t=1}^TE ||\nabla f(x_t)||^2$. We think that the form you mentioned could be meaningful for the future work. However, no results are given for $  \frac{1}{T}\sum_{t=1}^T E_{z_i}||\nabla f(x_t,z_i)||^2$, to our knowledge.
>
> **Response to Questions 1-4**
>
> 1. $\xi$ is the random vector following a probability distribution $\mathcal{D}$. We will add this statement in Line 14 of the revision.
>
> 3. The 'right parameter' means the hyper-parameter setup such as (6) in Theorem 3.1. The 'problem parameter' here refers to the (generalized) smooth and noise level parameters. We will make this clearer in the revision.
>
> 3. Proof for '(8) weaker than $L$-smoothness':  If $f$ is $L$-smooth, i.e., $|| \nabla f(x) - \nabla f(y)|| \leq L ||x-y||$ for all $x,y$, then $(8)$ is trivially satisfied for $L_0 = L$ and any $L_q>0$. \
> Proof for `$f(x)=x^4$ satisfies (8)':  First, $|f'(y)-f'(x)| = 4|y^3-x^3| =4 |y^2+xy+x^2||y-x| $. Observe that for any constant $L >0$, there always exist $x,y \in \mathbb{R}$ such that $4|y^2+xy+x^2 |> L$ and thus the $L$-smoothness condition does not hold. However, when restricting $|x-y| \le 1/L_q$, we could derive that
> $$
> 4|y^2+xy+x^2| \le 6(y^2 + x^2) \le 6(2x^2+\frac{2}{L_q^2} + x^2) \le 18x^2+\frac{12}{L_q^2} \le \frac{18}{4^{\frac{2}{3}}}  (4x^3)^{\frac{2}{3}} +\frac{12}{L_q^2},
> $$
> where we set $L_q = 18/(4^{3/2})$, $q= 2/3$ and $L_0 = 12/L_q^2$, which leads to (8).
>
> 3. [33, 40, 49] study Adam with random-reshuffling samplings and  their results are w.r.t. $\min_{k \in [T]}||\nabla f(x_{k,0})||$. Here $x_{k,0}$ denotes the output of the $k$-th epoch (i.e. $kn$-th iteration) of Adam, and  $n$ is the number of training data. We will add this remark after Table 1 in the revision.
>
> **Response to Limitations: We clarify that our high probability bounds are stronger than the expected bounds**, following from the 'Response to Weakness 4'. The brief introduction of our proof novelty could be seen in the global rebuttal.

---

> > ### Comment · Reviewer_hewP · 2024-08-08
> >
> > Thanks for the detailed responses to my questions. I will raise my score to 6. However, I am still confused about the sentence "In general, $\nabla f(x)$ does not need to be of the expectation form, while for Eq. (1)". Why is Eq. (1) of the expectation form, but the author says there is no need for expected form? And due to the implicit expectation of $\nabla f(x)$, the results of this paper are not fullly high-probability. Is my understanding right?

---

> ### Author Response · Authors · 2024-08-08
>
> We thank a lot for the reviewer's quick feedback and generosity in raising the score. As far as we can understand, the reviewer still has two questions:
>
> - whether there is an expectation form in $\nabla f(x)$;
> - the result is not full high probability.
>
> We then answer as follows:
>
> - **The loss function $f$ is not necessarily the expected form in Eq. (1).** We indeed only require $f$ to be differentiable and smooth. We will clarify this in the revised version. In this sense, $\nabla f(x)$ also not necessarily includes the expectation.
> - **$\\|\nabla f(x)\\|^2 \le \varepsilon$ implies that $x$ is near-stationary, which is a standard measure in non-convex smooth optimization.** Our high probability result shows that $\frac{1}{T}\sum_{t=1}^T \\|\nabla f(x_t)\\|^2\le \varepsilon$, indicating that there exists at least one near-stationary point $x_{\tau},\tau \in [T]$.
> - As far as we could understand, (please correct me if I was wrong), the fully high probability you mention is to derive a high probability bound for $\frac{1}{T}\sum_{t=1}^T \\|g(x_t,z_t)\\|^2$. We think that this could be meaningful. However, given the second point, the high probability bound related to $\frac{1}{T}\sum_{t=1}^T \\|\nabla f(x_t)\\|^2$ could implies a near-sationary point while the bound for $\frac{1}{T}\sum_{t=1}^T \\|g(x_t,z_t)\\|^2$ may not, which we will show through an example at the end.
> - In addition, since $\frac{1}{T}\sum_{t=1}^T \\|\nabla f(x_t)\\|^2$ is an entirely random variable with respect to $z_1,\cdots,z_{T}$, **our result is fully high probability over random samples $z_1,\cdots,z_T$​.** However, when Assumption (A2) holds, we could choose any random sample $z_t'$ that is independent from $z_t$ and obtain that $\nabla f(x_t) = \mathbb{E}_{z_t'}[g(x_t,z_t')]$. Hence, due to the additional expectation on $z_t'$, we agree that the obtained result is not fully high probability without restricting on $z_1,\cdots,z_T$.
>
> We also present a simple example to help better understand the difference of two types of measure. Suppose that the loss function is $f(x)=x^4$, which does not include expectation in its gradient. We now consider the following SGD algorithm (with differences to Adam in step-size):
>
> - Input $x_1$ and step-size $\eta > 0$;
> - For $t = 1\cdots T:$
>   - Draw a random sample $z_t$ such that $\mathbb{E}[z_t \mid x_t]=0$ (e.g., a Gaussian white noise);
>   - Generate the stochastic gradient $g(x_t,z_t) = 4x_t^3+z_t$ (which is an unbiased estimator);
>   - Update the sequence as $x_{t+1} =x_{t} - \eta (4x_{t}^3 + z_t)$.
>
> If we obtain that $\frac{1}{T}\sum_{t=1}^T (4x_{t}^3)^2 \le \varepsilon$ with high probability (as in our paper), which is fully high probability over $z_1,\cdots, z_{T}$, then we are able to deduce that at least one gradient $(4x_{\tau}^3)^2,\tau \in [T] $ is small enough and $x_{\tau}$ is near-stationary. However, if we obtain the fully high probability bound $\frac{1}{T}\sum_{t=1}^T (4x_{t}^3 + z_{t})^2 = \frac{1}{T}\sum_{t=1}^T [(4x_{t}^3)^2 + z_t^2  + 8x_t^3z_t]\le \varepsilon$, then we could not derive any stationary point from this inequality since the term $x_t^3 \cdot z_t$ may be non-zero.
>
> We hope that the above answer and example could address your questions. If you have any further inquiries, we are delighted to discuss them with you.
>
>
>
> Best regards,
>
> Authors.

---

### Official Review · Reviewer_4Xir · 2024-07-09

**Soundness:** 3
**Presentation:** 3
**Contribution:** 3
**Rating:** 6
**Confidence:** 5

**Summary:**

In this paper, the authors analyze the convergence of Adam under milder noise conditions (affline variance) and milder smoothness conditions (both $L$-smoothness and $(L_0,L_q)$-smoothness) and propose a $O(\text{polylog}(T)/\sqrt T)$ convergence rate.

**Strengths:**

This paper analyses the convergence of Adam under milder smoothness conditions compared to the previous work. The result is relatively solid and convincing. The writing structure is also relatively clear.

**Weaknesses:**

1. As the author claimed in their paper, they did not provide numerical experiments in this paper. While this paper is a theoretical paper focusing on the convergence analysis of Adam, some simple numerical experiments aligning with the results will make it more convincing.
2. This paper exhibits a slight lack of novelty. Since after checking out the proof details, I found that the crucial techniques were almost proposed by the previous related works.  However, this weakness is trivial, especially for a theoretical paper, and as I claimed in the Strength part, the result of this paper is solid.

**Questions:**

1. I suggest the authors could recall the readers of the definitions in the proof part, since numerous variables are introduced for proof, like $\mathcal{G}_t$, $M$, $\hat M$, $a_t$, $b_t$ and so on. It's a little inconvenient to check the definition in the previous pages each time.
2. Since coordinate-wise calculations are commonly used in the proof, I suggest the authors could also consider demonstrating their results based on the $L_\infty$ smoothness condition, as discussed in [1]. Also, I wonder about the difference of $(L_0,L_q)$-smoothness and local smoothness.
3. For lemma B.2, I happened also to use this result before and I suggest the author cite [2], as I found this result in lemma A.2 of [2].
4. How do the authors use the Cauchy-Schwarz inequality in the third inequality of line 740? Is this simply derived from $ab \leq 1/4a^2 + b^2$ ? (Here $a, b$ are both scalars).
5. In formula (58), line 557, where the last $\sqrt{\log}$ term comes from?
6. What's the meaning of formulas (59) and (60) since $G^2 \sim O(\text{polylog}T)$ has been claimed in formula (7)

[1] Balles, Lukas, Fabian Pedregosa, and Nicolas Le Roux. "The geometry of sign gradient descent." arXiv preprint arXiv:2002.08056 (2020).

[2] Zou, Difan, et al. "Understanding the Generalization of Adam in Learning Neural Networks with Proper Regularization." The Eleventh International Conference on Learning Representations.

**Limitations:**

No limitations.

---

> ### Author Rebuttal · Authors · 2024-08-06
>
> Thanks a lot for your valuable feedback and suggestions!
>
> **Response to Weakness**
>
> 1. Indeed, numerical results on Adam with/without corrective terms could be found in [10]. We also perform a simple experiment in Table 1 (the attached PDF), which roughly aligns with the result.
>
> 2. Thanks a lot for your suggestions and understanding of our paper.  For the proof novelty of this paper, please refer to the global rebuttal.
>
> **Response to  Question**
>
> 1. We will recall the notational definitions at the beginning of each proof part accordingly in the revision.
>
> 2. We appreciate the suggestion of the interesting topic to study convergence under $L_{\infty}$-smooth and the interesting work [1] you mentioned, which will be carefully cited and commented in the revision. Due to the limited time and space, we are not able to prove convergence under $L_{\infty}$-smoothness, and we shall leave it as a future work. \
> The major difference between local smoothness and  $(L_0,L_q)$-smoothness arises from the additional gradient norm in $(L_0,L_q)$-smoothness. If the gradient norm is globally bounded such that $||\nabla f(x)||\le G$, then $(L_0,L_q)$-smoothness implies local smoothness of $f$ as far as we understand, since following from $(L_0,L_q)$-smoothness: for any $x$ and $y \in B(x,1/L_q)$,
> 	$$
> 	||\nabla f(y) - \nabla f(x)|| \le (L_0 + L_q G^q)||y - x||.
> 	$$
> But in general case, $(L_0,L_q)$-smoothness could be considered weaker than local smoothness.
>
> 3. Thanks a lot for your reminder. We will add its citation in front of Lemma B.2 in our paper accordingly.
>
> 4. It's exactly derived from $ab \le a^2/4 + b^2$.  We will change 'Cauchy-Schwarz inequality' as '$ab \le a^2/4 + b^2,\forall a,b>0$' in Line 740 of the revision.
>
> 5. Note that there is $\mathcal{G}_T$ in Line 556, and recalling the definition from Eq. (18) in Line 229, the additional $\log$ term comes from $\mathcal{M}_T = \sqrt{\log(eT/\delta)}$. We will make this clearer in the revision.
>
> 6. [Eq. (59), Eq. (60)] prove that when $\beta_2 = 1-c/T$, $\log (T/\beta_2^T)$ in (49) is smaller than $\mathcal{O}(\log T)$ and further prove $G^2 \sim  \mathcal{O}(\text{poly}(\log T))$. In Line 562, we will revise as "Therefore combining (59), (60) and (49), we could verify the order in (7). Also, in Line 525, we will replace "as in  Theorem 3.1" with "given by (49)".

---

> > ### Comment · Reviewer_4Xir · 2024-08-09
> >
> > Thanks to the authors for their clarifications. There might exist some minor concerns about novelty and experiment results, But as I claimed in my review, I think it's a good paper, and I'm happy to retain both scores of rating and confidence. Moreover, I would like to mention that analyzing the Adam from the $\ell_\infty$ norm might be interesting, since the geometry of Adam might align with the $\ell_\infty$ norm better as studied in [1, 2, 3].
> >
> > [1] Kunstner, F., Chen, J., Lavington, J. W. and Schmidt, M. (2023). Noise is not the main factor behind the gap between sgd and adam on transformers, but sign descent might be. ICLR.
> >
> > [2] Xie, S. and Li, Z. (2024). Implicit bias of adamw: $\ell_\infty$ norm constrained optimization. ICML.
> >
> > [3] Zhang, C., Zou, D. and Cao, Y. (2024). The Implicit Bias of Adam on Separable Data. arXiv: 2406.10650.

---

> > > ### Author Response · Authors · 2024-08-09
> > > **Thank you for your positive feedback**
> > >
> > > We thank Reviewer 4Xir very much for his positive comment and prompt reply.  We also thank Reviewer 4Xir for the constructive comments on investigating Adam with the $\ell_\infty$ norm and we agree that it is important.  We thank Reviewer 4Xir for the interesting references which we think are interesting and can better enhance the understanding of Adam.

---

### Official Review · Reviewer_J5iU · 2024-07-13

**Soundness:** 4
**Presentation:** 4
**Contribution:** 3
**Rating:** 6
**Confidence:** 3

**Summary:**

This paper studies the high-probability convergence of Adam in the non-convex setting under relaxed assumptions. The authors consider a general noise condition that governs affine, sub-Gaussian, and bounded noise conditions. They also consider a generalized smoothness condition motivated by language model experiments. Under these assumptions, they obtain a convergence rate of $\text{poly}(\log T/\delta)/T$, where $T$ is the number of iterations and $\delta$ is the confidence level.

**Strengths:**

1. Their result look novel and significant. They have shown the high-probability convergence of Adam under relaxed conditions than all previous papers.
2. The proofs look correct.
3. The paper is well-written and results are clearly presented.

**Weaknesses:**

1. One major concern is, by choosing $\beta_2=1-1/T$, does the author essentially reduce Adam to SGD with momentum, as this makes $v_t$ almost a constant? Btw, I think for [18] and [23] in Table 1, $\beta_1$ should be $1-1/\sqrt{T}$. Please also check other rows more carefully.

I will increase the score if this concern is addressed.

**Questions:**

1. The term “affine variance noise" is confusing to me. I think it should only refer to Equation (2), not Equation (3), as "variance" is defined as an expectation. If I understand correctly, the term in Line 3 refers to (3), which means the condition (A3) is actually stronger than (2), right?
2. Why in Table 1 you did not include [19] which you discussed in Section 5.1?
3. The rate in Theorem 4.1 is dimension dependent, whereas the rate in Theorem 3.1 is dimension free. Do you think it is something fundamental in the relaxed smoothness condition?

**Limitations:**

See weaknesses and questions above

---

> ### Author Rebuttal · Authors · 2024-08-06
>
> We thanks a lot for the reviewer's effort and valuable suggestions on our manuscript.
>
> **Response to Weakness: We clarify as follows, and we will revise the presentation issue accordingly.**
>
> - First, in the non-asymptotic analysis (see Table 1) which we follows from, the total iteration number $T$ is treated as finite and fixed. Thereby, during the proof, the following two facts are essential:  $\beta_2=1-1/T$ as a constant that is not equal to 1; $v_t$ as the exponential moving average of past gradients instead of a constant. Several of our key results, such as Lemma B.2, Lemma B.3, Lemma B.7, and Lemma B.12, are based on these premises. This configuration distinguishes our approach from merely reducing Adam to SGD with momentum, as the proof techniques applicable to SGD with momentum do not directly extend to Adam.
>
> -  Second, as we mention in Table 1, at least two works [10, 38] consider the same setup of $\beta_2$ as ours. In addition, [49] have indicated that a $\beta_2$ value closed to one facilitates convergence, whereas a smaller $\beta_2$ may lead to divergence. This observation also somehow supports the validity of our choice for $\beta_2$.
>
> -  We will revise $\beta_1$ setup in two references accordingly.
>
> **Response to Questions**
>
> 1. Indeed,  "affine variance noise" is originally of expected version in Eq. (2). In this paper, in order to derive a stronger high-probability convergence bound instead of an expected bound, we require the stronger almost surely version in Eq. (3), which has been studied in, e.g., [2, 23].  We will add  'almost surely' in Line 3 to distinguish from the expected affine variance noise.
>
> 2. Thanks a lot for the reminder of this citation, which will be included in Table 1 in the revision.
>
> 3. We ignore the $d$ factor in the two upper bounds of the main theorem, to make the results concise.  All these two results are dimension dependent, see the proof parts. Furthermore, in Theorem 3.1, if $C_0\sim \mathcal{O}(1/d) $, then $G^2 \sim \mathcal{O}(1)$ and the convergence bound is of order $\mathcal{O}(d)$ w.r.t. $d$. In Theorem 4.1, if $E_0\sim \mathcal{O}(1/d)$, then $\hat{H} \sim \mathcal{O}(1)$ and the convergence bound is of order $\mathcal{O}\left(d\right)$ w.r.t. $d$. In this sense, there is no fundamental difference between the standard and relaxed smoothness. We will add these remarks in the appendix of the revision.

---

> > ### Comment · Reviewer_J5iU · 2024-08-12
> >
> > Thank you for the detailed explanations!
> >
> > I understand that $v_t$ is not a constant for $\beta_2=1-1/T$, but an exponential moving average of past gradient squared. Say $v_t=w_0 g_0^2+\cdots+w_{t} g_t^2$. One can show that $w_0=(1-1/T)^{t}\ge 1/e=O(1)$, which means $v_t$ never forgets the initial gradients. I feel this may simplify the analysis in some way. It should be more interesting to extend the result to a smaller $\beta_2$ like $1-1/\sqrt{T}$. That being said, I agree that the analysis is already non-trivial compared to SGDM, and would like to increase my score.

---

> ### Author Response · Authors · 2024-08-12
>
> Thank you so much for your reply and generosity in raising the score. We thank a lot for the valuable suggestion from the reviewer and agree that considering the smaller $\beta_2$ setup would be a quite interesting topic for the future investigation. We shall add a brief remark related to this point after our main results.
>
> Best regards,
>
> Authors.

---

### Author Rebuttal · Authors · 2024-08-06

We thank all reviewers for their comments and suggestions! In the global rebuttal, we have:
- **summarization of proof novelty**
- **a simple experiment in the attached PDF as supplementary material for our main results.**

While our proof borrows some ideas from [42, 10, 14, 2, 38, 19] as we mention in Line 221, **our proof novelty** lies in the following points  (and a few others):
- a new type of proxy step-size in Eq. (22) to break the entanglement of stochastic gradients and adaptive step-sizes in Eq. (21), and to handle the error brought by the affine variance noise in Line 256;
- a new decomposition to handle the mismatch between stochastic gradients and the momentum, please also refer to Eq. (19), Line 261 and Line 262;
- some new estimations to handle the corrective terms in Adam, which are seldom considered in previous works, please also refer to Lemma B.12;
- an induction argument to control the potential unbounded smooth parameter in the generalized smooth case, please also refer to Proposition C.6 and Proposition C.7.

---

### Comment · Area_Chair_R9UT · 2024-08-14
**Discussions**

Dear reviewers,

Thanks very much for your great efforts in the review process. Please read the authors' response and confirm it.  You are encouraged to discuss with all of us if you have any concerns.

Thanks,
AC

---

### Decision · Program_Chairs · 2024-09-25

**Decision:**

Accept (poster)

**Comment:**

All reviewers think the paper is interesting.  I recommend an acceptance.